# Mammo-AGE: deep learning estimation of breast age from mammograms

Xin Wang [1,2], Tao Tan [3,4] ✉, Yuan Gao [1,2,4], Hong-Yu Zhou [5], Tianyu Zhang [1,2,4], Luyi Han [1,4], Antonio Portaluri [1,4,6], Eric Marcus[7], Chunyao Lu [1,4], Caroline A. Drukker[8], Jonas Teuwen[4,7], Regina Beets-Tan[1,2], Shanshan Wang [9], Nico Karssemeijer[10] & Ritse Mann [1,4]

Biological age is an important indicator of organ functions and health. Although mammograms are widely used in breast cancer screening, the potential of mammogram-based biological age predictors remains under-explored. Here, we propose a deep learning model to estimate the biological age of the breast using healthy mammograms. The model is developed on three large datasets and externally validated on two additional datasets, encompassing 95,826 mammograms from 44,497 women aged 18 to 98 years. It demonstrates accurate age estimation (mean absolute error: 4.2 − 6.1 years) with strong correlation to chronological age. Predicted breast age stratifies breast cancer risk similarly to chronological age. Occlusion analysis, employed for model interpretation, reveals the aging-related pattern of the breast. The breast age gap (the difference between system-bias-corrected breast age and chronological age) may reflect breast health status. Breast cancer patients show higher breast age gaps than the healthy population. In two longitudinal datasets, larger breast age gaps are associated with increased future breast cancer risk, with hazard ratios of 1.013 − 1.022. Furthermore, we finetune the model specifically for downstream breast cancer diagnosis and risk prediction. Our approach outperforms other comparative methods, showing its potential for supporting both early detection and personalized screening strategies.

Breast cancer ranks among the most prevalent cancers globally, contributing significantly to cancer-related mortality among women[1–3]. Statistics reveal a pronounced increase in breast cancer incidence and mortality with advancing age, aligning with trends observed in many cancers and chronic diseases[4,5]. Breast tissue aging is also affected by reproductive factors, such as age at menarche,

pregnancies, and age at menopause. Thus, the relation between calendar age and breast cancer risk is not linear[6]. However, given the strong "residual" association between aging and breast cancer, population-wide mammography screening regimens primarily rely on age-based eligibility criteria for early detection and mortality reduction[7].

[1]Department of Radiology, The Netherlands Cancer Institute, Amsterdam, The Netherlands. [2]GROW School, Maastricht University, Maastricht, The Netherlands. [3]Faculty of Applied Sciences, Macao Polytechnic University, Macao, SAR, China. [4]Department of Medical Imaging, Radboud University Medical Centre, Nijmegen, The Netherlands. [5]School of Biomedical Engineering, Tsinghua University, Beijing, China. [6]Department of Biomedical Sciences and Morphologic and Functional Imaging, Policlinico Universitario "G. Martino", University of Messina, Policlinico "G. Martino", Messina, Italy. [7]Department of Radiation Oncology, The Netherlands Cancer Institute, Amsterdam, The Netherlands. [8]Department of Surgical Oncology, The Netherlands Cancer Institute, Amsterdam, The Netherlands. [9]Paul C. Lauterbur Research Center for Biomedical Imaging, Shenzhen Institutes of Advanced Technology, Chinese Academy of Sciences, Shenzhen, China. [10]Department of Medical Imaging, Radboud University Medical Center, Nijmegen, The Netherlands. ✉e-mail: taotanjs@gmail.com

Age, while correlated with the accumulation of biological changes, it is a variable factor, leading to substantial variability in health outcomes among individuals sharing the same chronological age[8,9]. Chronological age, therefore, serves as an imperfect surrogate for the dynamic biological changes over time. Previous research has developed multiple epigenetic clocks[10,11], which correlate with chronological age across various tissues and organs, including breast tissue[4,12,13]. Despite its strong correlation, the need for tissue, the high cost, and the time-consuming nature of this measurement limit its utility as a biomarker for large-scale population screening or longitudinal studies.

In recent years, deep learning techniques and the availability of extensive medical imaging datasets have spurred investigations into estimating biological age. Typically, biological age predictors are developed by training deep learning models to predict chronological age, with the model's output interpreted as the individual's biological age. Such predictors have been established for various organs using diverse imaging modalities, including brain, heart, liver, and pancreas magnetic resonance images (MRIs), chest, bone, full-body X-rays, eye fundus, and facial images[14–22]. Despite the fact that in breast cancer screening programs millions of mammograms[23–26] are generated every year from breast cancer screening programs, and the association of breast cancer risk and age is evident, the potential for mammogram-based biological age predictors has not been explored.

The quest for viable aging biomarkers extends to their association with age-related morbidity and mortality. Hypotheses suggesting that estimated age from different organs can measure an individual's biological age have gained support. The difference between estimated and actual chronological age, known as the "age gap," is believed to reflect variation in past rates of aging[27]. Observations have linked positive age gaps, termed age acceleration, to a heightened risk of mortality and various aging-related diseases, including heart disease, metabolic syndrome, and certain cancers[8,15,19,21,28,29]. Previous research has shown that DNA methylation-based biological age estimates exceeding chronological age are associated with an increased risk of developing breast cancer[4]. However, the clinical significance of age discrepancy derived from mammograms for breast cancer risk remains unclear. Therefore, the implication of estimated age derived from mammograms warrants further exploration and characterization. Unlike direct breast cancer prediction models[30] using mammograms, the goal of this exploration is to estimate breast age as a specific and continuous biomarker of cumulative tissue aging, potentially offering a non-invasive and scalable alternative to DNA methylation-based molecular aging clocks[4].

In this study, we hypothesize that estimated age from mammograms, utilizing deep learning, can indicate breast tissue aging status. Accordingly, we propose a deep learning model specifically designed for predicting age from mammograms, referred to as the "Mammo-AGE" model. Leveraging a substantial population-based screening mammogram dataset alongside publicly available datasets, we operate under the assumption that biological age accurately mirrors chronological age in individuals undergoing normal aging. Therefore, we use normal mammograms from both in-house and public datasets of healthy women with known chronological age to develop and validate a deep learning model to predict the biological age of the breast. Moreover, this study investigates the mammogram-based breast age acceleration and its association with breast cancer risk. Applying this baseline to the test dataset facilitates the assessment of the disparity between predicted biological age and chronological age, offering insights into pathological changes deviating from normal aging. Importantly, it provides practical insights into breast health status, highlighting the association between elevated breast age gap and breast cancer risk. Further, we investigate the benefits of this new biomarker in various situations, including as an aging biomarker analysis and for breast disease-related downstream tasks.

## Results

### Overview of the study design

Inspired by recent studies on the age of human organs, we hypothesized that breasts exhibit aging changes, leading us to utilize deep learning techniques to understand normal breast aging patterns through mammogram images of healthy women. In this study, we introduce a mammogram-based breast age prediction (Mammo-AGE) model, as depicted in Fig. 1A. This model is designed to learn from mammograms and predict the chronological age of women, termed breast age. Our proposed model takes advantage of prior knowledge regarding the consistency of age relationships observed in standard multi-view mammograms (craniocaudal (CC) and mediolateral oblique (MLO) views of both breasts). By incorporating information from multi-view mammograms rather than using a single view, the model can enhance its ability to capture features related to breast aging. To achieve this, we draw inspiration from the newly proposed global-local transformer (GLT) framework[17] and introduce an instance-bag transformer. Further details of the deep learning model design are provided in the Methods section and Supplementary Fig. 1. This transformer consists of self-attention and cross-attention blocks, facilitating the integration of information from multiple views. We also introduce multi-task learning of density prediction for auxiliary training. Breast density, defined as the proportion of fibroglandular tissue to fatty tissue in the breast, typically decreases with increasing age[31]. Regarding the constraint model, the mean squared error (MSE) loss function was leveraged to constrain the prediction age to match the ground truth age. In addition to constraining the predicted age directly, we also utilized probabilistic ordinal embedding (POE) loss[32] functions and mean-variance (MVL) loss[33] functions to guide the model in learning the reasonable ordinal distribution of the latent feature space and predicted probabilities to align with biologically meaningful aging trends. We enhance overall performance by ensembling five models with different backbones, including ResNet-18[34], ResNet-50[34], EfficientNet-b0[35], ConvNeXt-tiny[36], and DenseNet-121[37]. This ensemble is achieved through the weighted averaging of the five predicted ages, thereby improving the robustness and accuracy of the Mammo-AGE model.

To model the natural aging process of breasts, we developed the Mammo-AGE model using mammograms obtained from healthy women. A comprehensive dataset was formed by combining mammographic images from the Inhouse, VinDr, and RSNA datasets, resulting in 35,068 examinations with 140,272 mammograms (Fig. 1B). We employed a five-fold cross-validation methodology at the patient level for training and validating the model on 80% of the combined dataset, with the remaining 20% used as a separate test set (see the Methods section for further details). Notably, our in-house dataset encompassed longitudinal data, wherein women underwent multiple mammogram examinations at different time points. To enhance the model's ability to learn aging-related changes more effectively, we utilized all available series of mammograms from each participant in the training set. By following this approach, we aimed to develop a robust model that accurately captures and predicts the normal aging patterns of breasts. To assess the model's generalization ability, external validation was performed using two large public datasets, EMBED and CMMD. The characteristics of mammogram examinations for each dataset are summarized in Table 1. The datasets encompass a broad age range (18 - 98 years) and diverse regions, with varying distributions of breast density categories. Consequently, we comprehensively evaluated Mammo-AGE's performance in breast age estimation (Fig. 1C) and analyzed the relationship between the breast age gap and breast cancer (Fig. 1D). Furthermore, we explored the clinical applicability of Mammo-AGE by assessing its utility in downstream tasks, including diagnostic classification and future cancer risk prediction (Fig. 1E).

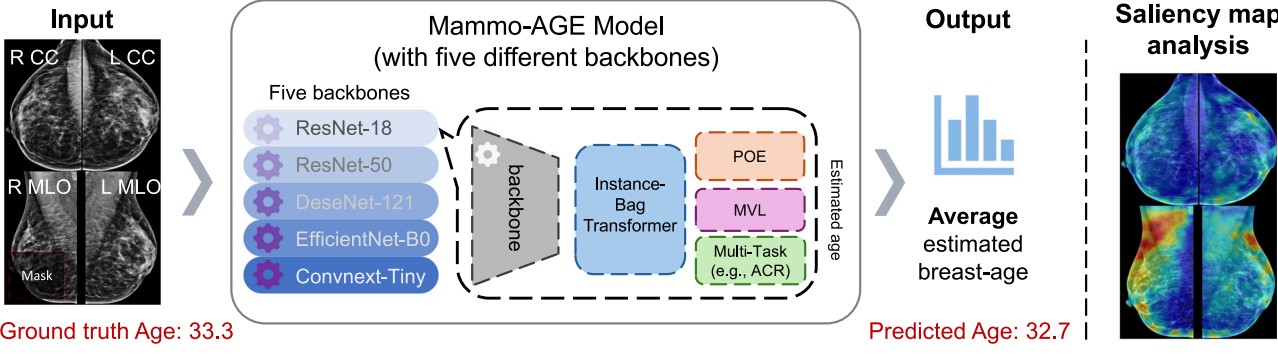

**A. Mammo-AGE Model**

Ground truth Age: 33.3

Predicted Age: 32.7

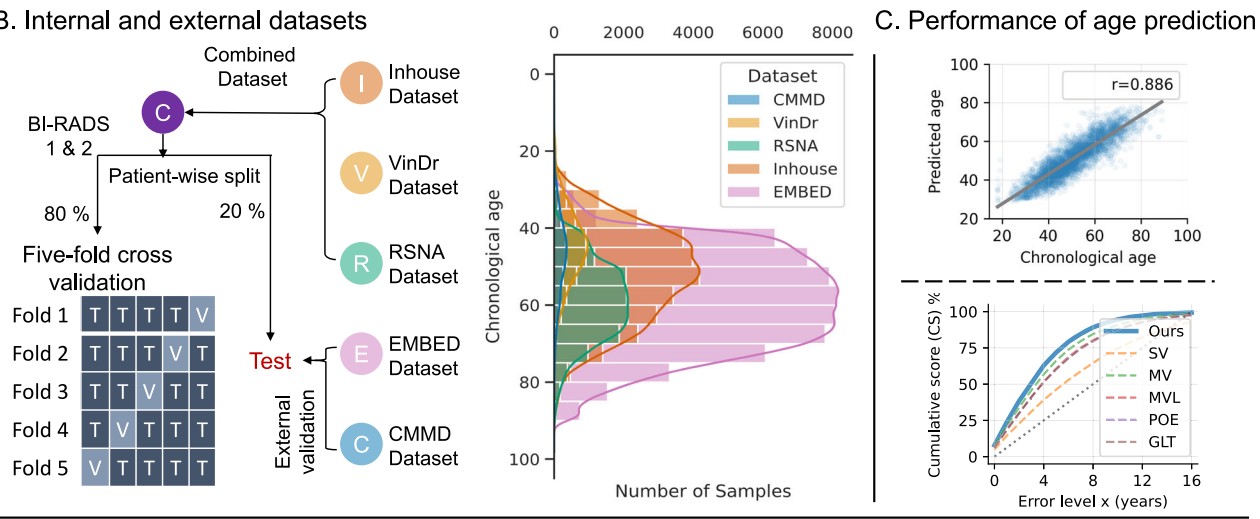

**B. Internal and external datasets**

**C. Performance of age prediction**

**D. Breast age gap analysis after bias-correction**

**E. Mammo-AGE model for downstream tasks**

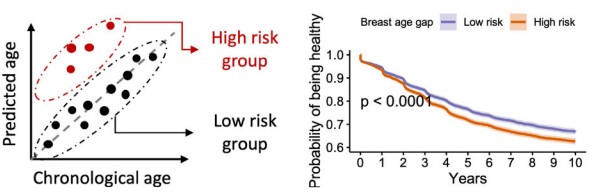

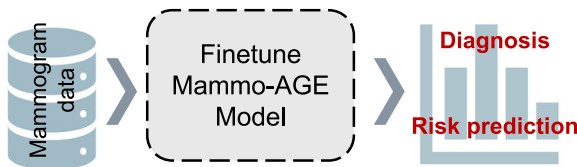

**Fig. 1 | Overview design of the study. A** Schematic architecture of the proposed Mammo-AGE model and illustration of the occlusion analysis. The model utilizes four-view mammograms (CC and MLO views of both breasts) as input to predict breast age. An instance-bag transformer, inspired by the global-local transformer framework, integrates self-attention and cross-attention mechanisms to fuse information across views. The model incorporates multi-task learning for breast density prediction and uses a combination of cross-entropy (CE) loss, mean-variance (MVL) loss, and probabilistic ordinal embedding (POE) loss to optimize learning. The model outputs age predictions, with saliency maps highlighting age-relevant regions. Five different backbone-based (ResNet-18, ResNet-50, ConvNeXt-Tiny, EfficientNet-B0, and DenseNet-121) models were ensembled by weighted averaging of predicted ages. The detailed description of the network architecture is provided in Supplementary Fig. 1. **B** Age distributions of the internal and external datasets. The combined dataset was split patient-wise for five-fold cross-validation. External validation was conducted on additional datasets. **C** Performance evaluation of the Mammo-AGE model on age prediction. **D** Association analysis between the breast age gap (predicted age minus chronological age) and breast cancer after bias correction. Specifically, Kaplan–Meier (KM) curves analysis of the inhouse cohort ($n = 10,392$) demonstrates that individuals with a higher breast age gap have a significantly (two-sided log-rank test) lower healthy probability over time compared to those with a lower breast age gap. Shaded areas represent the 95% confidence intervals. **E** Evaluation of the Mammo-AGE model on downstream clinical tasks, including breast cancer diagnosis and risk prediction.

## Deep learning model performance for breast age prediction

To evaluate the Mammo-AGE model, we employed four metrics: mean absolute error (MAE), cumulative score (CS) based on a threshold of error within five years ($\alpha = 5\,y$), the Pearson correlation coefficient ($r$), and the Spearman's rank correlation coefficient ($r_s$) between predicted breast age and chronological age. The overall performance of our proposed method (Ours ensembled; ensembled predictions from five models with different backbones) across five folds yielded MAE =

$4.174 \pm 0.028$, CS ($\alpha = 5\,y$) = $72.2\% \pm 0.7\%$, $r = 0.891 \pm 0.001$, and $r_s = 0.896 \pm 0.001$ on the combined test set (Table 2). Scatterplots depicting the test set predictions versus chronological age are presented in Fig. 2. To assess whether the trained model presents a dataset-specific bias, we also evaluated our model's performance on cross-validation datasets and external-validation datasets, respectively. The overall performance on the Inhouse, RSNA, and VinDr test sets was MAE = $4.090 \pm 0.047$, MAE = $4.548 \pm 0.022$, and MAE = $4.075$

**Table 1 | Characteristics for the different datasets**

| | Internal datasets | | | External datasets | |
|---|---|---|---|---|---|
| Dataset | Inhouse | RSNA | VinDr | EMBED | CMMD |
| Total *n* exams (patients) | 18,828 (5048) | 11,905 (11,905) | 4335 (4335) | 58,983 (21,434) | 1775 (1775) |
| Country | Netherlands | Australia and America | Vietnam | America | China |
| **Age** | | | | | |
| Mean (SD); years | 51.97 (11.67) | 58.64 (9.88) | 45.10 (9.77) | 58.95 (11.94) | 47.56 (10.81) |
| Minimum-maximum | 23–89 | 26–89 | 18–88 | 19–89 | 18–87 |
| **Breast density** | | | | | |
| ACR A | | | | | |
| Total *n* exams | 1187 (6.3%) | 553 (4.6%) | 22 (0.5%) | 6144 (10.4%) | NA |
| Age, mean (SD); years | 56.37 (10.87) | 60.34 (10.61) | 59.59 (9.80) | 62.63 (11.17) | NA |
| ACR B | | | | | |
| Total *n* exams | 10,843 (57.6%) | 2509 (21.1%) | 381 (8.8%) | 24,603 (41.7%) | NA |
| Age, mean (SD); years | 53.83 (11.50) | 59.98 (11.06) | 55.66 (9.86) | 61.19 (11.41) | NA |
| ACR C | | | | | |
| Total n exams | 4919 (26.1%) | 2432 (20.4%) | 3319 (76.6%) | 24,663 (41.8%) | NA |
| Age, mean (SD); years | 48.99 (11.14) | 54.80 (10.99) | 44.94 (8.87) | 56.83 (11.79) | NA |
| ACR D | | | | | |
| Total *n* exams | 1857 (9.9%) | 307 (2.6%) | 613 (14.1%) | 3217 (5.5%) | NA |
| Age, mean (SD); years | 46.02 (10.48) | 50.41 (10.14) | 38.90 (8.35) | 51.04 (11.44) | NA |
| Unknown | | | | | |
| Total *n* exams | 22 (0.1%) | 6104 (51.3%) | NA | 238 (0.4%) | 1775 (100.0%) |
| Age, mean (SD); years | 64.14 (13.18) | 59.87 (8.06) | NA | 57.68 (13.86) | 47.56 (10.81) |
| **Breast cancer** | | | | | |
| Negative | | | | | |
| Total *n* exams | 18,213 (96.7%) | 11,419 (95.9%) | 4250 (98.0%) | 58,430 (99.1%) | 465 (26.2%) |
| Age, mean (SD); years | 51.78 (11.61) | 58.43 (9.83) | 44.86 (9.57) | 58.91 (11.93) | 42.36 (9.46) |
| Positive | | | | | |
| Total *n* exams | 615 (3.3%) | 486 (4.1%) | 85 (2.0%) | 553 (0.9%) | 1310 (73.8%) |
| Age, mean (SD); years | 57.61 (11.97) | 63.49 (9.87) | 57.06 (11.89) | 62.31(13.31) | 49.41 (10.66) |
| **Race** | | | | | |
| White | | | | | |
| Total *n* exams | 6146 (32.6%) | NA | - | 25,340 (43.0%) | - |
| Age, mean (SD); years | 48.92 (10.90) | NA | - | 61.15 (12.00) | - |
| African | | | | | |
| Total *n* exams | 55 (0.3%) | NA | - | 25,899 (43.9%) | - |
| Age, mean (SD); years | 52.13 (14.78) | NA | - | 58.43 (11.63) | - |
| Asian | | | | | |
| Total *n* exams | 108 (0.6%) | NA | 4335 (100.0%) | 3295 (5.6%) | 1775 (100.0%) |
| Age, mean (SD); years | 48.49 (10.57) | NA | 45.10 (9.77) | 54.37 (10.99) | 47.56 (10.81) |
| Other or Unknown | | | | | |
| Total n exams | 12,519 (66.5%) | NA | - | 4449 (7.5%) | - |
| Age, mean (SD); years | 53.50 (11.73) | NA | - | 52.76 (10.56) | - |
| **Manufacturer information** | | | | | |
| | Selenia dimensions | NA | Mammomat inspiration | Lorad selenia | NA |
| | Lorad selenia | | GIOTTO CLASS | Selenia dimensions | |
| | Hologic selenia | | GIOTTO IMAGE 3DL | Senograph 2000D ADS_17.4.5 | |
| | | | Planmed nuance | Senograph 2000D ADS_17.5 | |
| | | | | Senographe essential VERSION ADS_53.40 | |
| | | | | Senographe pristina | |

*ACR* the American College of Radiology (ACR), *BI-RADS* breast density categories. The density grades are assessed by radiologists during clinical interpretation, *ACR A* (mostly fatty), *ACR B* (scattered fibroglandular), *ACR C* (heterogeneously dense), and *ACR D* (extremely dense). *SD* standard deviation, NA not Applicable.

**Table 2 | Comparison of model performance across different baseline models, individual backbone-based Mammo-AGE models, and the ensembled Mammo-AGE model**

| | | | Combined dataset | Inhouse dataset | RSNA dataset | VinDr dataset | EMBED dataset | CMMD dataset |
|---|---|---|---|---|---|---|---|---|
| **Test N sample** | | | 3946 | 2489 | 750 | 707 | 33,040 | 465 |
| **Age range** | | | 18-89 | 23-87 | 33-89 | 18-83 | 19-89 | 18-84 |
| | **Method** | | | | | | | |
| Compare with different baseline models | Baseline ResNet-18[34] | MAE | 7.429±0.156 | 6.988±0.134 | 8.362±0.300 | 7.991±0.373 | 9.741±0.338 | 7.213±0.354 |
| | | Pearson Correlation (r) | 0.676±0.017 | 0.718±0.011 | 0.591±0.028 | 0.509±0.031 | 0.600±0.024 | 0.438±0.048 |
| | | Spearman Correlation (rs) | 0.704±0.017 | 0.740±0.010 | 0.602±0.025 | 0.509±0.021 | 0.610±0.025 | 0.400±0.034 |
| | | CS (α = 5 year) | 45.6%±1.0% | 47.0%±1.0% | 40.4%±1.4% | 46.0%±2.1% | 36.0%±1.3% | 49.4%±2.1% |
| | Baseline ResNet-50[34] | MAE | 9.029±0.175 | 8.756±0.181 | 10.51±0.308 | 8.411±0.091 | 12.13±0.357 | 8.620±0.381 |
| | | Pearson correlation (r) | 0.593±0.042 | 0.613±0.038 | 0.532±0.036 | 0.413±0.018 | 0.543±0.043 | 0.376±0.067 |
| | | Spearman correlation (rs) | 0.638±0.030 | 0.648±0.032 | 0.549±0.035 | 0.427±0.030 | 0.560±0.055 | 0.318±0.060 |
| | | CS (α = 5 year) | 37.1%±0.7% | 37.1%±0.9% | 32.1%±0.7% | 42.4%±0.3% | 28.4%±0.6% | 41.5%±2.3% |
| | Baseline DenseNet-121[37] | MAE | 8.144±0.313 | 7.832±0.354 | 9.245±0.259 | 8.070±0.254 | 10.76±0.293 | 8.500±0.357 |
| | | Pearson correlation (r) | 0.657±0.012 | 0.685±0.015 | 0.540±0.020 | 0.423±0.036 | 0.546±0.024 | 0.443±0.035 |
| | | Spearman Correlation (rs) | 0.705±0.010 | 0.730±0.010 | 0.556±0.016 | 0.454±0.041 | 0.567±0.024 | 0.367±0.024 |
| | | CS (α = 5 year) | 41.2%±2.0% | 41.7%±2.3% | 35.6%±1.7% | 45.2%±1.2% | 32.2%±1.2% | 41.7%±2.5% |
| | Baseline ConvNeXt-Tiny[36] | MAE | 6.152±0.245 | 5.659±0.213 | 6.656±0.252 | 7.342±0.430 | 7.633±0.395 | 6.956±0.250 |
| | | Pearson correlation (r) | 0.741±0.018 | 0.778±0.015 | 0.672±0.011 | 0.596±0.046 | 0.685±0.018 | 0.473±0.035 |
| | | Spearman correlation (rs) | 0.760±0.017 | 0.792±0.014 | 0.681±0.010 | 0.571±0.049 | 0.694±0.017 | 0.419±0.033 |
| | | CS (α = 5 year) | 55.0%±1.9% | 57.9%±2.0% | 51.4%±1.9% | 48.8%±1.9% | 45.9%±1.9% | 49.2%±1.7% |
| | Baseline EfficientNet-B0[35] | MAE | 6.109±0.128 | 5.741±0.080 | 6.944±0.369 | 6.510±0.340 | 8.019±0.454 | 7.311±0.537 |
| | | Pearson correlation (r) | 0.747±0.013 | 0.774±0.006 | 0.647±0.034 | 0.594±0.014 | 0.658±0.035 | 0.402±0.095 |
| | | Spearman correlation (rs) | 0.769±0.011 | 0.791±0.006 | 0.661±0.036 | 0.604±0.013 | 0.669±0.038 | 0.337±0.089 |
| | | CS (α = 5 year) | 55.3%±1.1% | 57.3%±0.9% | 49.6%±2.5% | 54.5%±3.1% | 44.3%±2.2% | 47.6%±2.5% |
| | Baseline VIT[38] | MAE | 6.998±0.030 | 6.502±0.073 | 7.176±0.065 | 8.546±0.222 | 7.710±0.064 | 6.995±0.220 |
| | | Pearson correlation (r) | 0.672±0.003 | 0.711±0.007 | 0.589±0.010 | 0.480±0.030 | 0.620±0.007 | 0.455±0.026 |
| | | Spearman correlation (rs) | 0.686±0.003 | 0.724±0.006 | 0.593±0.009 | 0.482±0.042 | 0.626±0.006 | 0.392±0.025 |
| | | CS (α = 5 year) | 49.2%±0.3% | 51.6%±0.6% | 47.3%±0.6% | 43.0%±2.1% | 44.4%±0.2% | 48.5%±2.0% |
| | Baseline Swin-VIT[39] | MAE | 6.541±0.058 | 6.120±0.067 | 6.651±0.071 | 7.899±0.402 | 7.152±0.069 | 7.038±0.212 |
| | | Pearson correlation (r) | 0.718±0.004 | 0.748±0.007 | 0.653±0.005 | 0.537±0.027 | 0.675±0.004 | 0.448±0.028 |
| | | Spearman correlation (rs) | 0.729±0.004 | 0.760±0.007 | 0.657±0.005 | 0.524±0.033 | 0.681±0.004 | 0.395±0.023 |
| | | CS (α = 5 year) | 52.2%±0.5% | 54.4%±0.5% | 50.9%±1.0% | 45.9%±2.4% | 47.7%±0.4% | 48.1%±0.9% |
| Different backbone-based Mammo-AGE models | Mammo-AGE (ResNet-18) | MAE | 4.668±0.048 | 4.556±0.062 | 5.108±0.045 | 4.593±0.043 | 5.454±0.052 | 6.974±0.355 |
| | | Pearson correlation (r) | 0.864±0.003 | 0.867±0.003 | 0.800±0.006 | 0.810±0.006 | 0.824±0.004 | 0.598±0.019 |
| | | Spearman correlation (rs) | 0.870±0.003 | 0.873±0.003 | 0.806±0.006 | 0.800±0.005 | 0.833±0.004 | 0.539±0.012 |
| | | CS (α = 5 year) | 66.9%±0.5% | 67.8%±0.8% | 62.9%±0.6% | 67.7%±0.6% | 59.6%±0.5% | 48.0%±3.2% |
| | Mammo-AGE (ResNet-50) | MAE | 4.868±0.076 | 4.742±0.099 | 5.413±0.122 | 4.728±0.046 | 5.702±0.098 | 6.260±0.189 |
| | | Pearson correlation (r) | 0.852±0.003 | 0.854±0.004 | 0.780±0.006 | 0.793±0.005 | 0.807±0.004 | 0.614±0.016 |
| | | Spearman correlation (rs) | 0.860±0.003 | 0.864±0.004 | 0.784±0.009 | 0.781±0.007 | 0.816±0.004 | 0.531±0.018 |
| | | CS (α = 5 year) | 64.7%±0.7% | 65.7%±0.9% | 59.7%±0.8% | 66.3%±0.7% | 57.4%±0.8% | 51.7%±1.7% |
| | Mammo-AGE (ConvNeXt-Tiny) | MAE | 4.312±0.058 | 4.245±0.077 | 4.676±0.075 | 4.162±0.032 | 5.147±0.056 | 6.524±0.348 |

**Table 2 (continued) | Comparison of model performance across different baseline models, individual backbone-based Mammo-AGE models, and the ensembled Mammo-AGE model**

| Method | Combined dataset | Inhouse dataset | RSNA dataset | VinDr dataset | EMBED dataset | CMMD dataset |
|---|---|---|---|---|---|---|
| Test N sample | 3946 | 2489 | 750 | 707 | 33,040 | 465 |
| Age range | 18–89 | 23–87 | 33–89 | 18–83 | 19–89 | 18–84 |
| Pearson correlation (r) | 0.884 ± 0.003 | 0.884 ± 0.004 | 0.839 ± 0.003 | 0.835 ± 0.002 | 0.844 ± 0.003 | 0.661 ± 0.004 |
| Spearman correlation ($r_s$) | 0.889 ± 0.003 | 0.890 ± 0.004 | 0.844 ± 0.004 | 0.832 ± 0.002 | 0.850 ± 0.003 | 0.602 ± 0.004 |
| CS (α = 5 year) | 70.6% ± 0.8% | 71.1% ± 0.8% | 66.5% ± 0.9% | 73.4% ± 1.3% | 62.2% ± 0.5% | 51.6% ± 2.9% |
| **Mammo-AGE (DenseNet-121)** MAE | 4.591 ± 0.035 | 4.498 ± 0.041 | 4.955 ± 0.093 | 4.529 ± 0.075 | 5.538 ± 0.095 | 7.289 ± 0.738 |
| Pearson correlation (r) | 0.869 ± 0.002 | 0.869 ± 0.002 | 0.811 ± 0.010 | 0.814 ± 0.008 | 0.821 ± 0.006 | 0.650 ± 0.028 |
| Spearman correlation ($r_s$) | 0.874 ± 0.001 | 0.876 ± 0.002 | 0.816 ± 0.007 | 0.803 ± 0.007 | 0.828 ± 0.006 | 0.579 ± 0.019 |
| CS (α = 5 year) | 67.2% ± 0.5% | 67.9% ± 0.7% | 64.2% ± 1.9% | 67.9% ± 1.2% | 58.7% ± 0.8% | 44.9% ± 4.5% |
| **Mammo-AGE (EfficientNet-B0)** MAE | 4.699 ± 0.053 | 4.519 ± 0.091 | 5.399 ± 0.160 | 4.588 ± 0.122 | 5.635 ± 0.140 | 6.822 ± 0.259 |
| Pearson correlation (r) | 0.861 ± 0.005 | 0.867 ± 0.003 | 0.771 ± 0.032 | 0.808 ± 0.007 | 0.805 ± 0.015 | 0.616 ± 0.019 |
| Spearman correlation ($r_s$) | 0.870 ± 0.003 | 0.878 ± 0.004 | 0.781 ± 0.022 | 0.797 ± 0.006 | 0.814 ± 0.011 | 0.540 ± 0.021 |
| CS (α = 5 year) | 66.8% ± 0.9% | 68.5% ± 1.3% | 61.0% ± 0.7% | 66.7% ± 1.0% | 58.6% ± 0.6% | 48.4% ± 1.5% |
| **Ours ensembled Mammo-AGE** MAE | **4.174 ± 0.028** | **4.090 ± 0.047** | **4.548 ± 0.022** | **4.075 ± 0.010** | **5.010 ± 0.040** | **6.103 ± 0.222** |
| Pearson correlation (r) | **0.891 ± 0.001** | **0.892 ± 0.002** | **0.844 ± 0.001** | **0.847 ± 0.002** | **0.855 ± 0.002** | **0.705 ± 0.009** |
| Spearman correlation ($r_s$) | **0.896 ± 0.001** | **0.898 ± 0.002** | | **0.849 ± 0.002** | **0.843 ± 0.003** | **0.861 ± 0.002** |
| CS (α = 5 year) | 68.2% ± 0.7% | 73.8% ± 0.8% | 63.4% ± 0.5% | 54.8% ± 1.5% | 58.6% ± 0.6% | 48.4% ± 1.5% |
| **Ours ensembled Mammo-AGE (Bias-corrected)** MAE | 3.698 ± 0.039 | 3.618 ± 0.045 | 4.155 ± 0.062 | 3.497 ± 0.042 | 4.131 ± 0.060 | 4.568 ± 0.168 |
| Pearson correlation (r) | 0.929 ± 0.002 | 0.929 ± 0.002 | 0.898 ± 0.004 | 0.904 ± 0.003 | 0.907 ± 0.003 | 0.818 ± 0.009 |
| Spearman correlation ($r_s$) | 0.932 ± 0.002 | 0.933 ± 0.002 | 0.901 ± 0.004 | 0.900 ± 0.002 | 0.911 ± 0.003 | 0.763 ± 0.011 |
| CS (α = 5 year) | 77.6% ± 0.4% | 78.5% ± 0.6% | 72.5% ± 1.2% | 79.8% ± 0.6% | 72.3% ± 0.5% | 66.2% ± 2.6% |

Additional values appearing in the table near the ensembled / bias-corrected rows (Combined column region): 0.634 ± 0.009; 73.0% ± 0.8%; 72.2% ± 0.7%.

Pearson (r) and Spearman ($r_s$) correlation coefficients between predicted breast age and chronological age are also reported. Bold indicates that our ensembled Mammo-AGE model achieves the best performance compared to baseline models and individual backbone-based Mammo-AGE models.

MAE mean absolute error (lower is better), CS cumulative score based on a threshold of error within 5 years (α = 5 y, higher is better).

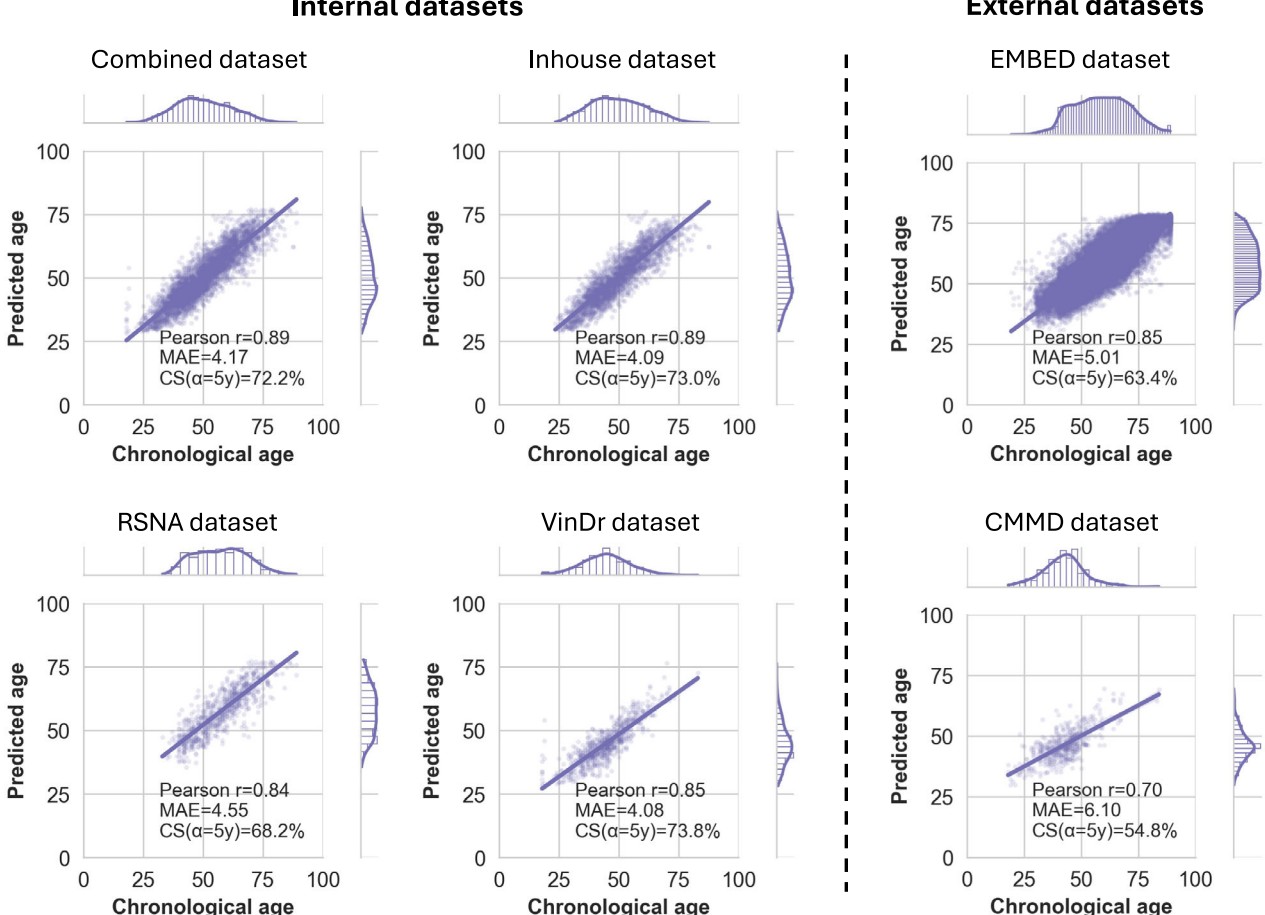

**Fig. 2 | The scatterplots of predicted breast age and chronological age on Internal and external datasets.** The scatterplots demonstrate the high correlation relationship between the predicted breast age and chronological age, with density plots on the margins to show the distribution of predictions and actual ages. Pearson ($r$) correlation coefficients between predicted breast age and chronological age are also reported; MAE: mean absolute error; CS ($\alpha$ = 5 y): cumulative score based on a threshold of error within five years. The high Pearson correlation coefficients and relatively low MAE values across both internal and external datasets indicate the robustness and generalizability of the Mammo-AGE model in predicting breast age accurately.

± 0.010, respectively (Table 2). Our Mammo-AGE model also accurately predicts the chronological age on the external datasets, with MAE = 5.010 ± 0.040 on the EMBED dataset and MAE = 6.103 ± 0.222 on the CMMD (benign group) dataset. Model performance assessed on cross-validation and external-validation datasets demonstrated consistent accuracy. Additionally, Supplementary Figs. 3 and 4 show that our method has robust generalizability performance based on different scanners and race groups across both inhouse and external EMBED datasets.

Chronological age is a well-established risk factor for future breast cancer. In addition to accurately predicting breast age, the ability of risk stratification by predicted breast age is further validated. Using data from the inhouse and EMBED datasets, we applied Kaplan-Meier survival analysis and Cox proportional hazards regression models to assess breast cancer risk. The inhouse dataset included a follow-up period of 10 years, and the external EMBED dataset comprised a follow-up period of 5 years. The results from the Kaplan-Meier curves (Supplementary Fig. 5) and the Cox proportional hazards regression models (Supplementary Table 3) demonstrate that the risk stratification capability of predicted breast age is comparable to that of chronological age. The hazard ratios per year of age are similar in both datasets, with the Inhouse dataset showing HRs of 1.032 for chronological age and 1.030 for predicted breast age, and the EMBED dataset showing HRs of 1.030 for chronological age and 1.038 for predicted breast age. These results show that predicted breast age, similar to chronological age, effectively stratifies breast cancer risk, demonstrating its potential utility in clinical risk assessment.

### Comparison with baseline models and State-of-the-Art (SOTA) methods

Our ensemble Mammo-AGE incorporates various advanced convolutional neural network (CNN) backbones (e.g., EfficientNet[35], ConvNeXt[36], DenseNet[37], and ResNet[34]). The detailed results for each CNN backbone-based Mammo-AGE are presented in Table 2. To evaluate the effectiveness of our proposed framework, we compared each backbone-based Mammo-AGE model with its corresponding CNN counterpart, including ResNet-18[34], ResNet-50[34], EfficientNet-b0[35], ConvNeXt-tiny[36], and DenseNet-121[37]. Furthermore, we extended the comparison to include two transformer-based variants: Vision Transformer (ViT)[38] and Swin Transformer (Swin-ViT)[39]. All these experiments were conducted using official implementations with pretrained weights. These results show that our proposed architecture consistently outperforms both standard CNNs and ViTs.

Furthermore, we compared our Mammo-AGE model with several other methods, including single-view (SV)[40], multi-view (MV) based late-fusion methods[41], as well as other state-of-the-art methods in natural or medical image fields, e.g., probabilistic ordinal embedding (POE)[32], mean-variance loss (MVL)[33], and global-local transformer (GLT)[17] methods. As illustrated in Fig. 3A, our proposed Mammo-AGE

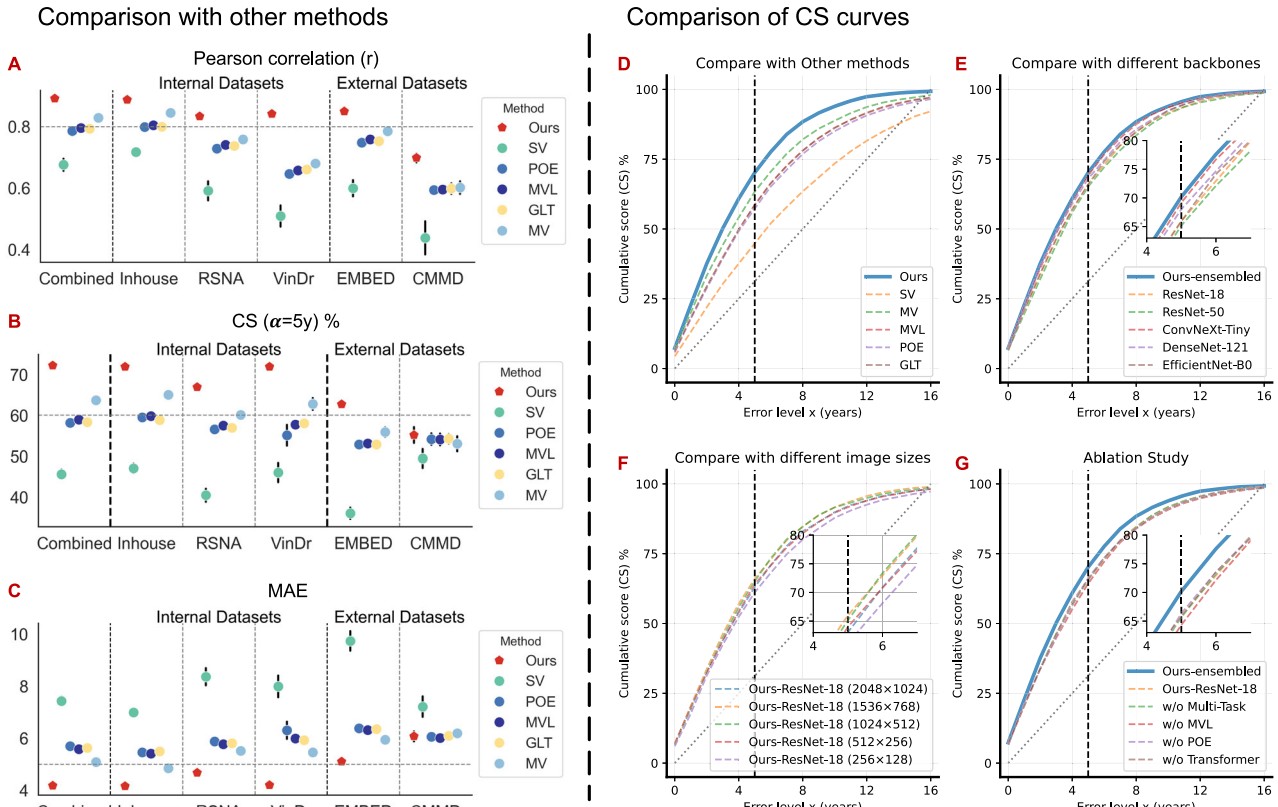

**Fig. 3 | Comparison experiments and ablation studies. A–C** Comparison of Pearson correlation ($r$), CS ($\alpha = 5$ y), and MAE performance for breast age estimation on internal and external datasets, including combined cohort ($n = 3946$), Inhouse ($n = 2489$), RSNA ($n = 750$), VinDr ($n = 707$), EMBED ($n = 33,040$), and CMMD ($n = 465$). Results are averaged over 5-fold cross-validation; error bars represent the standard deviation across folds. The figure illustrates that the ensembled Mammo-AGE model consistently outperforms all other methods across both internal and external datasets, as evidenced by lower MAE, higher Pearson correlation coefficients, and higher cumulative scores within 5 years. Ours ensembled Mammo-AGE model, SV single-view-based method, POE probabilistic ordinal embedding method, MVL mean-variance loss method, GLT global-local transformer method, MV multi-view-based method. **D–G** Comparison of CS curves for different error

levels based on the combined dataset ($n = 3946$). **D** presents the CS curves of different methods for different error levels. **E** displays the CS curves for our method using different backbones, including ResNet-18, ResNet-50, ConvNeXt-Tiny, DenseNet-121, and EfficientNet-B0, as well as our ensembled model. We note that the weighted-ensemble Mammo-AGE model achieved the best performance. Therefore, we selected this weighted ensemble model as our final model. **F** shows CS curves of our method with different input sizes of mammograms. For a fair comparison, all methods utilize the same backbone (ResNet-18). **G** The top right illustrates the CS curves of the ablation study, which investigates the contributions of the proposed instance-bag transformer module, additional loss functions, and multi-task learning employed in the model.

model surpasses these methods across all metrics. Supplementary Table 4 details the performance across different datasets, including three used for cross-validation and two used for external validation. Our ensembled Mammo-AGE model significantly outperforms other methods, with a reduction in MAE by 0.9–3.3 points ($P < 0.05$; two-sample Student's $t$-test, two-sided), achieving a MAE of $4.174 \pm 0.028$ on the combined dataset compared to the worst (Image-level MAE = $7.429 \pm 0.156$) and the best (Case-level MAE = $5.079 \pm 0.035$) alternative methods. Our model consistently achieves the lowest MAEs, highest CS ($\alpha = 5$ y), and highest $r$ values across all datasets involved in cross-validation and external validation. Figure 3D shows CS curves of different methods for different error levels. Based on the external validation, our proposed method is generalizable to different datasets from different sites and scanners.

Besides, we also provided an additional comparison (as shown in Supplementary Fig. 11) with the previous SOTA risk model[30], which aims to predict breast cancer risk, with age prediction as an auxiliary task in coarse bins (six age groups: 40-100 years old). Mammo-AGE consistently outperformed the SOTA risk model, achieving a lower MAE and also higher correlation and accuracy, demonstrating its superior ability to capture biologically relevant breast aging features.

## Ablation analysis for the Mammo-AGE model

Several ablation studies were conducted to optimize the model design. The performance of the Mammo-AGE model based on the different backbones for each fold is summarized in Supplementary Table 1. The weighted-ensemble Mammo-AGE model, which weighted averages the outputs of the different backbones (ResNet-18, ResNet-50, ConvNeXt-Tiny, DenseNet-121, and EfficientNet-B0), achieved the best performance.

Supplementary Table 2 presents the results of the ablation study on the combined dataset for five-fold cross-validation, investigating the impact of the proposed instance-bag transformer module, additional loss functions, and multi-task learning that were introduced to benefit the breast age estimation model. For fair comparison and computational efficiency, during the ablation studies, all methods comparing different modules or image sizes utilized the same backbone (ResNet-18). Our proposed approach, the ResNet18-based Mammo-AGE model, showed superior performance with MAE = $4.736 \pm 0.076$, CS ($\alpha = 5$ y) = $66.5\% \pm 0.7\%$, and $r = 0.855 \pm 0.004$ (Supplementary Table 2). Figure 3F shows CS curves in different error levels. Additionally, we note that when integrating predicted breast density into the embedding layer, the accuracy of the model could be further improved (Supplementary Table 5). We also provided the results of the breast density prediction of our Mamm-AGE model (Supplementary Table 6).

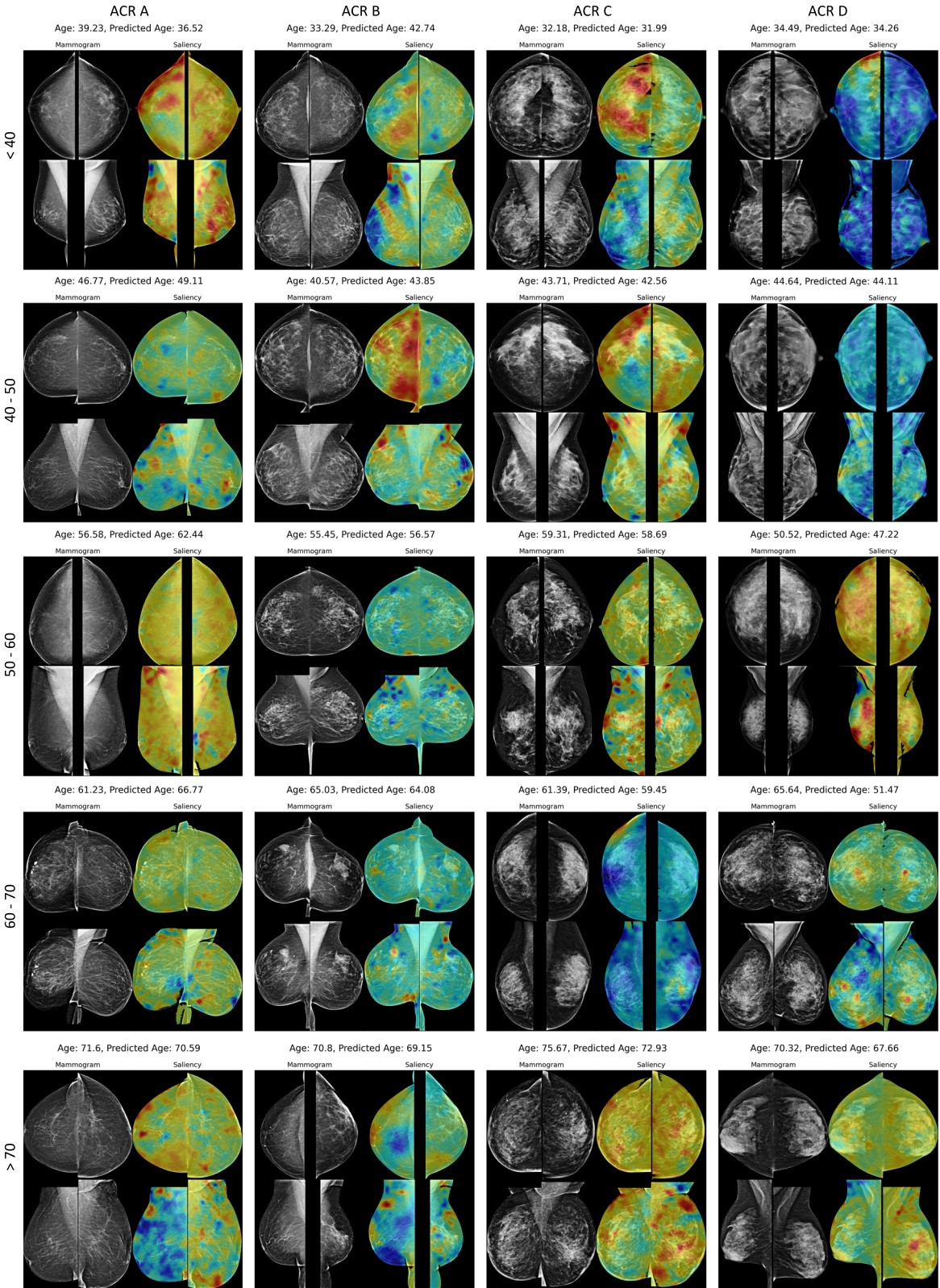

**Fig. 4 | Saliency map of the breast age prediction model.** 4-view (bilateral CC and MLO) mammograms from the inhouse dataset are shown. Saliency maps highlight regions that the model focuses on when predicting breast age. Age-specific saliency maps were computed by categorizing the data into five sub-age groups, delineated by 10-year intervals, ranging from ≤40 to ≥70 years, and into four density groups (**A**–**D**) based on ACR density classification for mammograms. Within each sub-group, different-sized masks (32 × 32, 64 × 64, 128 × 128, and 256 × 256) were used for the analysis. Normalization was carried out by dividing the entire image by the maximum value, ensuring that the values of the final saliency map ranged from 0 to 1. The minimum value within this range corresponds to blue color on the colorimetric map, while the maximum value corresponds to red. The saliency maps reveal that the model commonly focuses on features such as breast skin thickness, fibroglandular tissue, calcifications, masses, and breast vessels. The MLO views generally provide more informative features related to aging compared to the CC views.

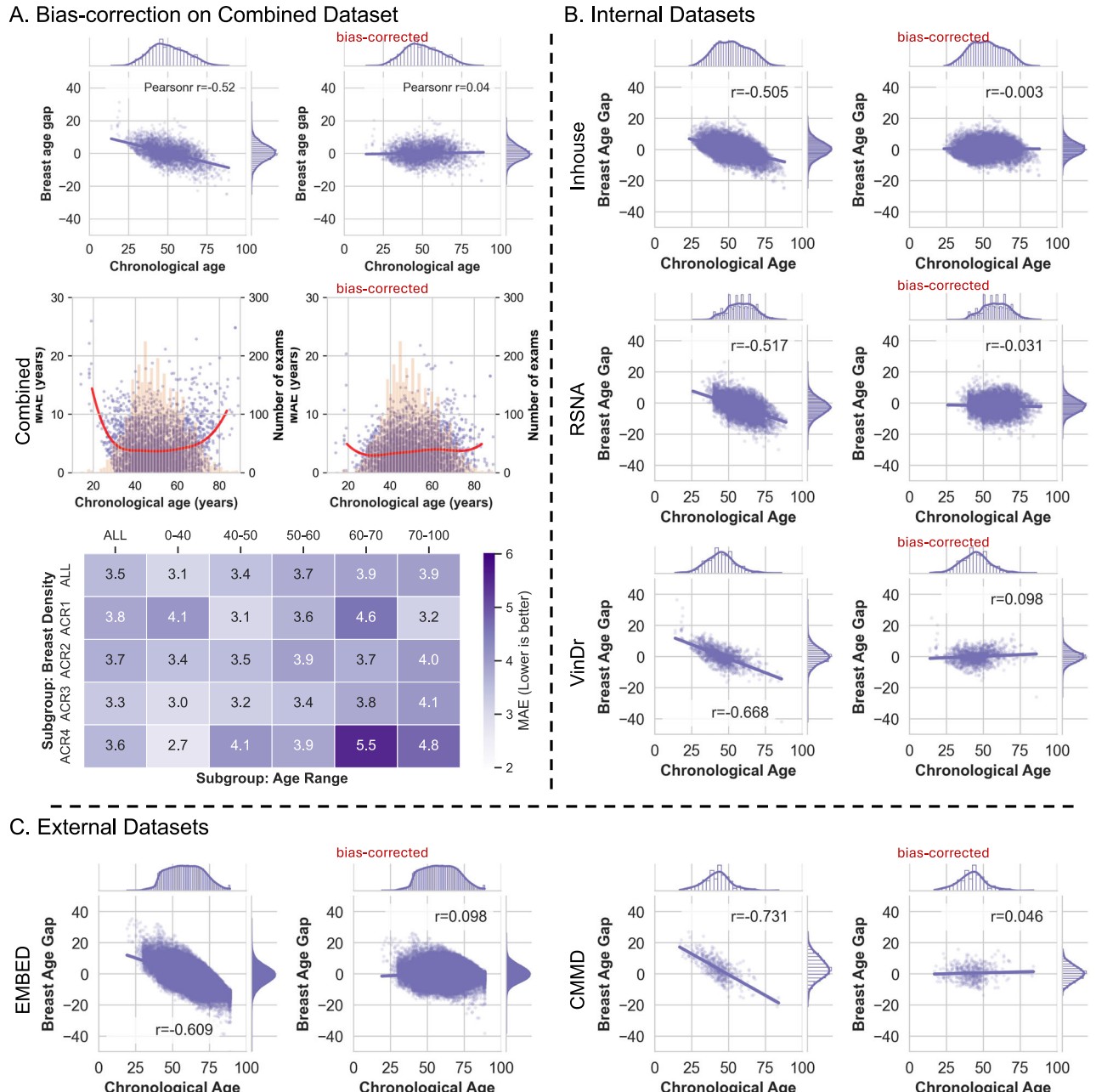

**Fig. 5 | Bias correction on each dataset. A** Results of bias correction on the combined test dataset. The scatter plots depict the relationship between the breast age gap (estimated age−chronological age) and chronological age. Before bias correction, the breast age gap against chronological age showed a notable correlation (Pearson correlation coefficient, $r = -0.52$). After bias correction, there are no obvious correlations between the corrected breast age gap and chronological age (Pearson correlation coefficient, $r = 0.04$). The MAE curves demonstrate that errors are higher in younger and older age groups before bias correction. Post-correction, the MAE curve shows improved stability and age independence. The heatmap shows the MAE results (after bias correction) across subgroups categorized by different age ranges and breast densities. **B** Results of bias correction on the separate internal datasets. **C** Results of bias correction on the separate external datasets.

Furthermore, we investigated the performance of the Mammo-AGE model (using ResNet-18 as the backbone) with different input sizes of mammograms (as training with full-resolution mammograms is very demanding on the computational resources). We tested image sizes of $2048 \times 1024$, $1536 \times 768$, $1024 \times 512$, $512 \times 256$, and $256 \times 128$. The comparative analysis revealed that the Mammo-AGE using $1536 \times 768$ achieved the optimal performance (Supplementary Table 2).

**Saliency map of breast age prediction model**
For model interpretability, we estimated saliency maps through occlusion sensitivity analysis[19]. Specifically, we utilized different-

sized masks ($32 \times 32$, $64 \times 64$, $128 \times 128$, and $256 \times 256$) to occlude parts of the breast areas in the input images by setting the corresponding areas to black images. Age-specific saliency maps were computed by categorizing the data into five sub-age groups, delineated by 10-year intervals, ranging from ≤40 to ≥70 years, and into four density groups (A-D) based on ACR density classification for mammograms. Results in Fig. 4 show that the MLO views provide more aging-related informative features compared to the CC views. Our proposed Mammo-AGE model commonly focuses on breast skin thickness, fibroglandular breast tissue, calcifications, masses, and breast vessels. Importantly, as shown in Supplementary Fig. 9, the

## A. Internal Datasets – Breast age gap differences between healthy and breast cancer groups

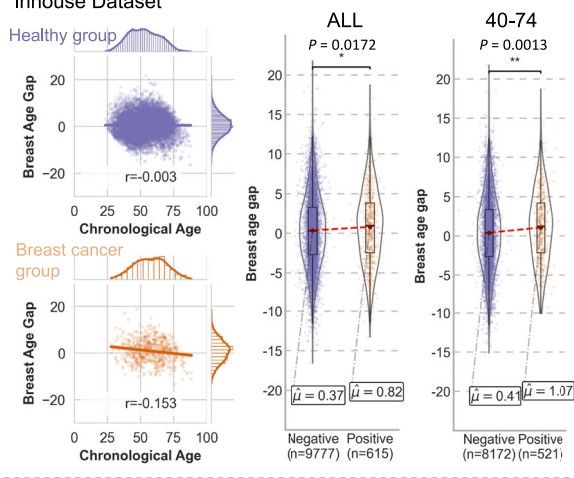

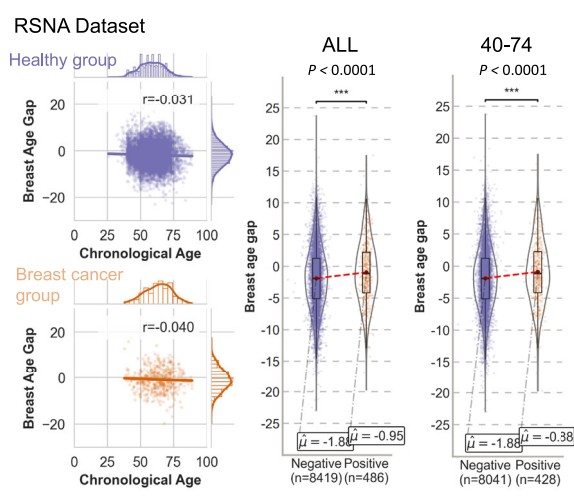

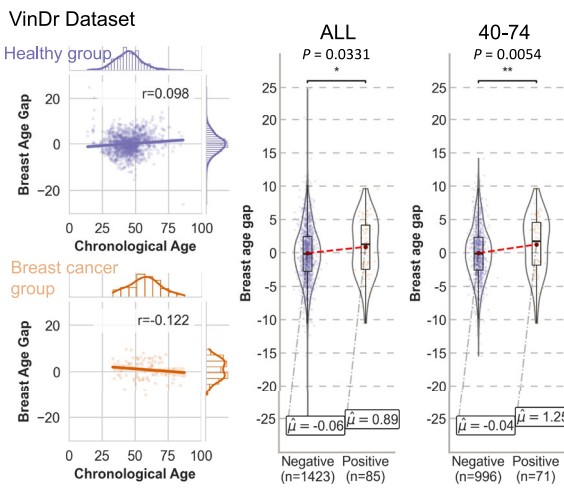

## B. External Datasets - Breast age gap differences between healthy and breast cancer groups

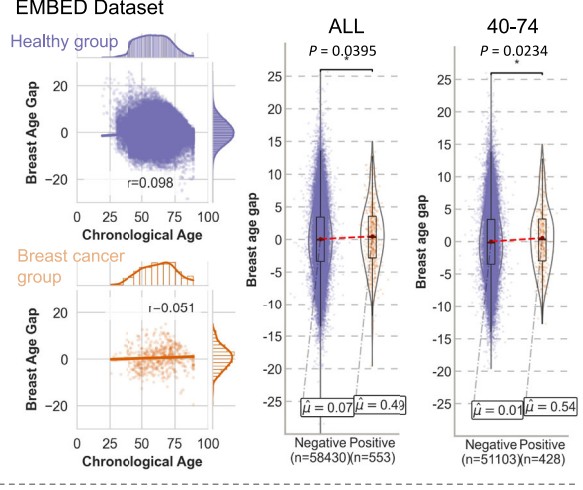

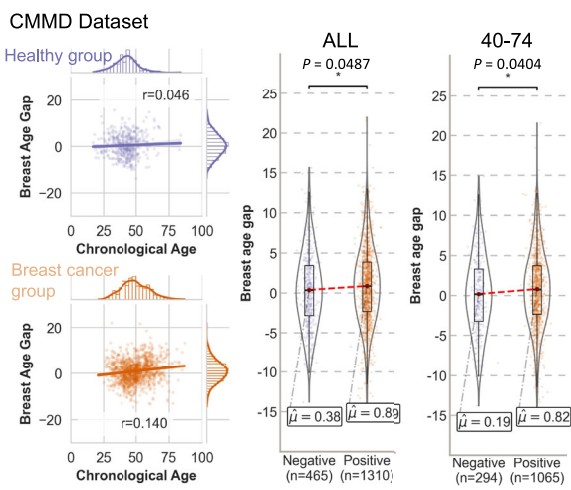

## C. Risk stratification of breast age gap

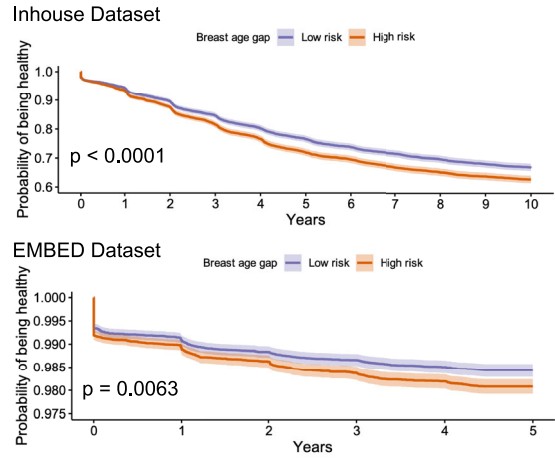

model could learn the consistent inherent aging pattern features for each woman over time. To accurately model natural biological breast tissue aging changes, we trained the model only on the healthy population. This means that unseen abnormal tissue may lead to higher deviations in age estimation for the model. We further analyzed saliency maps from time-point mammograms of patients who later developed breast cancer. We observed high-deviation areas (blue color) in focal fibroglandular regions and structural asymmetry

between views, suggesting an abnormal aging pattern associated with breast cancer risk.

### Bias correction for accurate breast age gap definition

The breast age gap is defined as the difference between the predicted breast age by the deep learning model and chronological age (predicted breast age−chronological age). The breast age gap, when positive, indicates "older" appearing breasts, while a negative breast

**Fig. 6 | Analysis of the association between breast age gap and breast cancer.** **A**, **B** show differences between healthy and breast cancer groups for internal and external datasets based on our ensembled Mammo-AGE model. We compare the breast age gap between healthy individuals and those diagnosed with breast cancer across various internal (**A**) and external (**B**) datasets. Scatter plots depict the relationship between the breast age gap and chronological age for each group. The scatter plots show that after age bias correction, the breast age gap is independent of chronological age in both healthy and cancer groups. The results demonstrate significant differences in the breast age gap between healthy and breast cancer groups across all ages and the 40–74 age groups in these datasets. Two-sided

ANCOVA tests are utilized, adjusting for the chronological age and breast density (Statistical significance: ns not significant; *: $P < 0.05$; **: $P < 0.01$; ***: $P < 0.001$). Box plots show the median (center line), the 25th and 75th percentiles (box), and whiskers extending to data within 1.5 times the interquartile range. **C** Kaplan–Meier (KM) curves for inhouse ($n = 10,392$) and EMBED ($n = 58,983$) datasets. The KM curves demonstrate that individuals with a higher breast age gap have a significantly (two-sided log-rank test) lower probability of remaining healthy over time compared to those with a lower breast age gap. Shaded areas represent the 95% confidence intervals.

age gap suggests "younger" appearing breasts. To ensure a precise assessment of aging through the breast age gap, it is crucial to address the age bias inherent in the model. Age bias commonly occurs in data-driven deep learning age-predicting models due to factors such as regression dilution, model regularization, and the influence of age distribution within the training dataset[19,42].

As depicted in Fig. 5A, before bias correction, the scatter plot illustrating the breast age gap against chronological age showed a notable correlation (Pearson correlation coefficient, $r = -0.52$). This correlation indicated a tendency of the model to overestimate breast age in the younger age group and underestimate it in the older age group. The MAE curve plot confirmed a higher error in these two age groups. To address the age bias issue, we implemented a linear regression-based bias correction method[42] to minimize the correlation between the breast age gap and chronological age.

After bias correction, there is no obvious correlation between the corrected breast age gap and chronological age (Pearson correlation coefficient, $r = 0.04$). The MAE curve demonstrates improved stability and age independence in the corrected predictions. Consequently, the overall performance of the model, evaluated using five-fold cross-validation, displays enhanced results: MAE = $3.698 \pm 0.039$, CS ($\alpha = 5$ y) = $77.6\% \pm 0.4\%$, and $r = 0.929 \pm 0.002$ on the combined test set (Table 2).

Figure 5A presents a heatmap illustrating the MAE results of the breast age model on subgroups categorized by different age ranges and breast densities, with MAE values ranging from 2.7 to 5.5. The model's performance demonstrated a higher degree of agreement between internal and external datasets, indicating its generalizability. On internal datasets, the MAE ranges from 3.497 to 4.155, while on external datasets, it ranges from 4.131 to 4.568 (as shown in Table 2). The bias-corrected model results in improved accuracy, minimizing the influence of age-related bias and enhancing its applicability across diverse datasets.

## Breast age gap as a biomarker relevant to breast cancer

To investigate the difference in the breast age gap between patients developing breast cancer and the healthy groups, we estimated the breast age gap for both internal (seen) datasets (Inhouse dataset, healthy group: $n = 9777$, cancer group: $n = 615$; RSNA dataset, healthy group: $n = 8419$, cancer group: $n = 486$; VinDr dataset, healthy group: $n = 1423$, cancer group: $n = 85$;) and external (unseen) datasets (EMBED dataset, healthy group: $n = 58,430$, cancer group: $n = 553$; CMMD dataset, healthy group: $n = 465$, cancer group: $n = 1310$). We leveraged our ensemble Mammo-AGE model, which was trained on the combined training set, as the inference model. The breast age was corrected using the same coefficients obtained during the bias correction for healthy women. Figure 6 illustrates the comparison of breast age gaps between healthy and breast cancer-diagnosed groups.

For internal (seen) datasets, the breast age gap of the breast cancer-diagnosed group was significantly higher than that of the healthy women group (inhouse dataset: $P = 0.017$; RSNA dataset: $P < 0.001$; VinDr dataset: $P = 0.033$; ANCOVA, adjusted by age and breast density). The results of the unadjusted $p$ values are reported in Supplementary Table 7, confirming that the observed differences are

independent of age and breast density. Scatterplots of the breast age gap and the chronological age are provided in Fig.6. We also calculated the Pearson correlation coefficients between the breast age gap and the chronological age for both groups. Results showed that after the bias-correction by chronological age, there is no obvious correlation between the breast age gap and the chronological age for both healthy and breast cancer groups among these datasets. For external (unseen) datasets (EMBED and CMMD), the breast age gap in the breast cancer-diagnosed group was significantly higher than in the healthy women group ($P = 0.039$ and $P = 0.049$, respectively; ANCOVA, adjusted by age and breast density).

To further eliminate potential model bias towards particularly young or older groups, we further explored differences in the breast age gap between normal and breast cancer groups in women aged 40–74 years[43] (the target group for breast cancer screening in normal-risk women). Similar group differences persisted among datasets (Inhouse dataset: $P = 0.001$; RSNA dataset: $P < 0.001$; VinDr dataset: $P = 0.005$; ANCOVA, adjusted by age and breast density). Significant differences were also observed in both CMMD and EMBED datasets for the screening target population ($P = 0.023$ and $P = 0.040$, respectively; ANCOVA, adjusted by age and breast density).

To explore the robustness and generalizability of the age-related biomarkers from mammograms defined using our approach, we retrained our Mammo-AGE model (only using ResNet-18) separately using each dataset (Inhouse, RSNA, VinDr, EMBED) from scratch. The CMMD dataset was excluded from the model training because it doesn't contain healthy mammogram examinations. Then, we evaluated breast age gap differences between healthy and breast cancer groups for every dataset based on each retrained model. Results based on each specific dataset-trained Mammo-AGE model were consistent with those based on the Mammo-AGE-Combined model, indicating that even with different dataset-trained models, the breast age gap in the breast cancer group tended to be higher than in the healthy group (as shown in Supplementary Fig. 6). These results support the robustness and generalizability of the breast age gap as a biomarker for indicating breast cancer.

## Breast age gap for breast cancer risk stratification

Based on observed differences in the breast age gap between healthy and cancer groups, we followed the method of previous studies[14,15] and explored the relationship between the breast age gap and the risk of future breast cancer events using longitudinal datasets (Inhouse and EMBED). Detailed breast cancer incident information of these two is summarized in Table 3. After a follow-up of 10 years in the inhouse dataset, 3656 (35.2%) exams were diagnosed with breast cancer, while in the EMBED dataset, after a 5-year follow-up, 1029 (1.7%) exams were diagnosed with breast cancer.

Considering the observed significant difference in the breast age gap between the healthy and cancer groups, we define the women whose breast age gaps are higher than the median (inhouse dataset: 0.36; EMBED dataset: −0.01) of the dataset as a high-risk group and those with lower gaps as a low-risk group. The Kaplan–Meier (KM) curves of women in the high and low-risk groups, which are stratified by breast age gaps of Inhouse and external EMBED datasets, are shown

**Table 3 | Association between breast age gap and future breast cancer using Cox proportional hazards regression models in Inhouse and EMBED datasets**

| Breast AGE | Years, mean ± SD | N | Events | Inc. | Model 1 (Unadj.) HR (95% CI) | P value | Model 2 (CA-ACR-adj.) HR (95% CI) | P value | Model 3 (RF-adj.) HR (95% CI) | P value |
|---|---|---|---|---|---|---|---|---|---|---|
| Inhouse BC events (10Years) | 0.51 ± 4.55 | 10,392 | 3656 | 35.2 | - | - | - | - | - | - |
| Age gap, per one age (Years) | - | - | - | - | 1.013 (1.006–1.020) | <0.001 | 1.016 (1.009–1.023) | <0.001 | 1.014 (1.007–1.022) | <0.001 |
| Low-risk group (Gap <0.36) | -3.08 ± 2.70 | 5196 | 1722 | 33.1 | Reference | - | Reference | - | Reference | - |
| High-risk group (Gap > 0.36) | 4.10 ± 2.88 | 5196 | 1934 | 37.2 | 1.164 (1.091–1.242) | <0.001 | 1.157 (1.084–1.235) | <0.001 | 1.140 (1.067–1.217) | <0.001 |
| EMBED BC events (5Years) | 0.08 ± 5.025 | 58,983 | 1029 | 3.49 | - | - | - | - | - | - |
| Age gap, per one age (Years) | - | - | - | - | 1.022 (1.001–1.034) | <0.001 | 1.020 (1.008–1.033) | 0.002 | 1.020 (1.008–1.033) | 0.002 |
| Low-risk group (Gap < -0.01) | -3.75 ± 2.85 | 29,492 | 463 | 3.14 | Reference | - | Reference | - | Reference | - |
| High-risk group (Gap > -0.01) | 3.88 ± 3.06 | 29,491 | 566 | 3.84 | 1.225 (1.083–1.385) | 0.001 | 1.152 (1.071–1.305) | 0.026 | 1.151 (1.016–1.304) | 0.027 |

P values are calculated using two-sided Wald tests.

Inc incidence per 1000 person-years, CI confidence interval, BC breast cancer, HR hazard ratio, Unadj. HR (Inhouse) HR unadjusted HR, CA-ACR-adj. HR HR adjusted HR on chronological age (CA) and breast density (ACR), RF-adj. HR (Inhouse) HR adjusted HR on more risk factors for inhouse dataset, including CA, ACR, Race, Gene mutation, Menarche, Menopausal status, and Manufacturer of mammograms, RF-adj. HR (EMBED) HR adjusted HR on more risk factors for EMBED dataset, including CA, ACR, Ethnicity, and Manufacturer of mammograms, Breast AGE deep learning-based breast biological age.

in Fig. 6C. We note that in both datasets, the breast cancer incidence rate in the low-risk group is indeed significantly lower than the high-risk group (both inhouse and EMBED datasets: $P < 0.01$).

After adjusting for confounding factors such as chronological age and breast density, each 1-year increase in breast age gap was associated with 1.6% and 2.0% increase in breast cancer risk on the inhouse and EMBED datasets, respectively (Inhouse dataset: HR = 1.016 [95% CI: 1.009–1.023], $P < 0.001$; EMBED dataset: HR = 1.020 [95% CI: 1.008 - 1.033], $P = 0.002$; Table 3, Model 2). To ensure the validity of the Cox model estimates, the proportional hazards assumption was verified by visual inspection and Schoenfeld residuals tests[15,44] (as shown in Supplementary Fig. 7). Women with a breast age gap higher than the median had significantly increased 10 or 5-year breast cancer risk compared to those with a low gap (Inhouse dataset: HR = 1.157 [95% CI: 1.084–1.235], $P < 0.001$; EMBED dataset: HR = 1.152 [95% CI: 1.071–1.305], $P = 0.026$). Further adjusting for additional breast cancer risk factors (Table 3, Model 3) did not substantially alter the association between breast age gap and cancer risk, supporting its robustness and independence. These results suggest that positive breast age gaps (where the breast appears older than the chronological age) are associated with a substantially increased risk of breast cancer.

Allowing for non-linearity, Supplementary Fig. 8 presents the estimated relationship between breast age gap and breast cancer risk. Significant overall and non-linear associations were observed (Inhouse dataset: $P_{overall} < 0.001$; $P_{non\text{-}linear} = 0.024$; EMBED dataset: $P_{overall} < 0.001$, $P_{non\text{-}linear} = 0.016$).

Besides, to better inform the optimal age at which to target mammographic screening, the odds ratios and diagnostic age analysis results (Supplementary Table 9) show that extremely accelerated-aging patients were diagnosed earlier (62 years, 95% CI: 47–80), compared to extremely decelerated-aging women, who had a later average diagnostic age (72 years, 95% CI: 47–98).

To assess the clinical relevance of the breast age gap compared to current standards, we benchmarked its predictive performance against breast density, a widely accepted imaging-based risk factor and a criterion used in screening practice (e.g., Netherlands Early Breast Neoplasm Screening (DENSE) trial[45]). Under matched high-risk population sizes based on the DENSE trial criteria (i.e., to select the same number of high-risk women), Mammo-AGE consistently identified more breast cancers than breast density across 1-, 5-, and 10-year follow-up periods (Supplementary Fig. 12). Notably, Mammo-AGE identified approximately 30% more cancers than the DENSE trial criteria in the EMBED screening dataset at 1-year follow-up. Additionally, Supplementary Table 8 presents the odds ratio analysis of breast cancer based on breast density and breast age gap.

### Mammo-AGE model for downstream clinical tasks beyond biological aging

Technically, this refined learning of biological features offers a more efficient granular assessment of tissue characteristics, which may enhance the efficiency of modeling for the downstream clinical tasks. To demonstrate such modeling efficiency, we finetuned the Mammo-AGE for breast cancer risk prediction and diagnostic classification on Inhouse and EMBED datasets and compared its performance with several SOTA methods on all datasets (Fig. 7). Inhouse and EMBED datasets were randomly split into training, validation, and test sets at the patient level. To further evaluate the models' generalizability, we also included a new public CSAW-CC dataset[46] for external validation directly. Importantly, both CMMD and CSAW-CC datasets were not accessed by the model during either the age learning or downstream task learning phases.

We evaluated the Mammo-AGE model for risk prediction across three datasets and for breast cancer diagnosis across six datasets. In all tasks, Mammo-AGE outperformed both the baseline model and SOTA risk model[7] and SOTA pretrained methods[47]. Specifically, Mammo-AGE

## A. Risk prediction

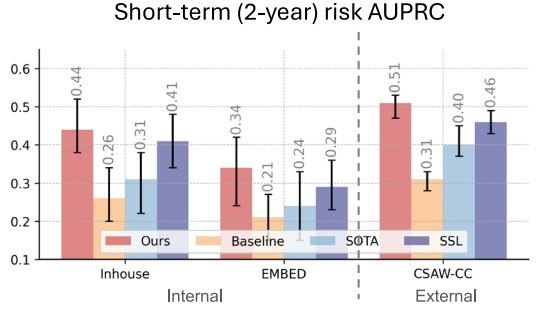

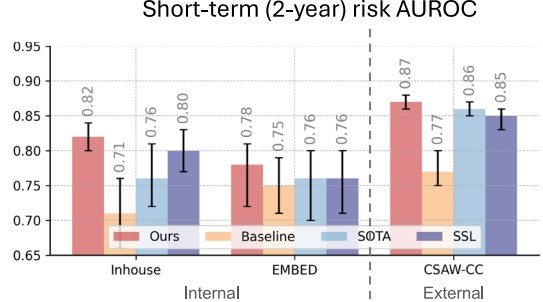

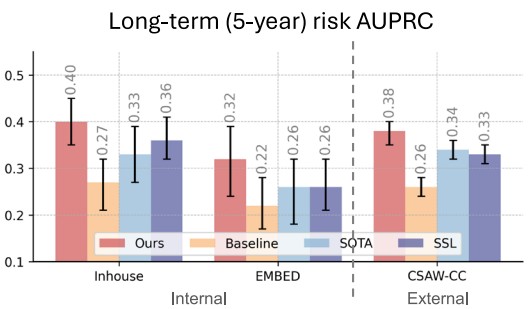

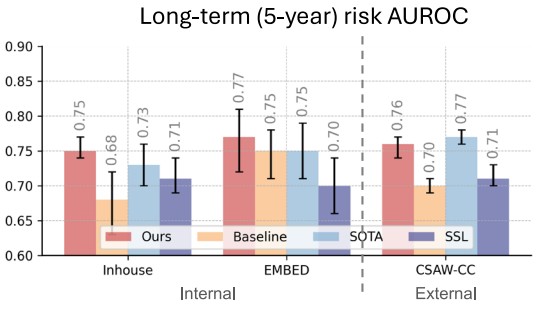

## B. Diagnosis

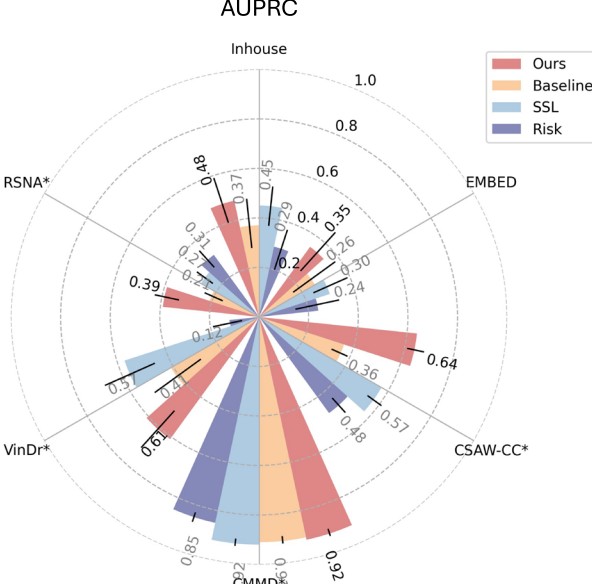

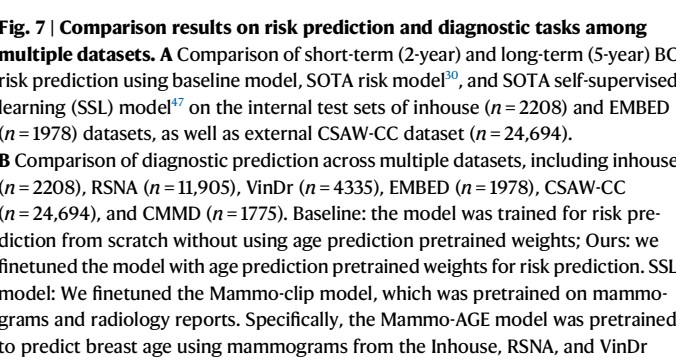

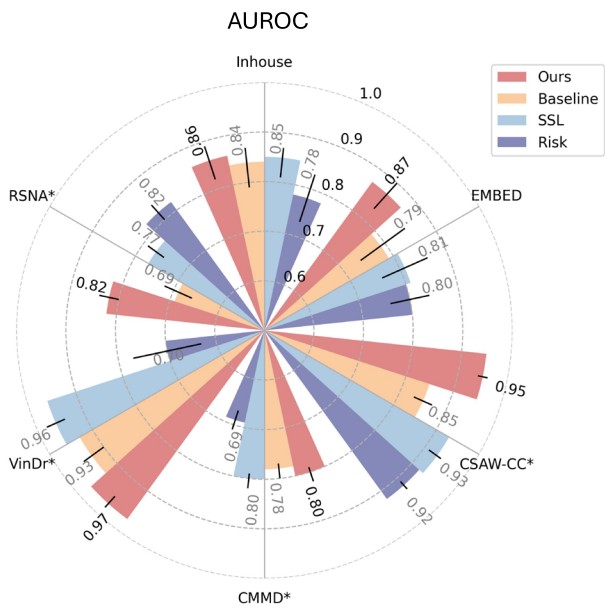

**Fig. 7 | Comparison results on risk prediction and diagnostic tasks among multiple datasets. A** Comparison of short-term (2-year) and long-term (5-year) BC risk prediction using baseline model, SOTA risk model[30], and SOTA self-supervised learning (SSL) model[47] on the internal test sets of inhouse ($n = 2208$) and EMBED ($n = 1978$) datasets, as well as external CSAW-CC dataset ($n = 24,694$).
**B** Comparison of diagnostic prediction across multiple datasets, including inhouse ($n = 2208$), RSNA ($n = 11,905$), VinDr ($n = 4335$), EMBED ($n = 1978$), CSAW-CC ($n = 24,694$), and CMMD ($n = 1775$). Baseline: the model was trained for risk prediction from scratch without using age prediction pretrained weights; Ours: we finetuned the model with age prediction pretrained weights for risk prediction. SSL model: We finetuned the Mammo-clip model, which was pretrained on mammograms and radiology reports. Specifically, the Mammo-AGE model was pretrained to predict breast age using mammograms from the Inhouse, RSNA, and VinDr

datasets. For the two downstream tasks, the models (baseline, ours, and SSL) were trained/finetuned using the Inhouse and EMBED datasets, which contain breast cancer outcomes. SOTA/Risk model: MIRAI risk prediction model. Given that MIRAI is already a well-validated model trained on large and diverse cohorts[30,71], we evaluated it without additional finetuning to preserve its benchmark integrity. To ensure a fair comparison and reflect real-world usage, the CSAW-CC (risk and diagnostic tasks) and CMMD (diagnostic task) datasets were used exclusively as external test sets. No training or finetuning was performed on CSAW-CC and CMMD for any of the models. AUROC Area Under Receiver Operating Characteristic Curve, AUPRC Area Under Precision-Recall Curve. The error bars represent the 95% confidence intervals (CI) of AUROC and AUPRC metrics, estimated from 1000 bootstrap samples for each measure. Center values represent the mean performance across bootstrap replicates.

(with age pretraining) surpassed the risk model across all metrics (AUPRC: 0.40 vs. 0.36; AUROC: 0.75 vs. 0.71 in the inhouse dataset, for long-term risk prediction). Importantly, performance improvements were consistently observed when fine-tuning with age-pretrained weights compared to training from scratch (baseline model). To further investigate this learning efficiency advantage, we fine-tuned Mammo-AGE using different fractions of training data for the breast cancer diagnosis task. As shown in Supplementary Fig. 13, Mammo-AGE consistently outperformed training from scratch (i.e., baseline), particularly under limited data conditions (e.g., using only 5% of the training set, approximately 2000 mammogram examinations).

## Discussion

In this study, we introduced a concept of breast age as obtained by the Mammo-AGE model, estimating breast age from mammograms without requiring additional clinical data. The breast age was estimated with an accuracy of about 4.2–6.1 years of MAE using various cohorts. Our Mammo-AGE successfully captured aging-related changes in mammograms, as demonstrated by saliency map analysis. We explored the clinical implications of estimated breast age, finding that patients who developed breast cancer were estimated to have a higher breast age gap of 0.42–1.29 years (had older appearing breasts on mammogram) compared to the healthy population. Importantly, separate dataset retraining experiments demonstrated the generalizability of the defined breast age gap biomarker. Breast age was independently associated with breast cancer outcomes, even after adjusting for covariates (chronological age and breast density), and exhibited an association with the risk of breast cancer. From these findings, we conclude that breast age can be accurately estimated from mammograms using our proposed model and that it serves as an indicator of breast cancer risk. Moreover, we applied our model for downstream clinical tasks, including breast cancer diagnosis and risk prediction, and found that fine-tuning Mammo-AGE with breast age pretraining consistently outperformed baseline and other comparative models across multiple datasets, demonstrating improved predictive performance and enhanced learning efficiency.

The Mammo-AGE model exhibited good performance, with MAEs ranging from 4.2 to 6.1 years, validated on various cohorts from populations around the world, surpassing existing research[40,48] in breast age prediction. Further subgroup analysis shows the robustness of our model across racial groups (Supplementary Fig. 4). Previous studies have shown MAEs of 8.11 years for mammographic[40], and 9.46 years for DNA methylation clock[48]. External validation results underscore the robustness of the model's performance on unseen datasets. Thus, the aging biomarkers identified from mammograms using our breast age estimation model are generalizable across different populations. Our study validates the hypothesis that, by leveraging the development of the deep learning technique through a data-driven approach and using a curated model design, healthy breast aging patterns can be accurately captured from mammograms in a high-dimensional latent space. However, the task is difficult for human physicians.

In breast cancer screening, age is a recognized risk factor related to breast cancer. Considering the accessibility of deep learning derived risk factors, there are already some studies that demonstrate the potential of leveraging advanced deep learning methods to learn breast cancer-related risk factors (including age) from screening mammograms, which can improve the performance of a risk prediction model[30,49]. However, these studies only aimed to estimate future breast cancer risk and didn't evaluate the performance of risk factor prediction. Unlike these risk models, which are designed for direct risk prediction, Mammo-AGE models biological breast aging as an image-based biomarker, while highlighting the broader clinical utility of breast tissue aging. Our study provides further evidence (Supplementary Table 3 and Supplementary Fig. 5) of the feasibility of learning

breast cancer-related risk factors, such as biological age, from mammograms.

To enhance comprehensibility, we generated breast maps delineating the pertinent characteristics employed in the age prediction model utilizing occlusion sensitivity analysis. Saliency map analysis (Fig. 4) indicates that Mammo-AGE predominantly focuses on biologically aging relevant breast structures, such as fibroglandular tissue, calcifications, blood vessels, skin thickness, and adipose tissue, which are known to change with age[50]. The longitudinal heatmap analysis (Supplementary Fig. 9) suggests that the model is consistently capturing biologically meaningful patterns from these informative areas. Additionally, our results (Supplementary Table 5) confirm that breast density contributes to improved age prediction accuracy. Interestingly, heatmap analysis on breast cancer patients (Supplementary Fig. 10) shows that Mammo-AGE may be sensitive to local physiological variations that precede or accompany malignant transformation.

It is well known that aging serves as a marker for the diverse biological changes that occur over the course of life and may indicate many adverse health events, including cancer and various other diseases[8,51]. Therefore, beyond estimating the breast age, we also extended the exploration to the definition of risk groups for the development of breast cancer. This study suggests that the breast age gap derived from mammograms may serve as a biomarker of abnormal aging and is associated with breast cancer development. Breast age gaps appear to be independent of the chronological age of the women. These findings are based on five large datasets gathered from around the world, including Europe (inhouse dataset), America, and Australia (RSNA and EMBED), and Asian countries (VinDr and CMMD). The breast aging-related biomarkers identified by our Mammo-AGE model demonstrate its generalizability and stability when applied to both internal and unseen external datasets (Fig. 6). Due to differences in breast density distributions across datasets, the breast density and age were carefully adjusted in all analyses. Our findings are further strengthened by the cross-dataset retraining experiment (Supplementary Fig. 6). The comparative analysis highlights the model's adaptability to diverse datasets and its ability to consistently capture breast aging patterns.

Our findings align with previous DNA methylation-based research[4,52–54]. For example, Jacob et al. utilized previously established methylation-based "clocks" (Hannum, Horvath, and Levine) to investigate biological age, revealing that women with estimates surpassing their chronological age faced an elevated risk of developing breast cancer. Similarly, James et al.[52] developed a breast tissue-specific model for estimating DNA methylation (DNAm) age, noting a significantly higher epigenetic age acceleration in breast tumor tissue compared to normal or adjacent-normal tissue. Existing studies on breast aging primarily rely on DNAm-based clocks[4,55–62], which estimate biological age using epigenetic markers but require tissue or blood samples. These methods are limited in large-scale screening applications. In contrast, our approach provides a non-invasive practical imaging-based alternative for estimating breast age and assessing its relationship with breast health. Our routinely used imaging-based breast age gap demonstrates similar effect sizes performance to those of previously published clinical biomarkers based on DNAm clock studies[4] for breast cancer risk stratification. Notably, the age of breast tissue appears to be influenced by various factors operating in tandem throughout an individual's lifetime, including the age of menarche, hormone replacement therapy use, alcohol consumption, and others[54]. The relation between the breast age gap and breast cancer risk is not linear (Supplementary Fig. 8). Our findings confirm the hypothesis that individuals with extreme deviations in breast tissue aging are at particularly elevated risk, echoing the disproportionate risk accumulation in the "extreme agers"[10].

Our study may have important clinical implications and clinical impact. Mammogram is a fast, non-invasive, and cost-effective

technique for breast imaging, and this nature enables it to be an accessible screening tool to detect tumors early and identify individuals at an increased risk of breast cancer. Our study supports the idea that mammograms provide insights into future breast cancer risk, and our study enhances the interpretability of these findings. This may help to further expand the potential of mammograms beyond its conventional role in early detection. Specifically, it identifies women whose breasts undergo accelerated aging as high risk, facilitating the selection of both low-risk and high-risk patients in screening programs. As evidenced by Supplementary Table 9, our findings suggest that breast age gap, as an imaging-based biomarker, has the potential to refine mammographic screening strategies. Specifically, women with a higher breast age gap may benefit from additional screening, while those with a lower gap might be candidates for less frequent screening. However, further research is needed to assess whether breast aging patterns remain stable over time using longitudinal mammogram-based model, which could support the development of personalized screening schedules.

Furthermore, when applying the Mammo-AGE model for downstream clinical tasks, e.g., diagnosis and risk prediction, our Mammo-AGE model outperformed the SOTA methods across multiple (internal and external) datasets (Fig. 7). The results of downstream tasks demonstrate that Mammo-AGE achieves superior age prediction accuracy while also providing meaningful information for breast cancer risk assessment and diagnosis. More importantly, our findings demonstrate that breast age estimation improves downstream tasks' learning efficiency and generalizability in mammography-based risk deep learning models. Fine-tuning on breast age features could enhance cancer classification even under limited data settings, highlighting its potential as an imaging-derived indicator for breast cancer detection.

This study had several limitations. Our proposed data-driven model extracted the aging-related data from a single-time point mammogram examination. Learning breast tissue changes through multi-time point mammograms may, however, capture a more accurate pattern of individual aging. Some lifestyle and clinical factors, such as BMI, smoking status, and comorbidities, were not available due to data privacy constraints and limited availability. Future studies could explore their impact on breast tissue aging. With the growing development of DNAm-based organ-specific aging clocks, future work should explore the comparison and integration of different predictive aging biomarkers to enhance clinical applications. Besides, how breast aging is associated with BC-related risk factors still needs to be explored. How to leverage breast aging as a biomarker, incorporate it in existing breast cancer risk models, and further improve breast cancer risk estimation and facilitate individualized breast cancer screening policy is still an open question. Beyond breast cancer, the relationship between the breast age gap and other breast diseases warrants further investigation.

In conclusion, we developed a deep learning solution that accurately estimates women's age from mammograms without any additional information. The estimated breast age (mammogram-based) may serve as an indicator of breast aging and breast cancer risk, presenting a potential tool for clinicians in predicting and early detection of breast cancer in the digital medicine era.

## Methods
Our retrospective study was conducted in accordance with all relevant ethical regulations and was approved by the Institutional Review Board (IRB) of the hospital with IRB number IRBd21-060, approved in 2021. All inhouse data were de-identified prior to model development. The IRB waived the requirement for individual informed consent due to the retrospective nature of the study. In addition, publicly available mammography datasets were used in accordance with their data usage policies and relevant ethical guidelines; these datasets were de-identified by their providers prior to release.

### Datasets collection
This study conducted experiments on five datasets from various origins and locations, as summarized in Table 1. These datasets include one large inhouse longitudinal dataset and four publicly available datasets: RSNA dataset[23,63], VinDr-Mammo dataset[26], EMory BrEast Imaging (EMBED) dataset[24], Chinese Mammography Database (CMMD) dataset[64]. Besides, we also collected the CSAW-CC[43] dataset for the external validation during the downstream risk prediction and breast cancer diagnostic tasks.

**Inhouse dataset.** For the retrospective study, we collected screening mammograms from the Netherlands Cancer Institute between January 1, 2004, and December 31, 2020. For each individual, all screening mammograms, also repeated over time, were included. In a total of 5048 women (aged from 19 to 98), 18,828 consecutive mammogram examinations with BI-RADS 1 and 2 scores were collected (as shown in Supplementary Fig. 2, the flowchart). Breast cancer diagnoses were collected through pathology follow-up results. Of these, 615 patients with breast cancer were diagnosed within one year after mammogram examination.

**RSNA dataset[23,63].** We collected this dataset from a recent Kaggle competition, "RSNA Screening Mammography Breast Cancer Detection AI Challenge (2023)". It comprises approximately 11,000 individual women aged 19–89. Each exam includes at least four images from different laterality (left and right) and views (CC and MLO), with additional features such as age, BIRADS, and density available for potential classification purposes.

**VinDr-Mammo dataset[26].** A benchmark dataset from VinDr Hospital in Vietnam for computer-aided diagnosis in full-field digital mammography (FFDM). All mammograms are scored in the BI-RADS range 1–5. After the exclusion of the mammograms without the age information, we collected a total of 4335 exams and including 85 exams as breast cancer with BI-RADS 5. The dataset also provides breast density information.

**EMBED dataset[24].** This large-scale longitudinal screening dataset contains 364,000 mammographic exams for 110,000 patients over an 8-year period. The EMBED AWS Open Data release represents 20% of the dataset. In this study, we collected a total of 21,434 women (aged 19–89), 58,983 consecutive mammogram examinations, including 553 exams diagnosed with breast cancer within one year.

**CMMD dataset[64].** The Chinese Mammography Database is a dataset of FFDM images from Chinese women. It contains 1775 studies from several Chinese institutions. All cases have breast-level benign and malignant confirmed by biopsy, including 1296 breast cancer patients. The dataset didn't provide the breast density label.

**CSAW-CC dataset[43].** The dataset contains mammography images from breast cancer screenings for women (age range: 40–74 years) at Karolinska University Hospital (Stockholm, Sweden) collected between 2008 and 2015. It includes 7850 healthy controls and 873 breast cancer cases which diagnostic during the time period. Due to the lack of a precise age label, we excluded the dataset for the age prediction and related breast age gap analysis. We only leverage the CSAW-CC dataset for the external validation during fine-tuning the Mammo-AGE model on the downstream risk prediction and breast cancer diagnostic tasks.

For individuals undergoing natural aging, the chronological age is typically close to the biological age. However, in patients, the difference between these two ages may reflect the phenomenon of disease-related aging[15,19]. In order to comprehend the normal aging pattern of the breast, we trained breast age models based on the healthy population. A combined dataset was created by selecting the normal breast mammogram examinations scored with BIRADS 1 and 2 from Inhouse, RSNA, and VinDr datasets. It includes a total of 35,068 examinations, with 140,272 images. Race information was obtained from electronic medical records for the Inhouse and EMBED datasets, assigned as Asian for the VinDr and CMMD datasets collected in Asian institutions, and was not available for the RSNA dataset.

Considering that the breast age model is trained on a large-scale population, five-fold cross-validation was adopted to balance computational cost (approximately 180 GPU hours per fold of an ensemble Mammo-AGE model) and estimation stability, following common practice in large-scale imaging studies[11,19]. Then, this combined dataset is randomly divided into a five-fold cross-validation dataset and an internal test dataset with a ratio of 8:2 at the patient level. Subsequently, the five-fold cross-validation dataset is randomly split into five folds (patient-wise) to build the five breast-age models. Finally, we validate the performance of the models on the internal test set and external datasets.

## Architecture of Mammo-AGE model

The overview framework of our Breast-Age model is illustrated in Fig. 1A. Our proposed Breast-Age model is a multi-view-based method, which utilizes four views of mammograms as input (right craniocaudal (CC), right mediolateral oblique (MLO), left CC, and left MLO views), considering bilateral breasts. The baseline convolutional neural network, such as ResNet, DenseNet, etc., is utilized as the shared-weights backbone to extract deep features from each input image. The specific dimension of input data was $1536 \times 768$ in our study. Experimental tests measuring how different backbones and the input image size affected the model's results were performed separately (see the ablation experiments).

The features before the last pooling and full connection of the backbone are fed into the instance-bag transformer block. The proposed instance-bag transformer is inspired by the previous Global-Local transformer[17] to combine the information of the examination for more accurately predicting breast age. The motivation of the proposed transformer module is to learn fine-grained information from a single view by fusing the global context visual information learned from the whole mammogram examination (four views of the bilateral breast). We define the features of each view as local features and fuse these four local features together to form global features. As shown in Fig. 1A, this transformer module includes self-attention and cross-attention blocks, which are similar to previous studies[65]. Both self-attention and cross-attention blocks are based on multi-head attention, where the global and local features are split into $h = 8$ parallel parts on the channel dimension. For self-attention, the query, key, and value vectors are from the same features to learn the fine-grained local information. For the cross-attention, we replace the key and value vectors with those from the global features to fuse the global context information. Then, local features and the self-attention and cross-attention generated features are concatenated in the channel dimension and fed into the feed-forward layers to fuse the information and maintain the same size as the local features. The transformer block is repeated $N = 1$ times to iteratively integrate information.

Following the transformer model, the features on each branch are channeled into the final layer for predicting the biological age of the breast. Different from the direct regression methods[11,12,14,16,25], which simply map the feature vector to the age through a fully connected layer, we utilized a Probabilistic Ordinal Embeddings (POE)[32] learning strategy in the final layer. This is an uncertainty-aware regression

method, which represents each data as a multivariate Gaussian distribution rather than a deterministic point in the latent space and learns a distribution following the ordinal constraint in the embedding space. For instance, it tries to define each mammogram as an age distribution with an uncertain range in the embedding space rather than an accurate point. Like a human, who will have different confidence in their judgments with different judgment bases. A mammogram with less content information or more noise will lead to larger uncertainty. Besides, the ordinal relations of different-aged women's mammograms are also considered in the embedding space. For instance, consider three mammograms from women aged 30, 50, and 70. In the embedding space, the ordinal relations of these samples are constrained (e.g., $30 < 50 < 70$). This resulting ordinal distribution constraint in the latent space contributes to ordered probabilistic prediction. Thus, incorporating such embedding learning can yield better predictive performance in the age-estimation task. This module includes several loss functions (referred to in ref.[32]); in this study, we collectively refer to them as a composite loss function of POE.

Similar to previous research[32,33], we treated the age estimation task as a classification task with 100 classes corresponding to ages 1–100. We also utilized mean-variance (MV) loss[33] in this study. Specifically, the MV loss consists of a mean loss, which penalizes the difference between the mean of the estimated age distribution and the ground-truth age, and a variance loss, which penalizes the variance of the estimated age distribution to ensure a concentrated distribution. Besides, considering the correlation between breast density and chronological age[31], we also constrained the model to learn the breast density of the mammograms through multi-task learning. The combined features are leveraged to feed into the density classification layer, and correspondingly, the cross-entropy (CE) was used to calculate the density classification loss.

## Implementation details of model development

In this study, we leveraged a variety of well-established neural network architectures, ResNet-18[34], ResNet-50[34], EfficientNet-b0[35], Convnext-tiny[36], and DenseNet-121[37], initializing them with ImageNet pre-trained weights as the backbone of all our methods. The weight ratios were derived using five-fold cross-validation by first calculating the weight ratios for each backbone model based on its performance (MAE) in each validation fold. These fold-specific weight ratios were then averaged across all five folds to obtain the final set of weight ratios. This ensures a balanced contribution from different architectures while maintaining diversity, reducing the risk of overfitting.

All models are implemented in PyTorch (version 1.12.1). The training strategies were standardized across all methods. Specifically, we opted for the Adam[66] optimizer with a rate of 0.5 for dropout[67] after every fully connected layer. The models underwent training for 100 epochs with an initial learning rate of $10^{-4}$, which is decayed by a factor of 10 every 20 epochs. Early stopping was applied when the validation loss did not decrease for 10 consecutive epochs. The experiments are performed on a Quadro A6000 GPU (48GB). We set the batch size of 32 when training the ResNet-18-based method and the batch size of 6 for other backbone-based models due to the limitation of the GPU memory. The selection of the best models for each method was based on the best MAE performance index on the validation set.

The source code is available at https://github.com/Netherlands-Cancer-Institute/Mammo-AGE. Mammo-AGE was developed based on standard DICOM mammograms. First, we converted the images into 16-bit PNG format and segmented the whole breast region to exclude the background[65]. For unifying the size of all images, we simultaneously perform resizing and padding to standardize the image size to $1536 \times 768$ pixels, which can preserve the original aspect ratio of the mammogram image and avoid structural distortion. We also employ data augmentation techniques (i.e., random flip, brightness, contrast, Gaussian blur, Gaussian noise, and coarse dropout) during training for

model robustness and overfitting prevention. All augmentations were executed using the Albumentations Python library (https://github.com/albumentations-team/albumentations). Besides, during model inference, it is hardware-friendly and runs efficiently (-4 s per exam on a 2080 Ti (12GB) GPU). It can be easily deployed on clinical servers or cloud-based systems, supporting integration into existing radiology workflows.

## Ablation analysis

To optimize our model design and assess the effectiveness of each proposed component in enhancing breast age estimation performance, we conducted a series of ablation experiments. In the first set of analyses, we explored the model's performance with various backbones, such as ResNet-18, ResNet-50, ConvNeXt-Tiny, DenseNet-121, EfficientNet-B0, and an ensemble model that combined all the predictions of these five models. The ensemble model's predictions were selected for further analysis and consideration as our final results in this study. Additionally, we evaluated the model's performance by removing specific components, including the proposed transformer block (w/o CI-Transformer), POE loss (w/o POE), MV loss (w/o MVL), and the breast density learning (w/o Multi-task). These experiments were implemented using ResNet-18 as the backbone. Furthermore, we investigated the model's performance with different sizes of mammograms, specifically $256 \times 128$, $512 \times 256$, $1024 \times 512$, $1536 \times 768$, and $2048 \times 1024$ pixels. All results are based on the five-fold cross-validation.

## Bias correction

The presence of systematic bias, induced by regression dilution, introduces a correlation between the estimated age gap and chronological age[19,68]. This leads to a tendency to overestimate age in younger individuals and underestimate it in older individuals[19,68,69]. To mitigate this phenomenon, we applied the linear bias correction method as outlined by Smith et al. [42].

The systematic bias is defined as $\Delta_{bias} = \hat{y} - x$, where $\hat{y}$ denotes the predicted breast age and $x$ is the chronological age. To correct for bias, we fit a linear regression model: $\hat{\Delta}_{bias} = ax + b$, using the healthy reference population to estimate systematic bias $\Delta_{bias}$. Here, a and b are the represent the slope and intercept derived from the healthy group. The bias-corrected estimated breast age is then computed as: $\hat{y}_{corrected} = \hat{y} - \hat{\Delta}_{bias} = \hat{y} - (ax + b)$, and the corrected breast age gap is: $\Delta_{breastage, corrected} = \hat{y}_{corrected} - x$.

Following the standard aging study method[19], bias correction coefficients were obtained from the validation set (healthy population) to ensure that the breast age gap reflects true biological aging rather than systematic prediction bias, remaining independent of health status. These coefficients, derived from the healthy group, are then applied consistently across breast cancer diagnostic groups for the purpose of bias correction. This linear correction ensures that the estimated breast age gap accounts for systematic bias, providing a more accurate assessment of biological aging across different diagnostic groups.

## Occlusion sensitivity analysis

To reflect the importance of different regions of the breast in estimating biological age, we employed the occlusion sensitivity analysis[19] to calculate the age-specific saliency map. This method calculates the difference in the model's performance when masking portions of input mammograms, offering insights into the significance of various breast structures, such as calcifications, fibroglandular structures, fatty tissue, and lymph nodes. Given the considerable size of mammograms and the varying sizes of their contents, reflecting distinct structures within the breast, we utilized masks of different sizes ($32 \times 32$, $64 \times 64$, $128 \times 128$, and $256 \times 256$). The occlusion process involved iteratively masking the original mammograms along a grid, with each mask size

corresponding to a grid cell. The occluded images, represented by the masked areas with a black image, were then evaluated using the MAE metric to assess the model's performance. The delta MAE, calculated as $MAE_{occluded} - MAE_{original}$, was employed to quantify the extent to which the model's performance was affected. Here, $MAE_{occluded}$ signifies the performance with occluded images, while $MAE_{original}$ represents the performance with the original mammogram. Then, four delta MAE metrics, each associated with a specific mask size, were integrated and reconstructed as a saliency map with the same size as the mammograms through cubic interpolation. Therefore, the saliency map can accurately and finely represent the importance of breast tissue related to breast aging, regardless of the size difference of breast tissue.

## Separate dataset retraining experiment

In this study, the primary results are derived from our proposed model, ensembled Mammo-AGE-Combined, which is trained on a combined dataset. To assess the robustness of the age-related biomarkers extracted by the Mammo-AGE model from mammograms, we conducted four separate dataset retraining experiments, resulting in four specific-dataset retrained Mammo-AGE models. Each model was retrained from scratch using the corresponding dataset. For the EMBED dataset, we also selected normal mammograms (BI-RADS 1 and 2) and split them into training, validation, and test sets at the patient level. Subsequently, we replicated the breast age gap analysis to ascertain the consistency of the results compared to those based on the Mammo-AGE-Combined model. This retraining experiment allows us to evaluate the generalizability and reliability of the age-related biomarkers extracted by our model across different datasets.

## Evaluation metrics

To measure the performance of our breast age prediction model, we employed four key evaluation metrics: Mean Absolute Error (MAE), the Pearson correlation coefficient ($r$), the Spearman's rank correlation coefficient ($r_s$), and Cumulative Score (CS). These metrics align with established methods in previous studies[17,19]. The MAE is defined as $MAE = \frac{1}{n} \sum_{i-1}^{n} |y-x| \times 100\%$ where $y$ is the predicted breast age and x is chronological age. It is widely used for evaluating the performance of age estimation; lower MAE means more accurate prediction. The Pearson correlation coefficient ($r$) is calculated based on the assumed linear relationship between predicted breast age and chronological age. The nonparametric based Spearman's rank correlation coefficient ($r_s$) is also reported. The CS represents the accuracy of age estimation given a threshold $\alpha$ (5 years in this study), calculated as $cs(\alpha) = \frac{N_{\varepsilon \leq \alpha}}{N} \times 100\%$ where $N_{\varepsilon \alpha}$ is the number of samples with an absolute error of age estimation $\varepsilon$ no higher than the threshold $\alpha$. A higher CS score signifies better performance.

## Statistical analysis

Descriptive statistics, including means and standard deviations (SDs), numbers, and percentages, were utilized to report baseline characteristics. Chronological age and breast density adjusted analysis of Covariance (ANCOVA)[70] was used to compare the difference in breast age gap between breast cancer patients and the healthy group. Cox proportional hazards regression models considering breast age gap as a continuous linear term were fitted to estimate the effect of a 1-year increase in breast age gap on breast cancer risk. Adjustment of covariates in Cox models was made for chronological age, breast density, and additional known risk factors or covariates. We tested the proportional hazards assumption using Schoenfeld residuals[15,44]. For covariates that violated this assumption, we employed stratification in the Cox models to account for their nonproportionality. Several sensitivity analyses were conducted, including covariate adjustments and dataset-specific model validation. Differences between Kaplan–Meier survival curves of high- and low-risk groups were assessed using the log-rank test. All the statistical tests are two-sided. The analyses were

performed using R (version 3.3.0, R Foundation for Statistical Computing, www.R-project.org, Vienna, Austria) and Stata (version 13, StataCorp, Texas, USA). The 95% confidence intervals (CI) of AUROC and AUPRC metrics are estimated by bootstrapping with 1000 bootstraps for each measure. No statistical method was used to predetermine sample size. All available eligible mammogram examinations meeting the inclusion criteria were included in the analyses. No data were excluded from the analyses.

## Data availability

The raw inhouse data of this study are protected and are not publicly available due to data privacy, while supporting the findings including the imaging data can be available under restricted access for non-commercial and academic purposes only. Access can be obtained by request to the corresponding author, T.T (email: taotanjs@gmail.com). The requirements will be evaluated concerning institutional policies, and data can only be shared for non-commercial academic usage with a formal material transfer agreement. All requests will be promptly reviewed within a timeframe of 20 working days. The public RSNA dataset is available at https://www.kaggle.com/competitions/rsna-breast-cancer-detection. The public VinDr dataset is available at https://vindr.ai/datasets/mammo. The public CMMD dataset is available at https://wiki.cancerimagingarchive.net/pages/viewpage.action?pageId=70230508#70230508eafc83b6ab624b80900e0f069203bacd. The public EMBED dataset is available at https://registry.opendata.aws/emory-breast-imaging-dataset-embed/. The public CSAW-CC dataset is available at https://researchdata.se/en/catalogue/dataset/2021-204-1. All other data supporting the findings of this study are present in the main article, supplementary or source data file. Source data are provided in this paper. Source data are provided with this paper.

## Code availability

Albumentations Python library (https://github.com/albumentations-team/albumentations). All comparison experiments for baseline models were conducted using official implementations with pretrained weights from the PyTorch model library (https://docs.pytorch.org/vision/main/models.html). All custom codes related to training and developing the deep learning models are available on https://github.com/Netherlands-Cancer-Institute/Mammo-AGE. It should only be used for academic research.

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

## Acknowledgements

The authors are grateful to Prof. Marjanka K. Schmidt and Dr. Terry Chan (The Netherlands Cancer Institute) for providing valuable suggestions. The authors would also like to acknowledge the Research High Performance Computing (RHPC) facility of the Netherlands Cancer Institute (NKI). This work is supported by the Shenzhen Medical Research Fund (Project No. D2501013, received by T.T.) and Science and Technology Development Fund of Macao (Project No. 0105/2022/A, received by T.T.). The authors thank the support from the Chinese Scholarship Council scholarship (CSC) (202107720016, 202006930001, and 202006240065, received by X.W., Y.G., and L.H.), and the Guangzhou Elite Project (TZ–JY201948, received by T.Z.).

## Author contributions

Conceptualization: X.W., T.T. and R.M. Data Collection: X.W., T.Z., L.H. and Y.G. Methodology: X.W., T.T., Y.G. and R.M. Investigation: X.W., Y.G. and A.P. Visualization: X.W., Y.G. and A.P. Supervision: T.T., R.M. and R.B.T. Writing-original draft: X.W., Y.G., C.L., T.Z. and L.H. Writing-review & editing: R.M., N.K., H-Y.Z., E.M., S.W., X.W., C.A.D., J.T. and T.T.

## Competing interests

The authors declare no competing interests.
