## [Transparent Peer Review file · Nature Communications]

Mammo-AGE: Deep Learning Estimation of Breast Age from Mammograms

Corresponding Author: Professor Tao Tan

Version 0:

Reviewer comments:

Reviewer #1

(Remarks to the Author)

AI applications to medical imaging promises significant opportunities to reduce disease mortality via early detection. In particular, AI improvements to mammographic breast screening are not limited to improved detection but also risk prediction, enabling risk stratification and personalized screening recommendations. The article by Wang and colleagues estimates age from mammograms, utilizing deep learning, and shows that abnormal breast aging is associated with increased breast cancer risk. These results are of considerable significance to the field as this novel AI-generated measure of breast age could be used as a biomarker to identify those at increased risk who could benefit from primary prevention strategies and/or supplemental screening.

The methodology is sound and there is sufficient detail to reproduce the work. The conclusions are supported by the results, with mostly minor suggestions for further improvement. They are as follows:

The association between breast tissue aging and breast cancer incidence is well documented since "Pike's Model" (Nature, 1983 PMID: 6866078) which describes breast tissue aging in relation to reproductive factors (e.g. age at menarche, pregnancies, age at menopause). The authors should consider a discussion of Pike's model.

Page 4 line 100 – Please define breast density the first time it is referenced.

Figure 1 – Please define Q, K, V within the Instance-Bag Transformer and CE in the description.

The distributions of BIRADS breast density are very different between internal datasets. Ethnic differences are expected but the difference between BIRADS B and C within Inhouse, RSNA and EMBED are surprising. Could the authors comment on the representativeness of the image sets and associated implications to the training/testing or risk estimates? For example, if images were sourced from private mammography versus population-based screening programs, some image sets could potentially over-represent higher risk women (with higher breast density and higher socioeconomic status).

Page 6 line 128: From Table 1, $18,213+11,419+4,250=33,882$ examinations negative for breast cancer (not 39,619). The total number of exams for Inhouse (23,379) does not equal the sum of exams from negative plus positive breast cancer ($18,213+615$), unlike the other study samples that do add up. I recognize that there are multiple exams per patient, but the difference ($23,379-18,213-615=4,551$) does not match the difference between $39,619-33,882=5,737$? How were previous negative mammograms from breast cancer cases treated? Was repeatability of the measurement in consecutive mammograms examined?

Minor comment – can the Pearson/MAE/CS stats be placed in bottom right corner of all 6 graphs in Figure 2?

Minor comment Page 8 line 180 "details" is spelt incorrectly.

Tables S1 and S2. Mammo-age spelt incorrectly in S1. The header descriptions are confusing; the S1 header should refer only to the S1 table, not the S2 table. Clarify that S1 is the first ablation (different backbones) and S2 is the second (different

modules) and third (different image sizes) ablations. The corresponding paragraphs in the results (page 9 lines 189-202) could also be clearer, outlining each of the ablation studies. Line 192 should clarify that, for a fair comparison, all methods comparing different modules or different image sizes, utilized the same backbone (ResNet-18).

Table 2: Age range in the Combined dataset is 14-89?

Figure 3 is a bit repetitive in combination with Table 2. Consider including Table S4 instead of Table 2.

Page 15. Line 289. The number of healthy patients/exams for internal datasets is not available in Table 1 e.g. 9,507, 8419, 1423 (but is available for external datasets 58,430 and 465)? The number for the Inhouse data set does not match Figure 6 (e.g. 9,507 vs 9,777).

Page 17 line 309 – ANCOVA generated p-values were adjusted for age and breast density. What were the unadjusted p-values?

Minor comment – Table 3 & S3, Inhouse misspelt. Table 3, EMBED also misspelt.

Table 3 – Is dichotomizing the age gap into high/low based on the median (or quartiles) informative? Consider other metrics to describe the measure's ability to discriminate between cases and controls (e.g. Dench et al 2019 PMID: 31892647).

In the discussion, the authors mention other AI-based mammographic measures that have been shown to strongly predict breast cancer risk, but do not comment on the comparative predictive power of breast age (or provide the metrics for comparison).

This investigation is interesting because instead of training AI algorithms to predict a dichotomous response (e.g. cancer case vs control), the authors model age – a continuous, known, and precise measurement. This is a potentially powerful approach that presumably requires relatively smaller image datasets for training and testing.

Finally, could this research inform optimal age in which to target mammographic screening (based on a single early screen)?

(Remarks on code availability)

Reviewer #2

(Remarks to the Author)

This study by Wang et al, "Mammo-AGE: Deep Learning Estimation of Breast 1 Age from Mammograms," introduces a deep learning approach to estimate the biological age of the breast using mammograms, termed "Mammo-Age". The model was developed and validated on a large dataset of over 100,000 mammogram exams from patients aged 18 to 98 years. It accurately estimates breast age, with a mean absolute error (MAE) of 4.2 to 5.3 years and shows strong correlations with chronological age. The difference between estimated breast age and chronological age (breast age gap) was found to reflect breast health, with higher gaps observed in breast cancer patients. Overall, the methods for developing Mammo-AGE are appropriate and the authors are careful to consider important elements in the age estimation, such as accounting for regression dilution. They also demonstrate impressive robustness across a diverse range of cohorts, which is a major advantage. The major weakness of this work is to explore the biological and clinical relevance of how this may be used beyond what is currently presented. Below are specific comments.

Major Comments:

- 1) To evaluate the clinical relevance, there are several clear questions that come to mind
 - a. One important way to evaluate the clinical utility is to compare this to the clinical findings from the mammogram. What is the accuracy of this in terms of diagnosing current breast cancer? How does this compare with the clinical evaluation of the mammogram (abnormal/normal/needs further biopsy etc.)?
 - b. Are there specific metrics (e.g. age gap greater than some value) that can discriminate between cases and controls? If Mammo-AGE isn't good at discriminating case status, can it be used in conjunction with mammogram evaluation to improve the accuracy of biopsy results or diagnosis?
- 2) How easy would this be to implement the algorithm with mammograms that are taken in the clinic?
- 3) The incident case associations generally have statistically significant associations, but the effect sizes seem very low when not stratified into low and high-risk groups. Even when stratified, the differences seem modest at best. It is not clear that this difference is clinically meaningful. Is there a way to translate this into its clinical impact? Are there any comparators with current clinical standards that can help assess the effect size? How well does this discriminate over different time periods (e.g. 1-year, 5-year, 10-year)?
- 4) In the development of Mammo-AGE are there some areas or regions of the breast that are more informative in this

measure than others? Are there any other things that can be gleaned about the understanding of breast aging that is extracted from the mammograms to develop Mammo-AGE?

5) It is surprising that the only covariates that were adjusted for are chronological age and breast density. This is particularly the case as there are other risk factors and confounders for breast cancer. How do these factors affect the prediction/association?

6) When reading the manuscript, it is not clear how much other work related to breast aging has been done, other than to say that people with breast cancer have increased biological age in DNAm studies. The manuscript should elaborate more on this. If it is limited to just that, it should be emphasized as more of a point of novelty. With the increase in DNAm organ specific aging clocks, are there other metrics of breast age that this can be compared with?

7) Are there other studies on the physiology of the breast and how it presents in a mammogram? Is there any way to assess if Mammo-AGE captures any important physiology of breast tissue changes leading to cancer? If so, what are they?

8) There is still a lack of general cohort characteristics presented that provides any additional details on the cohorts, like some of the basic demographics (age, BMI, race, ethnicity, education, comorbidities, smoking). Are

Minor:

Line 53 There is not just one "epigenetic clock" Please correct as per the literature (look in the Aging Biomarker Consortium MS in Nature Medicine).

Line 180 says "detailes"

In the discussion, there is a comment about the inclusion of cohorts around the world. It would be important to highlight the diversity and robustness of findings across these populations.

(Remarks on code availability)

Reviewer #3

(Remarks to the Author)

In this study, the authors develop a deep learning model using mammograms as input to predict the age of a patient. The model seems to produce reasonably good results. What's more interesting is, they found that the difference between the predicted and chronological age may serve as a risk factor for breast cancer. The study seems to be carefully conducted with nice results. However, close inspection by the reviewer revealed the findings to be questionable. I'm also suspicious if the study is properly motivated.

Major comments:

- What really is the goal of the study? Why do we even need to predict the age of breasts? If the goal is cancer risk prediction, then why not predict cancer directly? It seems the entire study is motivated by Ref. 11-12. But copying other people's homework is not considered as innovation.
- A proper baseline should be a model like Mirai that predicts breast cancer risk and age at the same time, and is trained on a large mammography dataset. If your model cannot beat Mirai, why bother developing your own?
- Although the age gap seems to be a risk factor for cancer, is it possible that that is simply because the cancer group is older than the healthy group (see Table 1)? This can be mitigated by matching the age of the healthy group with that of the cancer group, or use age as a covariate in your Cox regression analysis.
- The authors spent a lot of effort tinkering with different machine learning techniques like GLT, MVL, and POE but they all led to marginal improvements. The only change that led to significant improvement is larger image size. Maybe the authors should stop wasting time and just buy more GPUs?
- The model didn't do as well on the VinDr and CMMD data. Is it because of the younger population or the race/ethnicity? This is worth clarifying.
- Where did the weight ratios (2,2,5,1,5) of different backbones come from? This raises the suspicion of overfitting by hyperparameters.
- The authors claim that "The saliency maps reveal that the model commonly focuses on features such as breast skin thickness, fibroglandular tissue, calcifications, masses, and breast vessels". To be honest, this isn't obvious to my eyeballs. The saliency maps serve more as eye candy than insightful tools to understand the model. Additionally, why not look at the attention maps as well?
- The scatter plots in Fig. 6 are not useful. I don't see any trend between the healthy and cancer groups.
- The bias corrected age used chronological age as input, which is the target of your model. Was this strictly done on the train set, or on the test set (most likely)?
- Line 572-583, the description of bias correction is poor. I'm sure the equations are incorrect.
- The model is multi-tasking to predict age and density in parallel but the results on density prediction are not mentioned. If the density is associated with age (line 100), why not concatenate the predicted density with the embedding layer to predict the age? That may enhance the accuracy.

Minor comments:

- Methods like GLT, MVL, and POE are not standard and very recent. The authors should briefly describe these methods and provide some rationale why they were chosen. Fig. 1 looks confusing and doesn't help much in clarifying the methodology.
- Medical images are known to be biased by racial/ethnicity groups. Would like to see a breakdown based on that.

(Remarks on code availability)
HTTP 404.

Reviewer #4

(Remarks to the Author)

(Remarks on code availability)

Version 1:

Reviewer comments:

Reviewer #1

(Remarks to the Author)

The authors have adequately addressed my comments.

(Remarks on code availability)

Reviewer #3

(Remarks to the Author)

the authors have addressed most of my comments in great details. however, i found the model validation part - Figure R3.2.2 - to be concerning and would like more details:

* assuming the "baseline" is a network training from scratch and "ours" is a network pretrained to predict breast age. is there any overlap between the data used for pretraining and the data used for finetuning? the authors should clarify which dataset is used for pretraining and which is used for finetuning.

* did you finetune Mirai on your internal datasets?

* did you finetune the networks on the external dataset (CSAW-CC)?

* what do the error bars in the figure represent?

(Remarks on code availability)

Reviewer #4

(Remarks to the Author)

(Remarks on code availability)

None

Reviewer #5

(Remarks to the Author)

See the attached PDF file.

(Remarks on code availability)

Version 2:

Reviewer comments:

Reviewer #5

(Remarks to the Author)

The authors addressed my comments and I do not have further comments.

(Remarks on code availability)

REVIEWER COMMENTS

Reviewer #1 (breast cancer imaging, mammography):

AI applications to medical imaging promises significant opportunities to reduce disease mortality via early detection. In particular, AI improvements to mammographic breast screening are not limited to improved detection but also risk prediction, enabling risk stratification and personalized screening recommendations. The article by Wang and colleagues estimates age from mammograms, utilizing deep learning, and shows that abnormal breast aging is associated with increased breast cancer risk. These results are of considerable significance to the field as this novel AI-generated measure of breast age could be used as a biomarker to identify those at increased risk who could benefit from primary prevention strategies and/or supplemental screening. The methodology is sound and there is sufficient detail to reproduce the work. The conclusions are supported by the results, with mostly minor suggestions for further improvement.

We sincerely thank the referee for the very positive comments and support of our work. We especially appreciate the reviewer's recognition of the novelty and considerable clinical significance of our AI-based breast aging measure as a novel biomarker for risk stratification and personalized screening.

They are as follows:

Comment 1: The association between breast tissue aging and breast cancer incidence is well documented since "Pike's Model" (Nature, 1983 PMID: 6866078) which describes breast tissue aging in relation to reproductive factors (e.g. age at menarche, pregnancies, age at menopause). The authors should consider a discussion of Pike's model.

We appreciate the suggestion and have incorporated a discussion of Pike's Model ¹ in the revised manuscript.

Action taken: We now highlight Pike's Model as a foundational framework linking breast tissue aging to reproductive factors, and its relevance to breast cancer risk and screening strategies (page 2, lines 44-45): "... Breast tissue aging is also affected by reproductive factors, such as age at menarche, pregnancies, and age at menopause. Thus, the relation between calendar age and breast cancer risk is not linear ¹. However, given the strong "residual" association between aging and breast cancer, population-wide mammography screening regimens primarily rely on age-based eligibility criteria for early detection and mortality reduction."

Comment 2: Page 4 line 100 – Please define breast density the first time it is referenced.

We have added the definition of breast density in the revised manuscript.

- Page 4, line 105: "Breast density, defined as the proportion of fibroglandular to fatty tissue, typically decreases with age ²."
- Table 1, lines 150-152: "ACR A (mostly fatty), ACR B (scattered fibroglandular), ACR C (heterogeneously dense), and ACR D (extremely dense)."

Comment 3: Figure 1 – Please define Q, K, V within the Instance-Bag Transformer and CE in the description.

We agree with the reviewer that a detailed definition of terms in the description is more appropriate.

Action taken: As we have modified Fig.1 to serve as an overview of the study and moved the details of the deep learning model into Fig. S1, we have added description for Fig. S1 (Supplemental results, page 2):

“Q, K, and V in the Instance-Bag Transformer refer to the query (Q), key (K), and value (V) matrices used in the attention mechanisms to capture relationships between instances (single views) and bags (four-view examinations). The model incorporates multi-task learning for density prediction and uses a combination of cross-entropy (CE) loss, mean-variance (MVL) loss, and probabilistic ordinal embedding (POE) loss functions to constrain the learning process.”

Comment 4: The distributions of BIRADS breast density are very different between internal datasets. Ethnic differences are expected but the difference between BIRADS B and C within Inhouse, RSNA and EMBED are surprising. Could the authors comment on the representativeness of the image sets and associated implications to the training/testing or risk estimates? For example, if images were sourced from private mammography versus population-based screening programs, some image sets could potentially over-represent higher-risk women (with higher breast density and higher socioeconomic status).

We agree, as the reviewer correctly noted, breast density distributions vary across datasets.

Distributions of breast density: The observed differences in breast density distributions likely reflect the diversity of our data sources and the reporting habits of the radiologists involved. For instance, the inhouse dataset, sourced from population-based screening in the Netherlands, aligns with previously reported density distributions for this population, where ACR B classifications are more prevalent than ACR C, which is likely due to Dutch radiologists ranking densities lower than American radiologists on average (Wanders et al., *Breast Cancer Res. Treat.*, 2017³).

Representativeness and robustness validation: Recognizing the variability in real-world practices, as also reflected by the shown external data sources, we carefully adjusted the covariates, including breast density, in our analysis. Our results consistently show a significant association between breast age gap and breast cancer risk across different datasets (as shown in Fig. 6). Beyond adjusting for breast density, to further validate the robustness and minimize dataset-specific biases, we conducted extensive cross-dataset testing with varying density distributions (Fig. S6). Specifically, we trained models on one dataset and tested them on others to assess generalizability beyond cohort-specific characteristics. The consistent patterns observed across these independent cohorts indicate that our model captures fundamental biological aging features associated with breast cancer, rather than cohort-specific effects. This supports its generalizability and mitigates concerns about dataset representativeness.

Action taken: We have added a discussion in the revised manuscript emphasizing the robustness of our findings, Page 22, Lines 494-499: *“The breast aging-related biomarkers identified by our Mammo-AGE model demonstrate*

its generalizability and stability when applied to both internal and unseen external datasets (Fig. 6). Due to differences in breast density distributions across datasets, the breast density and age were carefully adjusted in all analysis. Our findings are further strengthened by the cross-dataset retraining experiment (Fig. S5). The comparative analysis highlights the model's adaptability to diverse datasets and its ability to consistently capture breast aging patterns.”

Comment 5: Page 6 line 128: From Table 1, $18,213 + 11,419 + 4,250 = 33,882$ examinations negative for breast cancer (not 39,619). The total number of exams for Inhouse (23,379) does not equal the sum of exams from negative plus positive breast cancer ($18,213 + 615$), unlike the other study samples that do add up. I recognize that there are multiple exams per patient, but the difference ($23,379 - 18,213 - 615 = 4,551$) does not match the difference between $39,619 - 33,882 = 5,737$? How were previous negative mammograms from breast cancer cases treated? Was repeatability of the measurement in consecutive mammograms examined?

This is a great point by the reviewer. We apologize for the confusion it may have caused, and for this oversight and miscommunication. The reviewer is correct that the total number of negative mammograms for the internal datasets in the study is $18,213$ (inhouse) + $11,419$ (RSNA) + $4,250$ (VinDr) = $33,882$.

Figure R1.5.1 The flowchart of the inhouse dataset collection.

Regarding the inhouse dataset, as shown in Figure R1.5.1, we initially collected 23,379 mammography examinations from 7063 women from our private hospital. Then, all screening mammograms reported to be negative

(BI-RADS 1 and 2) were selected, resulting in 18,828 examinations from 5,048 patients. Among these, 615 examinations were subsequently classified as positive, corresponding to patients diagnosed with breast cancer within one year after the mammogram. The remaining 18,213 examinations were from women without a cancer diagnosis within one year.

To assess the relationship between breast cancer and the breast age gap, for individuals who later developed breast cancer, their earlier negative mammograms were treated as negative exams until the diagnosis. Only the mammogram closest to diagnosis (within one year) was considered positive.

Action taken: We have clarified Table 1 to maintain consistency and clearly reflect the correct number (page 24, lines 561-562) and also provided a detailed flowchart (Figure S1) outlining the dataset collection process for clarity.

Comment 6: Minor comment – can the Pearson/MAE/CS stats be placed in bottom right corner of all 6 graphs in Figure 2?

We have adjusted Fig. 2 in the revised manuscript, as shown below in Figure R1.6.1.

Figure R1.6.1 The scatterplots of predicted breast age and chronological age for internal and external datasets.

Comment 7: Minor comment Page 8 line 180 “details” is spelt incorrectly.

We apologize for the oversight. We have carefully corrected all the typos in the revised manuscript.

Comment 8: Tables S1 and S2. Mammo-age spelt incorrectly in S1. The header descriptions are confusing; the S1 header should refer only to the S1 table, not the S2 table. Clarify that S1 is the first ablation (different backbones) and S2 is the second (different modules) and third (different image sizes) ablations. The corresponding paragraphs in the results (page 9 lines 189-202) could also be clearer, outlining each of the ablation studies. Line 192 should clarify that, for a fair comparison, all methods comparing different modules or different image sizes, utilized the same backbone (ResNet-18).

Thank you for the heads-up. We have revised the figures, tables, and corresponding text accordingly.

Action taken:

a) Spelling correction: We have corrected the spelling of “Mammo-AGE” in Table S1.

b) Header clarifications for Tables S1 and S2: Each header now accurately refers only to its respective table:

- *“Table S1. Results of Mammo-AGE models’ performance with ensemble and different backbones on the combined dataset using five-fold cross validation.”*
- *“Table S2. Results of the ablation studies on the combined dataset using five-fold cross-validation. The first ablation study compares model performance across different modules. The second ablation study examines the model performance with different image sizes. The default backbone is ResNet-18.”*

c) Clarification in the Results section: For clarity, we have explicitly stated that (page 9, lines 219-220): *“For a fair comparison, during the ablation studies, all methods comparing different modules or image sizes utilized the same backbone (ResNet-18).”*

Comment 9: Table 2: Age range in the Combined dataset is 14-89?

Thank you for pointing out this typo. We confirm that the correct age range in the Combined dataset is 18-89. We have corrected this in Table 2 of the revised manuscript.

Comment 10: Figure 3 is a bit repetitive in combination with Table 2. Consider including Table S4 instead of Table 2.

We agree and have revised Table 2 to reduce repetitive content and to include the Table S4 results, as suggested.

Comment 11: Page 15. Line 289. The number of healthy patients/exams for internal datasets is not available in Table 1 e.g. 9,507, 8419, 1423 (but is available for external datasets 58,430 and 465)? The number for the Inhouse data set does not match Figure 6 (e.g. 9,507 vs 9,777).

Thank you for pointing out these discrepancies.

Clarification on healthy patient counts: Table 1 reports the total dataset sizes, encompassing all splits. For clarity, the independent test set (20% of the whole internal dataset) includes 9,777, 8,419, and 1,423 healthy patients for the inhouse, RSNA, and VinDr datasets, respectively. The model does not access the independent test set during the five-fold cross-validation. Besides, the entire external datasets are leveraged for direct model inference, which shows a quite large number of images.

Correction for inhouse dataset count: We apologize for the inconsistency of the count number and have corrected it to 9,777 in the revised manuscript.

Action taken: We have updated the manuscript to ensure consistency across figures and tables.

Comment 12: Page 17 line 309 – ANCOVA generated p-values were adjusted for age and breast density. What were the unadjusted p-values?

We have included the unadjusted p-values (Table R1.12.1), which consistently show that the breast age gap remains significantly higher in the breast cancer-diagnosed group compared to healthy women. This further supports the robustness of our findings, demonstrating that the observed differences are independent of age and breast density.

Dataset	Breast age gap	Breast age gap	Un-adjusted p-values	Adjusted p-values
	(Mean±STD) Healthy population	(Mean±STD) Breast cancer patients		
Inhous	0.373±4.546	0.819±4.751	0.024 *	0.017 *
RSNA	-1.883±4.851	-0.951±5.031	<0.001 ***	<0.001 ***
VinDr	-0.059±4.149	0.888±4.225	0.048 *	0.033 *
EMBED	0.073±5.026	0.489±4.885	0.047 *	0.039 *
CMMD	0.383±4.732	0.887±4.773	0.049 *	0.049 *

Table R1.12.1 Breast age gap differences between healthy women and breast cancer patients, and statistical analysis with and without adjustment by age and breast density.

Action taken:

- The unadjusted p-values have been added to Table S7.
- Page 17, Lines 346–347: “The results of the unadjusted p-values are reported in Table S7, confirming that the observed differences are independent of age and breast density.”

Comment 13: Minor comment – Table 3 & S3, Inhouse misspelt. Table 3, EMBED also misspelt.

Thank you for pointing out these typos. We have corrected them and carefully reviewed the revised manuscript.

Comment 14: Table 3 – Is dichotomizing the age gap into high/low based on the median (or quartiles) informative? Consider other metrics to describe the measure’s ability to discriminate between cases and controls (e.g. Dench et al 2019 PMID: 31892647).

This is a valid point. The reviewer is correct that dichotomizing the age gap is informative.

According to Figure R1.14.1, we note that the relation between the breast age gap and breast cancer risk is non-linear. Therefore, dichotomizing the age gap into high/low based on the median is reasonable. Our findings are consistent with the previous studies showing that individuals with extreme deviations in breast tissue aging are at particularly elevated risk, echoing the disproportionate risk accumulation in “high-extreme agers”⁴ (Oh et al., Nature, 2023).

Furthermore, we analyzed breast age gap as a continuous variable and found that each additional year in breast age gap was also significantly associated with increased breast cancer risk ($p < 0.001$ in the inhouse dataset, $p = 0.002$ in EMBED, shown in Table 3).

Figure R1.14.1 Association between breast age gap and 10-year (inhouse dataset) or 5-year (EMBED dataset) breast cancer risk using restricted cubic spline regression models. HRs for future breast cancer events are shown according to the breast age gap, adjusted for age and density.

Additional discriminatory metrics: To further assess the discriminatory power of the breast age gap, we also performed odds ratio analysis, according to Dench et al (*BMJ Open*, 2019)⁵. Results show that our proposed breast age gap could identify more future breast cancer cases than breast density, which is a well-established but similarly modest risk factor from mammograms. This finding highlights its potential as an independent biomarker from mammograms for future risk stratification.

	Inhouse 10-year risk		EMBED 5-year risk	
	Odds ratios	P-value	Odds ratios	P-value
Breast density	1.060 (1.018-1.104)	0.005	1.069 (1.005-1.136)	0.03
Breast age gap	1.070 (1.028-1.114)	<0.001	1.114 (1.048-1.183)	<0.001

Table R1.14.1 Odds ratios analysis for breast cancer based on breast density and breast age gap.

Action taken: We have added the odds ratio results to Table S8. We have added the discussion on page 23, lines 515-515: *“The relation between the breast age gap and breast cancer risk is not linear (Fig. S7). Our findings confirm the hypothesis that individuals with extreme deviations in breast tissue aging are at particularly elevated risk, echoing the disproportionate risk accumulation in the “extreme agers”⁴.”*

Comment 15: In the discussion, the authors mention other AI-based mammographic measures that have been shown to strongly predict breast cancer risk, but do not comment on the comparative predictive power of breast age (or provide the metrics for comparison).

We appreciate the reviewer’s suggestion. Yes, as the reviewer rightfully notes that we initially did not include a direct comparison with an AI-based model specialized for breast cancer risk prediction. As our Mammo-AGE model is designed to estimate breast age, a biomarker reflecting broader biological aging processes rather than solely cancer risk prediction. However, we recognize the importance of benchmarking against established breast cancer risk models to better contextualize our findings.

Comparison with the risk model: To evaluate its predictive power, and inspired by other reviewers, we applied our Mammo-AGE model for downstream breast cancer risk prediction and compared its performance with different state-of-the-art (SOTA) models. As shown in Figure R1.15.1, our model consistently outperformed the SOTA models across both short-term (2-year) and long-term (5-year) risk prediction, demonstrating that deep-learning-derived breast age features improve risk prediction accuracy and the risk model’s generalizability.

Comparison experiments on diagnostic tasks: Furthermore, it is important to highlight that breast age estimation has potential value beyond risk prediction, as shown in other clinical applications such as breast cancer diagnosis (Figure R1.15.2).

Action taken: We have added the discussion on page 23, lines 528-533: "Furthermore, when applying the Mammo-AGE model for downstream clinical tasks, e.g., diagnosis and risk prediction, our Mammo-AGE model outperformed the SOTA methods across multiple (internal and external) datasets (Fig. 7). The results of downstream tasks demonstrate that Mammo-AGE achieves superior age prediction accuracy while also providing meaningful information for breast cancer risk assessment and diagnosis. More importantly, our findings demonstrate that breast age estimation improves downstream tasks learning efficiency and generalizability in mammography-based deep learning models."

A. Risk prediction

Figure R1.15.1 Comparison results of short-term (2-year) and long-term(5-year) BC risk prediction with baseline model, SOTA risk model⁶, and SOTA foundational model⁷. Baseline: trained the model for risk prediction without age prediction pretrained weight; Ours: we finetuned the model for risk prediction with age prediction pretrained weight. SOTA risk model: MIRAI risk prediction model. SOTA risk model: We applied the Mammo-clip model, which was pretrained on mammograms and radiology reports. AUROC: Area Under Receiver Operating Characteristic Curve; AUPRC: Area Under Precision-Recall Curve.

B. Diagnosis

Figure R1.15.2 Comparison results of diagnostic prediction. The “*” represents that the model is finetuned on inhouse and EMBED datasets, then tested directly on all others. Baseline: trained the model for downstream task without age prediction pretrained weight; Ours: we applied the model for downstream task with age prediction pretrained weight. Risk: MIRAI risk prediction model. SSL: We applied the Mammo-clip model, which was pretrained with mammograms and radiology reports.

Comment 16: This investigation is interesting because instead of training AI algorithms to predict a dichotomous response (e.g. cancer case vs control), the authors model age – a continuous, known, and precise measurement. This is a potentially powerful approach that presumably requires relatively smaller image datasets for training and testing.

Thank you for highlighting this important strength of our approach. We appreciate the reviewer’s recognition that modeling precise breast age offers a more efficient granular assessment of tissue characteristics.

To further evaluate this advantage in learning efficiency, we have applied Mammo-AGE to the task of breast cancer classification using different fractions of training data. As shown in Figure R1.16.1, fine-tuning Mammo-AGE consistently outperformed training from scratch (i.e., baseline), particularly when training data was limited (5% of the training dataset, about 2000 mammogram examinations).

Figure R1.16.1 AUC results with different fractions of training data during finetuning on the task of breast cancer classification.

Clinical Implications: These findings suggest that breast age estimation may serve as a robust imaging biomarker that improves model generalization while reducing the reliance on large-scale labeled datasets. This efficiency is particularly relevant in clinical settings, where access to millions of high-quality labeled images is often not feasible.

Action taken: We have incorporated these results in the manuscript, emphasizing our model's strengths and its potential clinical implications in the revised manuscript (page 23, lines 531–535): *"More importantly, our findings demonstrate that breast age estimation improves learning efficiency and generalizability in mammography-based deep learning models. Fine-tuning on breast age features significantly enhances cancer classification even under limited data settings, highlighting its potential as an imaging biomarker for breast cancer detection."*

Comment 17: Finally, could this research inform optimal age in which to target mammographic screening (based on a single early screen)?

This is a very interesting question. Current mammographic screening guidelines primarily use chronological age as the basis for eligibility but do not account for individual variations in breast tissue aging. **While this study does not determine an exact optimal screening age**, our findings suggest that biologically older breasts may benefit from additional screening, while biologically younger breasts may allow for less frequent screening.

To explore this, we stratified women into decelerated- (lower age-gap), medium-, and accelerated- (higher age-gap) aging groups, and evaluated their 2-year breast cancer risk using the public standard screening EMBED dataset.

Then we analyzed the average diagnostic age and calculated the odds ratios (ORs) for breast cancer across these groups (Table R1.17.1)

Decelerated-aging women had a later average diagnostic age (72 years, 95% CI: 47-98). Accelerated-aging patients were diagnosed significantly earlier (62 years, 95% CI: 47-80) and had a 2.996-fold increased risk of breast cancer. Each 1-standard deviation (SD) increase in breast age gap was associated with a 11.6% increase in breast cancer risk (OR = 1.116, 95% CI: 1.036–1.203), highlighting that even moderate shifts in biological breast aging significantly impact cancer risk.

Risk Group	Odds Ratios (95% CI)	P Value	Avg. Diagnostic Age (95% CI)
Per SD Increase in Breast Age Gap	1.116 (1.036-1.203)	0.0040	63 (38-89)
Decelerated aging group	Reference	-	72 (47-98)
Medium aging group	1.927 (0.957-3.878)	0.0661	63 (38-89]
Accelerated aging group	2.966 (1.341-6.563)	0.0073	62 (47-80)

Table R1.17.1 Odds ratios and diagnostic age of each risk group. Decelerated aging group: $gap < (Mean - 1.96SD)$; Medium aging group: $(Mean - 1.96SD, Mean + 1.96SD)$; and Accelerated aging group: $gap > (Mean + 1.96SD)$

However, the model in this study predicts breast age at a single time point, reflecting current tissue status rather than a long-term risk trajectory. It remains unclear if a breast appearing 5 years older at age 30 will still be 5 years older at age 45. Longitudinal studies tracking breast aging over time are needed to determine the stability and predictive value. Future work incorporating multi-time-point mammograms could provide more precise risk stratification and screening recommendations beyond static age cutoffs.

Action Taken: We have expanded the discussion in the revised manuscript on page 23, lines 522–527: "As evidenced by Table S9, our findings suggest that breast age gap, as an imaging-based biomarker, has the potential to refine mammographic screening strategies. Specifically, women with a higher breast age gap may benefit from additional screening, while those with a lower gap might be candidates for less frequent screening. However, further research is needed to assess whether breast aging patterns remain stable over time using longitudinal mammogram-based model, which could support the development of personalized screening schedules."

Reviewer #2 (epidemiology, statistics):

This study by Wang et al, "Mammo-AGE: Deep Learning Estimation of Breast Age from Mammograms," introduces a deep learning approach to estimate the biological age of the breast using mammograms, termed "Mammo-Age". The model was developed and validated on a large dataset of over 100,000 mammogram exams from patients aged 18 to 98 years. It accurately estimates breast age, with a mean absolute error (MAE) of 4.2 to 5.3 years and shows strong correlations with chronological age. The difference between estimated breast age and chronological age (breast age gap) was found to reflect breast health, with higher gaps observed in breast cancer patients. Overall, the methods for developing Mammo-AGE are appropriate and the authors are careful to consider important elements in the age estimation, such as accounting for regression dilution. They also demonstrate impressive robustness across a diverse range of cohorts, which is a major advantage. The major weakness of this work is to explore the biological and clinical relevance of how this may be use beyond what is currently presented.

We appreciate the referee's enthusiasm for our study and for recognizing the strengths of the proposed Mammo-AGE model, including its robustness and careful consideration of key elements like regression dilution. We also appreciate your suggestion for improving the understanding of breast aging biology and the clinical relevance of the identified aging biomarker. We have provided a point-by-point response to your helpful comments below.

Below are specific comments.

Major Comments:

Comment 1: To evaluate the clinical relevance, there are several clear questions that come to mind.

- a. One important way to evaluate the clinical utility is to compare this to the clinical findings from the mammogram. What is the accuracy of this in terms of diagnosing current breast cancer? How does this compare with the clinical evaluation of the mammogram (abnormal/normal/needs further biopsy etc.)?
- b. Are there specific metrics (e.g. age gap greater than some value) that can discriminate between cases and controls? If Mammo-AGE isn't good at discriminating case status, can it be used in conjunction with mammogram evaluation to improve the accuracy of biopsy results or diagnosis?

We appreciate the reviewer's insightful suggestions regarding the clinical relevance of Mammo-AGE.

Comparison with mammographic findings: To assess Mammo-AGE's potential in diagnosing current breast cancer, we have compared it to radiologist evaluations of BI-RADS (Breast Imaging-Reporting and Data System) classifications across multiple datasets. As shown in Table R2.1.1, breast age gap achieved comparable performance to BI-RADS (Inhouse, EMBED, RSNA, and VinDr). The CMMD dataset was excluded due to lack of BI-RADS information.

Threshold and integration with mammogram evaluation: To define a meaningful threshold for risk assessment, we applied a breast age gap cutoff of $> \text{Mean} + 1 \text{ SD}$, corresponding to the abnormal aging group.^{8,9} Results show that combining breast age gap with BI-RADS further improves predictive accuracy across all datasets. This

suggests that the breast age gap may provide complementary information that enhances conventional mammographic assessments.

Dataset	Age gap (Thr>mean+1SD)	BI-RADS (Thr>BIRADS2)	Age gap score + BI-RADS score (Thr>0.5)
Inhouse	0.866 (0.858-0.873)	0.941 (0.936-0.945)	0.943 (0.938-0.948)
EMBED	0.835 (0.832-0.838)	0.829 (0.825-0.832)	0.965 (0.964-0.966)
RSNA	0.812 (0.805-0.820)	0.861 (0.854-0.868)	0.945 (0.941-0.950)
VinDr	0.818 (0.798-0.838)	0.792 (0.771-0.814)	0.931 (0.918-0.944)

Table R2.1.1. Accuracy results for breast cancer diagnosis across datasets.

Translational potential beyond biological aging: To further evaluate practical utility, we directly applied Mammo-AGE to the downstream breast cancer diagnostic task and compared its performance with several state-of-the-art (SOTA) deep learning models, which have already demonstrated better performance than radiologists⁶. As shown in Figure R2.1.1, our model consistently outperformed SOTA approaches across multiple datasets. The model was fine-tuned on the inhouse and EMBED datasets and then tested directly on other datasets. Importantly, we introduced a new public dataset, CSAW-CC¹⁰, which was not accessed by the model during either the age learning or diagnostic learning phases.

B. Diagnosis

Figure R2.1.1 Comparison results of diagnostic prediction. The “*” represents that the model is finetuned on inhouse and EMBED datasets and then tested directly on all others. Baseline: trained the model for the downstream task without age prediction pretrained weight; Ours: we applied the model for the downstream task with age prediction pretrained weight. Risk: MIRAI risk prediction model⁶. SSL: We applied the Mammo-clip

model⁷, which was pretrained on mammograms and radiology reports. AUROC: Area Under Receiver Operating Characteristic Curve; AUPRC: Area Under Precision-Recall Curve.

Comment 2: How easy would this be to implement the algorithm with mammograms that are taken in the clinic?

This is a very practical point regarding clinical implementation.

For mammogram use, it remains the standard imaging technique for breast cancer screening and diagnosis in many clinical settings¹¹, and is virtually always available. Our Mammo-AGE is designed for easy integration into clinical workflows, as it supports directly processes standard DICOM mammograms used in routine practice. The model has been validated across different manufacturers, demonstrating robust performance across imaging sources (Fig. S2). It is computationally efficient, requiring only ~4 seconds per four-view mammogram on a 2080Ti (12GB) GPU, making deployment feasible in clinical settings. The source code is publicly available. Potentially, screening patients can be informed of breast age in addition to breast density.

Action taken: We have added this discussion to highlight the model's clinical feasibility (page 27, lines 665–668): *“Mammo-AGE was developed based on standard DICOM mammograms. Besides, during model inference, it is hardware-friendly and runs efficiently (~4 seconds per exam on a 2080 Ti (12GB) GPU). It can be easily deployed on clinical servers or cloud-based systems, supporting integration into existing radiology workflows.”*

Comment 3: The incident case associations generally have statistically significant associations, but the effect sizes seem very low when not stratified into low and high-risk groups. Even when stratified, the differences seem modest at best. It is not clear that this difference is clinically meaningful. Is there a way to translate this into its clinical impact? Are there any comparators with current clinical standards that can help assess the effect size? How well does this discriminate over different time periods (e.g. 1-year, 5-year, 10-year)?

We appreciate this important and insightful comment. We agree that understanding the clinical relevance of observed effect sizes is essential for translating predictive biomarkers into practice. While the observed effect sizes in our study may appear modest, they are consistent with those reported for aging-related biomarkers, such as DNA methylation (DNAm) and epigenetic aging clocks associated with breast cancer¹².

Compare with clinical standards and clinical relevance: To better assess the clinical relevance of the breast age gap, we have benchmarked its predictive performance against the clinical standard, breast density, a widely accepted imaging-based risk factor and a criterion used in screening practice (e.g., Netherlands Early Breast Neoplasm Screening (DENISE) trial¹³). When matching the number of high-risk women defined by the DENISE trial criteria, Mammo-AGE consistently identified more breast cancer cases than breast density across 1-, 5-, and 10-year follow-up periods (Figure R2.3.1). Specifically, Mammo-AGE identified approximately 30% more cancers than the DENISE trial criteria in the EMBED screening dataset at 1-year follow-up (36 vs. 27 cancers).

Odds ratio (OR) analyses further showed that each one-year increase in the breast age gap was associated with a 7.0%–11.4% increase in future breast cancer risk (Table R2.3.1), supporting the breast age gap as a clinically meaningful and temporally robust biomarker for risk stratification.

Figure R2.3.1 Venn diagrams comparing true positive cancer detection between the Mammo-AGE model (orange) and women with extremely dense breasts (purple), based on the selection criteria used in the DENSE trial. Red areas represent cancers diagnosed within 1-, 5-, or 10-year follow-up periods in the inhouse and EMBED datasets. Under equivalent high-risk selection thresholds (i.e., selecting the same number of women), the Mammo-AGE model identifies a greater number of cancers.

	Inhouse 10-year risk		EMBED 5-year risk	
	Odds ratios	P-value	Odds ratios	P-value
Breast density	1.060 (1.018-1.104)	0.005	1.069 (1.005-1.136)	0.03
Breast age gap	1.070 (1.028-1.114)	<0.001	1.114 (1.048-1.183)	<0.001

Table R2.3.1 Odds ratios results for breast cancer risk prediction based on breast age gap and breast density.

Translational potential beyond biological aging: To further evaluate its practical utility, we have applied Mammo-AGE directly to downstream tasks such as breast cancer risk prediction and compared its performance with several state-of-the-art (SOTA) models. As shown in Figure R2.3.2, our model consistently outperformed SOTA approaches in both short-term (2-year) and long-term (5-year) prediction tasks. These results demonstrate that breast age representations encode clinically relevant features that improve risk model performance and generalizability.

A. Risk prediction

Figure R2.3.2 Comparison results of short-term (2-year) and long-term (5-year) BC risk prediction with baseline model, SOTA risk model⁶, and SOTA foundational model⁷. Baseline: trained the model for risk prediction without age prediction pretrained weight; Ours: we finetuned the model for risk prediction with age prediction pretrained weight. SOTA risk model: MIRAI risk prediction model. SOTA SSL model: We finetuned the Mammo-clip model, which was pretrained on mammograms and radiology reports.

Action Taken:

- We have added Fig. S11 for the Venn diagram comparison between breast age gap and breast density across different follow-up periods.
- Page 19, Lines 404-410: “To assess the clinical relevance of the breast age gap compared to current standards, we benchmarked its predictive performance against breast density, a widely accepted imaging-based risk factor used in clinical practice (e.g., Netherlands Early Breast Neoplasm Screening (DENISE) trial¹³). Under matched high-risk population sizes based on the DENISE trial criteria (i.e., to select the same number of high-risk women), Mammo-AGE consistently identified more breast cancers than breast density across 1-, 5-, and 10-year follow-up periods (Fig. S11). Notably, Mammo-AGE identified approximately 30% more cancers than the DENISE trial criteria in the EMBED screening dataset at 1-year follow-up.”
- We have added a discussion on page 23, lines 508-510: “Our routinely used imaging-based breast age gap demonstrates similar effect sizes performance to those of previously published clinical biomarkers based on DNAm clock studies¹² for breast cancer risk stratification.”

Comment 4: In the development of Mammo-AGE are there some areas or regions of the breast that are more informative in this measure than others? Are there any other things that can be gleaned about the understanding of breast aging that is extracted from the mammograms to develop Mammo-AGE?

Regarding informative areas: The reviewer is very correct. Our analysis of saliency maps (Fig. 4), confirmed by expert breast radiologists in our team, shows that Mammo-AGE consistently focuses on specific anatomical regions rather than randomly scanning the entire breast. These include fibroglandular tissue, skin thickness, calcifications, masses, and vascular structures, which have been shown in previous breast morphology studies to be associated with age-related breast changes ¹⁴. For instance, with increasing age, the epidermis of the female breast continues to thin, the elasticity of the mammary gland decreases, and the mammary gland matrix undergoes ptosis as it is replaced by fatty tissue. Importantly, the model could learn consistent, inherent aging pattern features for each woman over time, as shown in Figure R2.5.1. These suggest that the model is consistently capturing biologically meaningful patterns from these informative areas.

Figure R2.5.1 Saliency maps based on longitudinal mammogram examinations for each woman.

Regarding other aging indicators: The most evident feature, also widely recognized in clinical practice, is breast density, which naturally decreases with age ¹⁵. Ablation studies (Table S5) show that incorporating breast density information improves age prediction accuracy, reducing ~0.1 years of mean absolute error (MAE). Notably,

Mammo-AGE does not require breast density labels as input, but can infer density directly from the images and leverage it to refine age predictions, further supporting its relevance in assessing breast tissue aging.

Action taken: We have added Fig. S8 for the longitudinal saliency maps analysis, along with the corresponding description on page 11, lines 260–261: “Importantly, as shown in Fig. S8, the model could learn the consistent inherent aging pattern features for each woman over time.”; and on page 22, lines 479–484: “Saliency map analysis (Fig. 4) indicates that Mammo-AGE predominantly focuses on biologically aging relevant breast structures, such as fibroglandular tissue, calcifications, blood vessels, skin thickness, and adipose tissue, which are known to change with age¹⁵. The longitudinal heatmap analysis (Fig. S8) suggests that the model is consistently capturing biologically meaningful patterns from these informative areas. Additionally, our results (Table S5) confirm that breast density contributes to improved age prediction accuracy.”

Comment 5: It is surprising that the only covariates that were adjusted for are chronological age and breast density. This is particularly the case as there are other risk factors and confounders for breast cancer. How do these factors affect the prediction/association?

This is a valuable point. We agree that adjusting for additional covariates provides a more comprehensive assessment of the breast age gap’s association with breast cancer.

BreastAGE Years, mean±SD	N	Events	Inc.	Model 1 (Unadj.)		Model 2 (CA-ACR-adj.)		Model 3 (RF-adj.)		
				HR (95%CI)	P value	HR (95%CI)	P value	HR (95%CI)	P value	
Inhouse BC events (10Ys)	0.51±4.55	10,392	3,656	35.2	-	-	-	-	-	
Age gap, per one age (Ys)	-	-	-	-	1.013 (1.006-1.020)	< 0.001	1.016 (1.009-1.023)	< 0.001	1.014 (1.007-1.022)	< 0.001
Low-risk group (Gap < 0.36)	-3.08±2.70	5,196	1,722	33.1	Reference	-	Reference	-	Reference	-
High-risk group (Gap > 0.36)	4.10±2.88	5,196	1,934	37.2	1.164 (1.091-1.242)	< 0.001	1.157 (1.084-1.235)	< 0.001	1.140 (1.067-1.217)	< 0.001
EMBED BC events (5Ys)	0.08±5.025	58,983	1,029	3.49	-	-	-	-	-	
Age gap, per one age (Ys)	-	-	-	-	1.022 (1.001-1.034)	< 0.001	1.020 (1.008-1.033)	0.002	1.020 (1.008-1.033)	0.002
Low-risk group (Gap < -0.01)	-3.75±2.85	29,492	463	3.14	Reference	-	Reference	-	Reference	-
High-risk group (Gap > -0.01)	3.88±3.06	29,491	566	3.84	1.225 (1.083-1.385)	< 0.001	1.152 (1.071-1.305)	0.026	1.151 (1.016-1.304)	0.027

Inc = incidence per 1,000 person-years; CI = confidence interval; BC = breast cancer; HR = hazard ratio; Unadj. HR = unadjusted HR; CA-ACR-adj. HR = HR adjusted HR on chronological age (CA) and breast density (ACR); RF-adj. HR (Inhouse) = HR adjusted HR on more risk factors for inhouse dataset, including CA, ACR, Race, Gene mutation, Menarche, Menopausal status, and Manufacturer of mammograms; RF-adj. HR (EMBED) = HR adjusted HR on more risk factors for EMBED dataset, including CA, ACR, Ethnicity, and Manufacturer of mammograms;
BreastAGE = deep learning-based breast biological age.

Table R2.5.1 Association between breast age gap and future breast cancer using Cox proportional hazards regression models in inhouse and EMBED datasets.

Adjusted model on age and density: However, age adjustment is standard practice in aging-related biomarker studies and has been used in landmark aging clock models (Levine et al., *Aging* 2018; Lu et al., *Aging* 2019; Belsky et al., *eLife* 2022)^{16–18}. Given that breast density changes with age and is an independent risk factor for breast cancer, we conservatively adjusted for both chronological age and breast density in our analysis.

Adjusted model on risk factors: To address the reviewer’s concern, we have conducted further analysis incorporating additional covariates where available (including chronological age, breast density, race, gene mutation, menarche, menopausal status, and manufacturer of mammogram). As shown in Table R2.5.1 (**Model 3**),

adjusting for these additional risk factors had minimal impact on the hazard ratio across different datasets, confirming that the association between breast age gap and breast cancer remains stable and independent of these covariates.

Action taken:

- We have updated Table 3, reporting Cox model results with additional adjustments.
- Page 19, Lines 397–399: *“Further adjusting for additional breast cancer risk factors (Table 3, Model 3) did not substantially alter the association between breast age gap and cancer risk, supporting its robustness and independence.”*

Comment 6: When reading the manuscript, it is not clear how much other work related to breast aging has been done, other than to say that people with breast cancer have increase biological age in DNAm studies. The manuscript should elaborate more on this. If it is limited to just that, it should be emphasized as more of a point of novelty. With the increase in DNAm organ specific aging clocks, are there other metrics of breast age that that this can be compared with?

We appreciate the reviewer’s suggestion and agree that further elaborating on prior works in breast aging will help highlight the novelty of our approach.

To date, most studies on breast aging have relied on DNAm-based clocks, such as the Horvath and Levine epigenetic age models^{12,19–26}. These studies demonstrated that epigenetic age acceleration in breast tissue correlates with increased breast cancer risk^{12,19,23–26}. More recently, single-cell and spatial transcriptomic studies (e.g., Angarola et al., *Nature Aging*, 2024; Sayaman et al., *eLife*, 2024)^{27,28} have uncovered age-related transcriptomic reprogramming in normal breast tissue, including lineage fidelity loss, chromatin remodeling, and immune changes.

While these molecular studies have advanced our understanding of breast aging, they require invasive tissue or blood samples and have limited applicability for large-scale, population-level breast cancer screening. In contrast, our work is the first to model breast aging non-invasively using mammographic data at scale, enabling biological aging estimation within a routine screening setting. This practical imaging-based approach identifies individuals with accelerated breast aging, offering a clinically scalable alternative.

Although there are currently no paired datasets for direct comparison between DNAm-based and imaging-derived breast aging, our results show that the mammogram-based breast age gap behaves analogously to DNAm age acceleration²⁹, as demonstrated by breast age gap analysis in Figure 6. Furthermore, our publicly available code supports future development by enabling integration of other predictive assays, which could help establish unified, multi-scale biomarkers of breast aging.

Action taken:

- We have added the corresponding text in the Discussion section on page 23, lines 505-508: *“Existing studies on breast aging primarily rely on DNAm-based clocks^{12,19–26}, which estimate biological age using epigenetic markers but require tissue or blood samples. These methods are limited in large-scale screening applications.*

In contrast, our approach provides a non-invasive practical imaging-based alternative for estimating breast age and assessing its relationship with breast health.”;

- And on page 24, lines 540–542: *“With the growing development of DNAm-based organ-specific aging clocks, future work should explore the comparison and integration of different predictive aging biomarkers to enhance clinical applications.”*

Comment 7: Are there other studies on the physiology of the breast and how it presents in a mammogram? Is there any way to assess if Mammo-AGE captures any important physiology of breast tissue changes leading to cancer? If so, what are they?

We appreciate the reviewer’s valuable suggestion and have expanded our analysis accordingly. In addition to examining model attention in healthy women, we newly analyzed saliency maps from patients who were later diagnosed with breast cancer to assess whether Mammo-AGE captures physiologically meaningful features related to disease.

As shown in the updated saliency analysis (Figure R2.7.1), the model consistently focuses on regions associated with breast tissue composition and aging. These regions are known to reflect physiological processes such as glandular involution, tissue remodeling, and local vascular changes.

Figure R2.7.1 Longitudinal saliency map analysis of the breast age prediction model. Red areas indicate regions important for accurate age estimation. Blue areas indicate abnormal regions that result in higher prediction deviations. Longitudinal analysis shows that the blue area is biologically abnormal and has a high risk of developing breast cancer in the future.

Notably, in longitudinal images of patients who later developed breast cancer, we observed that the model detected atypical patterns, such as focal fibroglandular areas and structural asymmetry between views. These areas correspond to deviations from expected aging trajectories, reflected by increased estimation discrepancies (visualized in blue regions). Importantly, since Mammo-AGE was trained only on healthy populations to accurately

model natural biological aging, the presence of atypical or pathological tissue leads to deviations from expected age prediction rather than model failure.

These results indicate that Mammo-AGE not only learns from global age-related trends but may also be sensitive to local physiological variations that precede or accompany abnormal aging processes with malignant transformation risk.

Action taken: we have added the following sentence to the Results section on page 11, lines 262–267: *“To accurately model natural biological breast tissue aging changes, we trained the model only on the healthy population. This means that unseen abnormal tissue may lead to higher deviations in age estimation for the model. We further analyzed saliency maps from time-point mammograms of patients who later developed breast cancer. We observed high-deviation areas (blue color) in focal fibroglandular regions and structural asymmetry between views, suggesting an abnormal aging pattern associated with breast cancer risk.”*

Comment 8: There is still a lack of general cohort characteristics presented that provides any additional details on the cohorts, like some of the basic demographics (age, BMI, race, ethnicity, education, comorbidities, smoking).

Thanks for this valuable point. We agree that providing additional demographic details enhances the transparency and interpretability of our study. We have now expanded the cohort characteristics by including key variables such as race and imaging manufacturer details (Table R2.8.1).

However, as our collected mammogram datasets originate from large-scale breast screening programs, some clinical and lifestyle factors, such as BMI, smoking status, and comorbidities, are not routinely collected due to privacy and data availability constraints.

Action taken:

- We have updated Table 1 to include the additional demographic characteristics collected.
- We have added the following statement to the Discussion section (page 23, lines 538–540): *“Some lifestyle and clinical factors such as BMI, smoking status, and comorbidities, were not available due to data privacy constraints and limited availability. Future studies could explore their impact on breast tissue aging.”*

Dataset	Internal datasets			External datasets	
	Inhouse	RSNA	VinDr	EMBED	CMMMD
Total n exams (patients)	18,828 (5,048)	11,905 (11,905)	4,335 (4,335)	58,983 (21,434)	1,775 (1,775)
Country	Netherlands	Australia and America	Vietnam	America	China
Age					
Mean (SD); years	51.97 (11.67)	58.64 (9.88)	45.10 (9.77)	58.95 (11.94)	47.56 (10.81)
Minimum-maximum	23-89	26-89	18-88	19-89	18-87
Breast Density					
ACR A					
Total n exams	1,187 (6.3%)	553 (4.6%)	22 (0.5%)	6144 (10.4%)	NA
Age, mean (SD); years	56.37 (10.87)	60.34 (10.61)	59.59 (9.80)	62.63 (11.17)	NA
ACR B					
Total n exams	10,843 (57.6%)	2,509 (21.1%)	381 (8.8%)	24,603 (41.7%)	NA
Age, mean (SD); years	53.83 (11.50)	59.98 (11.06)	55.66 (9.86)	61.19 (11.41)	NA
ACR C					
Total n exams	4,919 (26.1%)	2,432 (20.4%)	3,319 (76.6%)	24,663 (41.8%)	NA
Age, mean (SD); years	48.99 (11.14)	54.80 (10.99)	44.94 (8.87)	56.83 (11.79)	NA
ACR D					
Total n exams	1,857 (9.9%)	307 (2.6%)	613 (14.1%)	3,217 (5.5%)	NA
Age, mean (SD); years	46.02 (10.48)	50.41 (10.14)	38.90 (8.35)	51.04 (11.44)	NA
Unknown					
Total n exams	22 (0.1%)	6,104 (51.3%)	NA	238 (0.4%)	1,775 (100.0%)
Age, mean (SD); years	64.14 (13.18)	59.87 (8.06)	NA	57.68 (13.86)	47.56 (10.81)
Breast cancer					
Negative					
Total n exams	18,213 (96.7%)	11,419 (95.9%)	4,250 (98.0%)	58,430 (99.1%)	465 (26.2%)
Age, mean (SD); years	51.78 (11.61)	58.43 (9.83)	44.86 (9.57)	58.91 (11.93)	42.36 (9.46)
Postive					
Total n exams	615 (3.3%)	486 (4.1%)	85 (2.0%)	553 (0.9%)	1,310 (73.8%)
Age, mean (SD); years	57.61 (11.97)	63.49 (9.87)	57.06 (11.89)	62.31(13.31)	49.41 (10.66)
Race					
White					
Total n exams	6146 (32.6%)	NA	-	25340 (43.0%)	-
Age, mean (SD); years	48.92 (10.90)	NA	-	61.15 (12.00)	-
African					
Total n exams	55 (0.3%)	NA	-	25899 (43.9%)	-
Age, mean (SD); years	52.13 (14.78)	NA	-	58.43 (11.63)	-
Asian					
Total n exams	108 (0.6%)	NA	4,335 (100.0%)	3295 (5.6%)	1,775 (100.0%)
Age, mean (SD); years	48.49 (10.57)	NA	45.10 (9.77)	54.37 (10.99)	47.56 (10.81)
Other or Unknown					
Total n exams	12519 (66.5%)	NA	-	4449 (7.5%)	-
Age, mean (SD); years	53.50 (11.73)	NA	-	52.76 (10.56)	-
Manufacturer information					
	Selenia Dimensions	NA	Mammomat Inspiration	Lorad Selenia	NA
	Lorad Selenia		GIOTTO CLASS	Selenia Dimensions	
	Hologic Selenia		GIOTTO IMAGE 3DL	Senograph 2000D ADS_17.4.5	
			Mammomat Inspiration	Senograph 2000D ADS_17.5	
			Planmed Nuance	Senographe Essential VERSION ADS_53.40	
				Senographe Pristina	

Table R2.8.1 Characteristics of the different datasets. ACR: the American College of Radiology (ACR) BI-RADS breast density categories. The density grades are assessed by radiologists during clinical interpretation: ACR A (mostly fatty), ACR B (scattered fibroglandular), ACR C (heterogeneously dense), and ACR D (extremely dense).

Minor:

Comment 9: Line 53 There is not just one “epigenetic clock” Please correct as per the literature (look in the Aging Biomarker Consortium MS in Nature Medicine).

Thanks for pointing this out. We agree with the reviewer and have corrected the statement to reflect that there are multiple “epigenetic clocks,” each tailored to different tissues and purposes on page 2, line 51: *“Previous research has developed multiple epigenetic clocks,^{4,30} which correlate with chronological age across various tissues and organs, including breast tissue.”*

Comment 10: Line 180 says “detailes”

Thanks for pointing out the typo. We have carefully reviewed the manuscript to correct this and any other spelling errors.

Comment 11: In the discussion, there is a comment about the inclusion of cohorts around the world. It would be important to highlight the diversity and robustness of findings across these populations.

Thanks for this suggestion. Moreover, we have also added a subgroup analysis by race (Figure S4), confirming the robustness of age prediction by Mammo-AGE across diverse populations.

Action taken: We have added the discussion to highlight robustness of our Mammo-AGE model on page 21, lines 458-460: *“The Mammo-AGE model exhibited outstanding performance, with MAEs ranging from 4.2 to 6.1 years, validated on various cohorts from populations around the world, surpassing existing research^{31,32} in breast age prediction. Further subgroup analyses show the robustness of our model across racial groups (Fig. S4).”*

Reviewer #3 (machine learning model):

In this study, the authors develop a deep learning model using mammograms as input to predict the age of a patient. The model seems to produce reasonably good result. What's more interesting is, they found that the difference between the predicted and chronological age may serve as a risk factor for breast cancer. The study seems to be carefully conducted with nice results. However, close inspection by the reviewer revealed the findings to be questionable. I'm also suspicious if the study is properly motivated.

We thank the referee for the detailed and thoughtful evaluation of our manuscript. We especially appreciate the recognition of our model's performance and the potential clinical value of the breast age concept. Inspired by the referee's feedback and valuable comments, we have added additional analyses and downstream validations, which further demonstrate the rationale of this work and the benefits of utilizing breast age gap as a specific biomarker for breast health assessment, as well as its broader applicability to clinical tasks such as risk prediction and diagnostic classification. We have responded to your constructive comments below.

Major comments:

Comment 1: What really is the goal of the study? Why do we even need to predict the age of breasts? If the goal is cancer risk prediction, then why not predict cancer directly? It seems the entire study is motivated by Ref. 11-12. But copying other people's homework is not considered as innovation.

We appreciate the reviewer's thoughtful feedback. We acknowledge the importance of clarifying the motivation-goal and the distinction between our study and breast cancer risk prediction models.

Goals: The goal of this study is to investigate whether the deep-learning model could capture the biological breast age reflected by breast tissue from mammograms. According to the learned biological age from images, we further aim to explore any differences in tissue aging that can be associated with current breast cancer status and future breast cancer risk, which have already been observed at the molecular or protein level in DNAm-based studies ^{12,19,23–26}.

Why this matters: Biological aging is well-established as an important determinant of cancer susceptibility ⁴ (Oh et al., Nat, 2023). Prior research ^{12,19–28}, primarily molecular studies using DNA methylation-based aging clocks, has consistently shown accelerated biological aging in breast tissue from cancer patients. However, DNAm methods require invasive sampling, limiting their utility for large-scale population screening or longitudinal studies. Despite millions of mammograms performed globally each year, there currently exists no imaging-based biomarker to measure breast tissue aging at the population level.

Our proposed "breast age gap" provides a practical, non-invasive alternative for capturing subtle yet biologically meaningful differences in breast tissue aging. Rather than directly predicting cancer presence (a binary outcome well-addressed by existing models like MIRAI ⁶ or other classifications), which function as black-box cancer risk predictors and offer limited insight into why someone is at higher risk, the breast age gap precisely quantifies deviations in tissue aging trajectories, revealing continuous and subclinical changes that may precede malignancy (Figure 6 and Figure S7, in the manuscript). Our findings also show how an exact elevated breast age gap is

associated with breast cancer risk (Figure 6, in the manuscript), which is independent of chronological age and breast density. Furthermore, technically, this refined learning could significantly enhance the performance of modeling for downstream tasks related to breast cancer, as evidenced by the experiments (Figure R3.2.2, see Comment 2), highlighting its potential for personalized risk assessment and earlier preventive interventions.

Novelty: As Reviewer #1 noted, the key innovation of our study is that it introduces an AI-driven, non-invasive, and scalable approach to breast age estimation, distinct from DNAm-based methods. To our knowledge, this study is the first systematic assessment of breast aging and its link to breast health using mammograms, offering a new perspective on imaging-derived metrics for breast research. Technically, unlike the referred studies^{33,34}, we enhance age estimation through a novel instance-bag transformer module for multi-view integration and introduce probabilistic ordinal embedding (POE) and mean-variance loss (MVL) to model the ordinal nature of biological aging. Moreover, we further explore the clinical utility of our proposed Mammo-AGE in broader downstream applications, including breast cancer diagnosis and risk prediction, benchmarking it against state-of-the-art models.

Action taken: We have highlighted the motivation and the significance of breast age estimation in the revised manuscript on page 3, lines 83-89: *“Unlike direct breast cancer prediction models⁶ using mammograms, the goal of this study is to estimate breast age as a specific and continuous biomarker of cumulative tissue aging, offering a non-invasive and scalable alternative to DNA methylation-based molecular aging clocks¹². Importantly, it provides new and practical insights into breast health status, precisely revealing the association between elevated breast age gap and breast cancer risk. Further, we investigated the benefits of this new biomarker in various situations, including as an aging biomarker analysis and for breast disease-related downstream tasks.”*

Comment 2: A proper baseline should be a model like Mirai that predicts breast cancer risk and age at the same time, and is trained on a large mammography dataset. If your model cannot beat Mirai, why bother developing your own?

We appreciate the reviewer’s insightful suggestion. Although the two models serve different purposes, we agree that benchmarking against MIRAI provides valuable context.

Age prediction: MIRAI aims to predict breast cancer risk, with age prediction as an auxiliary task. We have implemented comparison experiments across multiple datasets (Figure R3.2.1). Mammo-AGE consistently outperformed MIRAI, achieving a lower MAE (e.g., 4.152 vs. 8.345 in the inhouse dataset), as well as higher correlation and accuracy, demonstrating its superior ability to capture biologically relevant breast aging features. Unlike MIRAI, which predicts age in coarse bins (six age groups: 40-100 years), Mammo-AGE estimates continuous biological age, allowing for a more precise and clinically meaningful assessment of breast tissue aging.

Figure R3.2.1 Comparison results of age prediction with other comparative model (MIRAI)⁶.

A. Risk prediction

Figure R3.2.2 Comparison results of short-term (2-year) and long-term (5-year) BC risk prediction with baseline model SOTA risk model⁶ and SOTA self-supervised learning (SSL) model⁷. Baseline: trained the model for risk

prediction without age prediction pretrained weight; Ours: we applied the model for risk prediction task with age prediction pretrained weight. SOTA risk model: MIRAI risk prediction model. SOTA SSL model: We applied the Mammo-clip model, which was pretrained with mammograms and radiology reports. AUROC: Area Under Receiver Operating Characteristic Curve; AUPRC: Area Under Precision-Recall Curve.

Risk prediction: Although Mammo-AGE was not initially designed for risk prediction, to compare the risk prediction directly, we applied it to this task and compared it with MIRAI on the inhouse and EMBED datasets (Figure R3.2.2). Mammo-AGE (age-pretrained and risk finetuned) outperformed MIRAI across all metrics (AUPRC: 0.40 vs. 0.36; AUROC: 0.75 vs. 0.71 in the inhouse dataset, for long-term risk prediction). Notably, to further evaluate the generalizability, we involved a new public dataset, CSAW-CC¹⁰, for direct external validation. The stable performance advantage of our model further evidences its generalizability.

Broader clinical potential: Motivated by reviewer's suggestion, we further applied Mammo-AGE for diagnostic analysis among six datasets (Figure R3.2.3). The results demonstrate that by pre-learning biological aging features from images, our Mammo-AGE model enhances downstream breast cancer diagnosis performance and generalizability across datasets.

B. Diagnosis

Figure R3.2.3 Comparison results of diagnostic prediction. The “*” represents that the model is finetuned on inhouse and EMBED datasets and then tested directly on all others. Baseline: trained the model for the downstream task without age prediction pretrained weight; Ours: we applied the model for the downstream task with age prediction pretrained weight. Risk: MIRAI risk prediction model. SSL: We applied the Mammo-clip model, which was pretrained with mammograms and radiology reports.

Action Taken: We have revised the manuscript as follows:

- We have clarified the difference in model purpose on page 22, lines 473–475: *"Unlike these risk models, which are designed for direct risk prediction, Mammo-AGE models biological breast aging as an image-based biomarker, while highlighting the broader clinical utility of breast tissue aging."*
- We have added the description of the age prediction comparison results on page 9, lines 211-215.
- We have highlighted the broader clinical applicability of Mammo-AGE on page 19, lines 412-443.

Comment 3: Although the age gap seems to be a risk factor for cancer, is it possible that that is simply because the cancer group is older than the healthy group (see Table 1)? This can be mitigated by matching the age of the healthy group with that of the cancer group or use age as a covariate in your Cox regression analysis.

We understand the reviewer's concern regarding the potential confounding effect of age between the cancer and healthy group.

Age gap as an independent marker: We have revised the manuscript to clarify that the age gap has been appropriately adjusted by chronological age ^{35,36}. This is to ensure that the breast age gap is assessed as an independent marker of biological status, rather than being influenced by chronological age differences.

Adjusted age as a covariate: We had initially recognized and indeed included age as a covariate to account for its potential confounding effects in the Cox regression analysis (as shown in Table 3 of the manuscript). The hazard ratio results are based on the model adjusted by age and breast density. These steps confirm that our findings are not confounded by age.

Action taken: We have highlighted the covariate adjustments in the manuscript on page 18, lines 387–391: *"After adjusting for confounding factors such as chronological age and breast density, each 1-year increase in breast age gap was associated with 1.6% and 2.0% increase in breast cancer risk on the inhouse and EMBED datasets, respectively (inhouse dataset: HR = 1.016 [95% CI: 1.009-1.023], P < 0.001; EMBED dataset: HR = 1.020 [95% CI: 1.008-1.033], P = 0.002; Table 3, Model 2) "*

Comment 4: The authors spent a lot of effort tinkering with different machine learning techniques like GLT, MVL, and POE but they all led to marginal improvements. The only change that led to significant improvement is larger image size. Maybe the authors should stop wasting time and just buy more GPUs?

We appreciate the reviewer's perspective. While increasing image size (from 256×128 to 1024×512) did contribute to performance gains to some extent, our exploration of different machine learning techniques was driven by the need to balance efficiency, generalizability, and computational feasibility. As shown in Table R3.4.1, incorporating these techniques led to a statistically significant improvement (MAE reduction from 5.079 to 4.728, p<0.001), demonstrating that architectural refinements meaningfully enhance model learning efficiency and performance. Simply scaling up image resolution may not always be a practical or scalable solution, due to the higher risk of overfitting and the complexity of the optimization problem ³⁷. In contrast, according to the characteristics of the

target task (age prediction), deep learning models need to extract age-related features from both global structures and fine-grained textures ³⁸.

Model	MAE	Pearson Correlation (r)	CS % ($\alpha=5$ year)
Baseline (w/o GLT, MVL, POE)	5.079±0.035	0.829±0.004	63.7%±0.2%
Mammo-AGE	4.728±0.034	0.855±0.003	66.5%±0.4%
P value	3.92E-07	4.22E-06	6.42E-07

Table R3.4.1 Comparison of model performances with baseline method and Mammo-AGE (ResNet-18) using the same image size (**1024×512**).

Comparison experiment: To further evaluate the impact of image resolution, we followed the reviewer’s suggestion to use more GPUs. We reviewed mammogram-related studies ^{39–42} and the RSNA Screening Mammography challenge, finding that commonly used resolutions range from 224 to 2048 pixels. We conducted additional experiments using larger image sizes (1536×768 and 2048×1024) and identified an optimal resolution of 1536×768 (MAE = 4.724) within our current computational constraints (NVIDIA A100). Notably, performance declined at 2048×1024, indicating that resolution alone is not the sole determinant of model accuracy, and that well-designed learning strategies remain crucial (Table R3.4.2).

Image Size	MAE	Pearson Correlation (r)	CS % ($\alpha=5$ year)
256×128	5.403±0.028	0.803±0.003	60.9%±0.5%
512×256	5.019±0.035	0.836±0.002	64.1%±0.2%
1024×512	4.728±0.034	0.855±0.003	66.5%±0.4%
1536×768	4.668±0.053	0.864±0.003	66.9%±0.6%
2048×1024	5.202±0.123	0.832±0.006	61.8%±1.0%

Table R3.4.2 Comparison of Mammo-AGE model (**ResNet-18**) performances based on different image sizes.

Action taken: We have updated the manuscript with refined model results using larger image sizes and integrated predicted breast density following the reviewer’s suggestion (as per Comment 11). Figures 1–6 have been updated accordingly. Importantly, the main findings of the study remain consistent with our previous results, reinforcing the robustness of our findings.

Comment 5: The model didn’t do as well on the VinDr and CMMD data. Is it because of the younger population or the race/ethnicity? This is worth clarifying.

The reviewer is correct. Indeed, the VinDr and CMMD datasets primarily represent Asian populations and younger age groups.

Dataset	MAE	Pearson Correlation (r)	CS % ($\alpha=5$ year)
Inhsoue	4.090±0.047	0.892±0.002	73.0%±0.8%
RSNA	4.548±0.022	0.844±0.001	68.2%±0.7%
VinDr	4.075±0.010	0.847±0.002	73.8%±0.8%
CMMD	6.103±0.222	0.705±0.009	54.8%±1.5%
EMBED	5.010±0.040	0.855±0.002	63.4%±0.5%

Table R3.5.1 Model performances of age prediction across different datasets.

Our model maintains a comparable MAE and CS score on VinDr, indicating consistent predictive accuracy, as shown in *Table R3.5.1*. To further investigate the model's generalizability, we conducted additional subgroup analyses based on race/ethnicity (see Comment 13; Fig. R3.13), which demonstrate the model's robustness across diverse populations.

The CMMD dataset consists primarily of unilateral mammograms, providing only CC and MLO views of a single breast. This limited imaging perspective likely contributes to the model's reduced accuracy, as bilateral imaging may provide additional anatomical context beneficial for age estimation. This can be further appreciated by Figure 3 in the manuscript, which shows improved performance for the multi-view based model compared to that of the single-view based model.

Action taken: We have updated the manuscript to highlight the model's robustness across various racial and ethnic groups (see page 21, line 462): "*Further subgroup analysis shows the robustness of our model across different racial groups (Fig S4).*"

Comment 6: Where did the weight ratios (2,2,5,1,5) of different backbones come from? This raises the suspicion of overfitting by hyperparameters.

Thanks for raising this important point. We agree that inappropriate ensemble weight tuning can introduce overfitting. To avoid this, the ensemble weights in our study were determined automatically through a data-driven and validation-based approach in line with the study by Xiao et al ⁴³.

Specifically, the training-validation process followed the nested five-fold cross-validation strategy. **Test data remained independent for this process.** For each fold, all five backbone models were trained on the training subset and evaluated on the corresponding validation subset. The contribution of each backbone (e.g., ResNet18, ResNet50, ConvNext-Tiny, etc.) was determined based on its MAE performance on the validation set. These fold-specific weights were then averaged across all five folds to derive the final weight ratios used in the ensemble. No tuning was performed on the test set, ensuring strict independence.

This approach follows best practices commonly used in machine learning competitions (e.g., the RSNA Breast Cancer Detection Challenge ¹), where weighted ensembles are used to stabilize and improve predictions by leveraging complementary strengths of diverse architectures. By relying on systematic performance-derived weights and avoiding manual hyperparameter tuning, we minimize the risk of overfitting while preserving model diversity.

Action taken: We have updated the manuscript to detail the weight ratio calculation process on page 27, lines 646–649: *"The weight ratios were derived using five-fold cross-validation by first calculating the weight ratios for each backbone model based on its performance (MAE) in each validation fold. These fold-specific weight ratios were then averaged across all five folds to obtain the final set of weight ratios. This ensures a balanced contribution from different architectures while maintaining diversity, reducing the risk of overfitting."*

Comment 7: The authors claim that "The saliency maps reveal that the model commonly focuses on features such as breast skin thickness, fibroglandular tissue, calcifications, masses, and breast vessels". To be honest, this isn't obvious to my eyeballs. The saliency maps serve more as eye candy than insightful tools to understand the model. Additionally, why not look at the attention maps as well?

We appreciate the reviewer's feedback and agree that clearer visualization is important for understanding model behavior. Saliency maps were used to quantify positive/negative contributions to accurate age prediction rather than merely indicating areas of focus (i.e., attention maps). These maps allow us to assess the influence of regions (e.g., fibroglandular tissue, skin thickness, and calcifications) on the model's output, providing a detailed breakdown of their importance (as shown in Figure R3.7.1). In future work, we will focus on quantitatively analyzing the specific contributions of different breast tissues to accurate age prediction.

Importantly, to accurately model natural biological changes in breast tissue aging, we trained the model only on healthy populations. This means that previously unseen abnormal tissue may lead to higher deviations in age estimation for the model. In time-point images of patients who later developed cancer, we observed high-deviation areas (blue color, vice versa for red color) in focal fibroglandular regions and structural asymmetry between views, suggesting an abnormal aging pattern related to breast cancer risk (as shown in Figure R3.7.2).

¹ <https://www.kaggle.com/competitions/rsna-breast-cancer-detection/overview>

Figure R3.7.1. Saliency maps based on longitudinal mammogram examinations.

Figure R3.7.2. Longitudinal saliency map analysis of the breast age prediction model. Red areas indicate regions important for accurate age estimation. Blue areas indicate abnormal regions that result in higher prediction deviations. Longitudinal analysis shows that the blue area is biologically abnormal and has a high risk of developing breast cancer in the future.

Comment 8: The scatter plots in Fig. 6 are not useful. I don't see any trend between the healthy and cancer groups.

Thanks for raising this valuable point. We agree that the scatter plots in Figure 6 do not illustrate a visible trend between the healthy and cancer groups. However, the purpose of these plots is to show that the breast age gap is not obviously linked to chronological age in either the healthy or cancer groups. These findings highlight that the observed accelerated biological aging in the cancer group is independent of their average chronological age.

Action taken: We have revised the caption of Fig. 6 to clarify this point and address the potential confusion on page 17, lines 335-336: *“The scatter plots show that after age bias correction, the breast age gap is independent of chronological age in both healthy and cancer groups.”*

Comment 9: The bias corrected age used chronological age as input, which is the target of your model. Was this strictly done on the train set, or on the test set (most likely)?

We appreciate the reviewer’s attention to this detail. The chronological age is required for bias correction, which was performed on the validation set (healthy population), following standard practice in aging-related study³⁶ (Lee, et al., Nature Aging, 2022). Below is detailed explanation.

Model target and breast age gap definition: Our model was trained using healthy mammograms, for which the chronological age accurately reflects the biological breast tissue aging status. The goal is to establish an aging trajectory for healthy breast tissue. Previous studies^{16–18} have shown that, after bias correction in healthy populations, the difference between the predicted breast age and chronological age (also referred to as ‘age gap’ or ‘ Δ age’) reflects individual variations in aging, with positive values (age acceleration) being associated with increased risk for various aging-related diseases, including cancer.

Bias correction approach: However, age prediction models commonly exhibit systematic bias, overestimating age in younger individuals and underestimating it in older ones. To ensure that the breast age gap accurately reflects accelerated aging rather than being confounded by chronological age, bias correction is necessary. For this analysis, we assume that the chronological ages are known. The bias correction models are fitted using the (healthy population) test data directly, which aligns with best practices in biological aging research³⁶. This approach ensures that the adjustment was independent of disease status. Further details on this correction process are provided in the next comment.

Action taken: We have clarified this methodology in the manuscript (see page 28, lines 695-697): *“Following standard aging study method³⁶, bias correction coefficients were obtained from the validation set (healthy population) to ensure that the breast age gap reflects true biological aging rather than systematic prediction bias, remaining independent of health status.”*

Comment 10: Line 572-583, the description of bias correction is poor. I’m sure the equations are incorrect.

Thanks for your valuable feedback. We agree that it is important to provide a clearer description of the bias correction method. The original equation was adapted from prior studies (Lee et al., Nature Aging 2022)³⁶, which

may have led to confusion due to simplified notation. To ensure accuracy, we have revisited the methodology and added the equations clarification accordingly.

Action taken: We have revised the text on pages 28, lines 687-691: “The presence of systematic bias, induced by regression dilution, introduces a correlation between the estimated age gap and chronological age^{35,36}. This leads to a tendency to overestimate age in younger individuals and underestimate it in older individuals^{35,36,44}. To mitigate this phenomenon, we applied the linear bias correction method as outlined by Smith et al.⁴⁵. The systematic bias is defined as: $\Delta_{bias} = \hat{y} - x$, where \hat{y} denotes the predicted breast age and x is the chronological age. To correct for bias, we fit a linear regression model: $\hat{\Delta}_{bias} = ax + b$, using the healthy reference population to estimate systematic bias Δ_{bias} . Here a , and b represent the slope and intercept derived from the healthy group. The bias-corrected estimated breast age is then computed as: $\hat{y}_{corrected} = \hat{y} - \hat{\Delta}_{bias} = \hat{y} - (ax + b)$ and the corrected breast age gap is: $\Delta_{breast\ age, corrected} = \hat{y}_{corrected} - x \dots$ ”

Comment 11: The model is multi-tasking to predict age and density in parallel but the results on density prediction are not mentioned. If the density is associated with age (line 100), why not concatenate the predicted density with the embedding layer to predict the age? That may enhance the accuracy.

It is a great point. We agree that incorporating predicted density may potentially improve model accuracy of age prediction. Following this advice, we slightly modified the model by concatenating the predicted density with the final embedding layer for age prediction. This adjustment indeed led to performance improvements, reducing MAE by ($\Delta=0.037$) our ensemble Mammo-AGE model (Table R3.11.1). The detailed results across different backbone-based models are provided in the Supplementary Table S5. Besides, the performance of density prediction on each dataset is shown in Table R3.11.2.

Model	MAE	Pearson Correlation (r)	CS % ($\alpha=5$ year)
Mammo-AGE (Ensemble)	4.264±0.018	0.881±0.001	71.8%±0.5%
(w/o ACR)	4.301±0.030	0.881±0.002	71.4%±0.5%

Table R3.11.1 Ablation study on integrating predicted breast density into the embedding layer for age prediction, evaluated using five-fold cross-validation on the combined dataset. We compare the performance of Mammo-AGE models with and without incorporating predicted density across different backbones and a final ensemble model.

Dataset	AUC (Four classes)	ACC (Four classes)	AUC (Two classes)	ACC (Two classes)
Inhouse	0.790±0.005	0.677±0.005	0.877±0.007	0.811±0.003
RSNA	0.792±0.009	0.623±0.016	0.888±0.008	0.785±0.018
VinDr	0.780±0.017	0.799±0.015	0.949±0.007	0.938±0.008
EMBED	0.783±0.006	0.614±0.012	0.898±0.003	0.792±0.009

Table R3.11.2 Accuracy of density prediction on each dataset. The CMMD dataset was excluded due to the lack of breast density labels.

Action taken: We have updated the manuscript accordingly, including revised results in Figures 1–6. Importantly, the main findings of the study remain consistent with our previous results, reinforcing the robustness of our approach. We also added the density prediction results in the supplementary (Table S6).

Minor comments:

Comment 12: Methods like GLT, MVL, and POE are not standard and very recent. The authors should briefly describe these methods and provide some rationale why they were chosen. Fig. 1 looks confusing and doesn't help much in clarifying the methodology.

We agree with the reviewer that these methods are very recent. We have revised Fig. 1 to focus on introducing the overall design of our study and added a brief description of each part to make the methodology clearer to understand. The detailed description of the network architecture is provided in Figure R3.12.2 (Supplementary Figure S1).

Action taken: Page 3, Lines 99-110: *“By incorporating information from multi-view mammograms rather than using a single view, the model can enhance its ability to capture features related to breast aging. To achieve this, we draw inspiration from the newly proposed global-local transformer (GLT) framework³⁸ and introduce an instance-bag transformer. Further details of the deep learning model design are provided in the Methods section and Fig. S1. This transformer consists of self-attention and cross-attention blocks, facilitating the integration of information from multiple views. Regarding the constraint model, the mean squared error (MSE) loss function was leveraged to constrain the predicted age to match the ground truth age. In addition to constraining the predicted age directly, we also utilized probabilistic ordinal embedding (POE) loss⁴⁶ and mean-variance (MVL) loss⁴⁷ functions to guide the model in learning the reasonable ordinal distribution of the latent feature space and predicted probabilities to align with biologically meaningful aging trends.”*

A. Mammo-AGE Model

B. Internal and external datasets

C. Performance of age prediction

D. Breast age gap analysis after bias-correction

E. Mammo-AGE model for downstream tasks

Figure R3.12.1 Overall design of the study. (A) Schematic architecture of the proposed Mammo-AGE model and illustration of the occlusion analysis. The model utilizes four-view mammograms (CC and MLO views of both breasts) as input to predict breast age. An instance-bag transformer, inspired by the global-local transformer framework, integrates self-attention and cross-attention mechanisms to fuse information across views. The model incorporates multi-task learning for breast density prediction and uses a combination of cross-entropy (CE) loss, mean-variance (MVL) loss, and probabilistic ordinal embedding (POE) loss to optimize learning. The model outputs age predictions, with saliency maps highlighting age-relevant regions. Five different backbone based models (ResNet-18, ResNet-50, ConvNeXt-Tiny, EfficientNet-B0, and DenseNet-121) were ensembled by weighted averaging of predicted ages. The detailed description of the network architecture is provided in Supplementary Figure S1. (B) Age distributions of the internal and external datasets. The combined dataset was

split patient-wise for five-fold cross-validation. External validation was conducted on additional datasets. (C) Performance evaluation of the Mammo-AGE model on age prediction. (D) Association analysis between the breast age gap (predicted age minus chronological age) and breast cancer after bias correction. (E) Evaluation of the Mammo-AGE model on downstream clinical tasks, including diagnostic classification and risk prediction.

Mammo-AGE: Breast Age Prediction Model

Figure R3.12.2. Detailed description of the network architecture. The model is designed to learn from mammograms and predict the chronological age of women, termed “breast age”. By incorporating information from multi-view mammograms rather than using a single view, the model can enhance its ability to capture features related to breast aging. To achieve this, we draw inspiration from the global-local transformer (GLT) framework proposed by³⁸ and introduce an instance-bag transformer. This transformer consists of self-attention and cross-attention blocks, facilitating the integration of information from multiple views. Breast density, defined as the proportion of fibroglandular tissue to fatty tissue in the breast, typically decreases with increasing age². We also introduce multi-task learning for density prediction as auxiliary training. Regarding the constrained model, in addition to the mean squared error (MSE) loss function, we also utilized mean-variance (MVL) loss and probabilistic ordinal embedding (POE) loss⁴⁶ functions to constrain the model to learn the reasonable distributions of the latent feature space and predicted probabilities. Q, K, and V in the Instance-Bag Transformer refer to the query (Q), key (K), and value (V) matrices used in the attention mechanisms to capture relationships between instances and bags. Five different backbone-based models (ResNet-18, ResNet-50, ConvNeXt-Tiny, EfficientNet-B0, and DenseNet-121) were ensemble by weighted averaging of predicted ages.

Comment 13: Medical images are known to be biased by racial/ethnicity groups. Would like to see a breakdown based on that.

Thank you for your suggestions for the enhancement. We have expanded our subgroup analyses to include race/ethnicity (as shown in Figure R3.13.1). The results indicate that our age prediction model maintains strong, consistent performance across these diverse subgroups, further supporting its robustness and generalizability.

Figure R3.13.1. Our model's performance across different race subgroups in both inhouse and external EMBED datasets.

Reference

1. Pike, M. C., Krailo, M. D., Henderson, B. E., Casagrande, J. T. & Hoel, D. G. 'hormonal' risk factors, 'breast tissue age' and the age-incidence of breast cancer. *Nature* **303**, 767–770 (1983).
2. Checka, C. M., Chun, J. E., Schnabel, F. R., Lee, J. & Toth, H. The relationship of mammographic density and age: Implications for breast cancer screening. *American Journal of Roentgenology* **198**, W292–W295 (2012).
3. Wanders, J. O. P. *et al.* Volumetric breast density affects performance of digital screening mammography. *Breast Cancer Res Treat* **162**, 95–103 (2017).
4. Oh, H. S.-H. *et al.* Organ aging signatures in the plasma proteome track health and disease. *Nature* **624**, 164–172 (2023).
5. Dench, E. *et al.* Measurement challenge: Protocol for international case–control comparison of mammographic measures that predict breast cancer risk. *BMJ Open* **9**, e031041 (2019).
6. Adam Yala, Peter G Mikhael 1 , Fredrik Strand 2, 3 , Gigin Lin 4 , Kevin Smith 5, 6 , Yung-Liang Wan 4 , Leslie Lamb 7 , Kevin Hughes 8 , Constance Lehman† 7, R. B. Towards Robust Mammography-Based Models for Breast Cancer Risk. *SCIENCE TRANSLATIONAL MEDICINE* **13**, 1–51 (2021).
7. Ghosh, S., Poynton, C. B., Visweswaran, S. & Batmanghelich, K. Mammo-CLIP: A Vision Language Foundation Model to Enhance Data Efficiency and Robustness in Mammography. in *Medical Image Computing and Computer Assisted Intervention – MICCAI 2024* (eds. Linguraru, M. G. *et al.*) vol. 15012 632–642 (Springer Nature Switzerland, Cham, 2024).
8. Mutz, J., Iniesta, R. & Lewis, C. M. Metabolomic age (MileAge) predicts health and life span: A comparison of multiple machine learning algorithms. *Science ADvAnceS* (2024) doi:10.1126/sciadv.adp3743.
9. Li, R. *et al.* LensAge index as a deep learning-based biological age for self-monitoring the risks of age-related diseases and mortality. *Nat. Commun.* **14**, 7126 (2023).
10. Strand, F. CSAW-CC (mammografi) – ett dataset för AI-forskning för att förbättra screening, diagnostik och prognostik för bröstcancerCSAW-CC (mammography) – a dataset for AI research to improve screening, diagnostics and prognostics of breast cancer. 8.81 MiB, 19 variables, 8723 cases Karolinska Institutet <https://doi.org/10.5878/45VM-T798> (2022).
11. Bleyer, A. & Welch, H. G. Effect of Three Decades of Screening Mammography on Breast-Cancer Incidence. *N Engl J Med* **367**, 1998–2005 (2012).
12. Kresovich, J. K. *et al.* Methylation-based biological age and breast cancer risk. *J Natl Cancer Inst* **111**, 1051–1058 (2019).
13. Bakker, M. F. *et al.* Supplemental MRI screening for women with extremely dense breast tissue. *N. Engl. J. Med.* **381**, 2091–2102 (2019).

14. Lin, J. *et al.* Changes in the mammary gland during aging and its links with breast diseases. *Acta Biochim. Biophys. Sin.* **55**, 1001–1019 (2023).
15. Checka, C. M., Chun, J. E., Schnabel, F. R., Lee, J. & Toth, H. The relationship of mammographic density and age: Implications for breast cancer screening. *American Journal of Roentgenology* **198**, W292–W295 (2012).
16. Levine, M. E. *et al.* An epigenetic biomarker of aging for lifespan and healthspan. *Aging (Albany NY)* **10**, 573–591 (2018).
17. Lu, A. T. *et al.* DNA methylation GrimAge strongly predicts lifespan and healthspan. *Aging (Albany NY)* **11**, 303–327 (2019).
18. Belsky, D. W. *et al.* DunedinPACE, a DNA methylation biomarker of the pace of aging. *eLife* **11**, e73420 (2022).
19. Jung, S. Y., Yu, H., Deng, Y. & Pellegrini, M. DNA-methylation age and accelerated epigenetic aging in blood as a tumor marker for predicting breast cancer susceptibility. *Aging (Albany NY)* **16**, 13534–13562 (2024).
20. Chen, M. *et al.* DNA methylation-based biological age, genome-wide average DNA methylation, and conventional breast cancer risk factors. *Sci Rep* **9**, 15055 (2019).
21. Jain, N. *et al.* DNA methylation correlates of chronological age in diverse human tissue types. *Epigenetics & Chromatin* **17**, 25 (2024).
22. Ren, J.-T., Wang, M.-X., Su, Y., Tang, L.-Y. & Ren, Z.-F. Decelerated DNA methylation age predicts poor prognosis of breast cancer. *BMC Cancer* **18**, 989 (2018).
23. Wu, Y., Miller, M. E., Gilmore, H. L., Thompson, C. L. & Schumacher, F. R. Epigenetic aging differentially impacts breast cancer risk by self-reported race. *PLoS One* **19**, e0308174 (2024).
24. Mak, J. K. L. *et al.* Clinical biomarker-based biological aging and risk of cancer in the UK biobank. *Br J Cancer* **129**, 94–103 (2023).
25. Castle, J. R. *et al.* Estimating breast tissue-specific DNA methylation age using next-generation sequencing data. *Clinical Epigenetics* **12**, 45 (2020).
26. Kresovich, J. K. *et al.* Changes in methylation-based aging in women who do and do not develop breast cancer. *J Natl Cancer Inst* **115**, 1329–1336 (2023).
27. Sayaman, R. W. *et al.* Luminal epithelial cells integrate variable responses to aging into stereotypical changes that underlie breast cancer susceptibility. *eLife* **13**, e95720 (2024).
28. Angarola, B. L. *et al.* Comprehensive single-cell aging atlas of healthy mammary tissues reveals shared epigenomic and transcriptomic signatures of aging and cancer. *Nat Aging* **5**, 122–143 (2024).

29. Hofstatter, E. W. *et al.* Increased epigenetic age in normal breast tissue from luminal breast cancer patients. *Clinical Epigenetics* **10**, 112 (2018).
30. Argentieri, M. A. *et al.* Proteomic aging clock predicts mortality and risk of common age-related diseases in diverse populations. *Nat Med* **30**, 2450–2460 (2024).
31. Lekamlage, C. D., Afzal, F., Westerberg, E. & Cheddad, A. Mini-DDSM: Mammography-based Automatic Age Estimation. in *2020 3rd International Conference on Digital Medicine and Image Processing* 1–6 (Association for Computing Machinery, 2020). doi:10.1145/3441369.3441370.
32. De Lima Camillo, L. P., Lapierre, L. R. & Singh, R. A pan-tissue DNA-methylation epigenetic clock based on deep learning. *npj Aging* **8**, 4 (2022).
33. Nusinovici, S. *et al.* Retinal photograph-based deep learning predicts biological age, and stratifies morbidity and mortality risk. *Age Ageing* **51**, afac065 (2022).
34. Zhu, Z. *et al.* Retinal age gap as a predictive biomarker for mortality risk. *Br. J. Ophthalmol.* **107**, 547–554 (2023).
35. Zhang, B., Zhang, S., Feng, J. & Zhang, S. Age-level bias correction in brain age prediction. *NeuroImage: Clinical* **37**, (2023).
36. Lee, J. *et al.* Deep learning-based brain age prediction in normal aging and dementia. *Nature Aging* **2**, 412–424 (2022).
37. Sabottke, C. F. & Spieler, B. M. The effect of image resolution on deep learning in radiography. *Radiology: Artificial Intelligence* **2**, e190015 (2020).
38. He, S., Grant, P. E. & Ou, Y. Global-Local Transformer for Brain Age Estimation. *IEEE Trans. Med. Imag.* **41**, 213–224 (2021).
39. Wang, X. *et al.* Ordinal learning: Longitudinal attention alignment model for predicting time to future breast cancer events from mammograms. in *Medical Image Computing and Computer Assisted Intervention – MICCAI 2024* (eds. Linguraru, M. G. *et al.*) vol. 15001 155–165 (Springer Nature Switzerland, Cham, 2024).
40. Yeoh, H. H. *et al.* RADIFUSION: A multi-radiomics deep learning based breast cancer risk prediction model using sequential mammographic images with image attention and bilateral asymmetry refinement. *arXiv preprint arXiv:2304.00257* (2023).
41. Dembrower, K. *et al.* Comparison of a deep learning risk score and standard mammographic density score for breast cancer risk prediction. *Radiology* **294**, 265–272 (2020).
42. Liu, Y., Azizpour, H., Strand, F. & Smith, K. Decoupling Inherent Risk and Early Cancer Signs in Image-Based Breast Cancer Risk Models. *Lecture Notes in Computer Science (including subseries Lecture Notes in Artificial Intelligence and Lecture Notes in Bioinformatics)* **12266**, 230–240 (2020).

43. Xiao, Y., Wu, J., Lin, Z. & Zhao, X. A deep learning-based multi-model ensemble method for cancer prediction. *Computer Methods and Programs in Biomedicine* **153**, 1–9 (2018).
44. Peng, H., Gong, W., Beckmann, C. F., Vedaldi, A. & Smith, S. M. Accurate brain age prediction with lightweight deep neural networks. *Med. Image Anal.* **68**, 101871 (2021).
45. Smith, S. M., Vidaurre, D., Alfaro-Almagro, F., Nichols, T. E. & Miller, K. L. Estimation of brain age delta from brain imaging. *NeuroImage* **200**, 528–539 (2019).
46. Li, W., Huang, X., Lu, J., Feng, J. & Zhou, J. Learning probabilistic ordinal embeddings for uncertainty-aware regression. *Proceedings of the IEEE Computer Society Conference on Computer Vision and Pattern Recognition* 13891–13900 (2021) doi:10.1109/CVPR46437.2021.01368.
47. Pan, H., Han, H., Shan, S. & Chen, X. Mean-variance loss for deep age estimation from a face. in *Proceedings of the IEEE conference on computer vision and pattern recognition* 5285–5294 (2018).

REVIEWER COMMENTS

Reviewer #1:

The authors have adequately addressed my comments.

We appreciate the referee's recognition, and the effort dedicated to reviewing our manuscript and responses.

Reviewer #3:

The authors have addressed most of my comments in great details.

We sincerely thank the referee for the constructive feedback and for acknowledging the improvements made to our manuscript.

However, i found the model validation part - Figure R3.2.2 - to be concerning and would like more details:

* Assuming the "baseline" is a network training from scratch and "ours" is a network pretrained to predict breast age. is there any overlap between the data used for pretraining and the data used for finetuning? the authors should clarify which dataset is used for pretraining and which is used for finetuning.

We appreciate the reviewer's insightful feedback. The reviewer is correct about the definition of "baseline" and "ours".

Regarding the data usage, we acknowledge that in a total of six datasets, one dataset (i.e., Inhouse) was used for both pretraining and finetuning. Specifically, for the pretraining, the Inhouse, RSNA, and VinDr datasets were used (Table R3.1.1). For finetuning the downstream tasks (Fig. 7 in the manuscript; i.e., Figure R3.2.2), we used Inhouse and EMBED datasets, both of which contain follow-up information. Importantly, we show a consistent advantage of our method on both internal (RSNA, VinDr, EMBED, Inhouse) and entirely independent external (CMMMD, CSAW-CC) test sets for both downstream tasks (Fig. 7 in the manuscript).

Training stage	Tasks	Cross-validation (Train/Val) sets	Internal Test sets	External Test sets
Pretraining	Age prediction	RSNA, VinDr, Inhouse	RSNA, VinDr, Inhouse,	EMBED, CMMMD
Downstream finetuning	Risk, Diagnostic	EMBED, Inhouse	RSNA, VinDr, EMBED, Inhouse	CMMMD, CSAW-CC

Table R3.1.1 Dataset usage for the model pretraining and downstream tasks finetuning (risk prediction and diagnostic tasks). External test indicates that the datasets are used for test directly without any training/finetuning performed on them.

Action taken: We have added the definition of "baseline" and "ours" in the revised manuscript (page 21, lines 435-437): "Baseline: the model was trained for risk prediction from scratch without using age prediction pretrained weights; Ours: we finetuned the model with age prediction pretrained weights for risk prediction."

We have also detailed the dataset usage across training stages as suggested (page 21, lines 438-445): "Specifically, the Mammo-AGE model was pretrained to predict breast age using mammograms from the Inhouse, RSNA, and VinDr datasets. For the two downstream tasks, the models (baseline, ours, and SSL) were trained/finetuned using the Inhouse and EMBED datasets, which contain breast cancer outcomes. ...To ensure a fair comparison, the CSAW-CC and CMMD were used strictly as external test sets, with no training or fine-tuning performed on them for any model."

* Did you finetune Mirai on your internal datasets? * Did you finetune the networks on the external dataset (CSAW-CC)?

Given that MIRAI is a well-validated risk model trained on large and diverse cohorts (Yala et al., *Sci. Transl. Med.*, 2021; *J Clin Oncol.*, 2022)^{1,2}, we treated it as a benchmark reference. **Therefore, we evaluated the MIRAI without any additional finetuning to preserve its benchmark integrity.**

To ensure fair comparison and better reflect real-world usage, we further included another large-scale benchmark screening dataset (CSAW-CC^{3,4}) as an external validation cohort. **The entire CSAW-CC dataset was used strictly for testing purposes.** No training or finetuning was performed on CSAW-CC for any model, including Mammo-AGE and MIRAI.

Nevertheless, we have also performed the supplementary experiments to explore the learnability and adaptability of the MIRAI model by finetuning it on our internal finetuning set (Inhouse and EMBED datasets). We observed that the finetuned MIRAI was prone to overfitting on the internal datasets. Specifically, the risk prediction results (Table R3.2.1) showed that finetuning slightly improved MIRAI's performance on the internal datasets, e.g., a 4% increase in 2-year AUPRC on both the EMBED and Inhouse datasets, and a decline in performance was observed on the external CSAW-CC dataset after finetuning MIRAI. In contrast, our proposed method consistently outperformed both the finetuned MIRAI and original MIRAI models.

Dataset	Risk Model	2 Year AUPRC	5 Year AUPRC	2 Year AUROC	5 Year AUROC
Inhouse	SOTA	0.31 (0.22-0.38)	0.33 (0.27-0.39)	0.76 (0.72-0.80)	0.73 (0.70-0.76)
	SOTA-Finetune	0.35 (0.28-0.43)	0.33 (0.27-0.38)	0.78 (0.74-0.82)	0.71 (0.68-0.75)
	Baseline	0.26 (0.20-0.34)	0.27 (0.21-0.32)	0.71 (0.66-0.76)	0.68 (0.63-0.72)
	SSL	0.41 (0.34-0.48)	0.36 (0.32-0.41)	0.80 (0.77-0.83)	0.71 (0.69-0.74)
	Ours	0.44 (0.38-0.52)	0.40 (0.35-0.45)	0.82 (0.80-0.84)	0.75 (0.74-0.77)
EMBED	SOTA	0.24 (0.15-0.33)	0.26 (0.18-0.32)	0.76 (0.71-0.80)	0.75 (0.71-0.79)
	SOTA-Finetune	0.28 (0.19-0.37)	0.27 (0.20-0.34)	0.75 (0.69-0.80)	0.72 (0.66-0.76)
	Baseline	0.21 (0.15-0.27)	0.22 (0.17-0.28)	0.75 (0.71-0.79)	0.75 (0.71-0.78)
	SSL	0.29 (0.23-0.36)	0.26 (0.21-0.32)	0.76 (0.71-0.80)	0.70 (0.66-0.74)
	Ours	0.34 (0.24-0.42)	0.32 (0.24-0.39)	0.78 (0.72-0.81)	0.77 (0.72-0.81)

	SOTA	0.40 (0.37-0.45)	0.34 (0.32-0.36)	0.86 (0.85-0.87)	0.77 (0.76-0.78)
	SOTA-Finetune	0.31 (0.28-0.32)	0.30 (0.28-0.32)	0.83 (0.81-0.85)	0.73 (0.72-0.75)
CSAW-CC	Baseline	0.31 (0.28-0.33)	0.26 (0.24-0.28)	0.77 (0.75-0.80)	0.70 (0.69-0.71)
	SSL	0.46 (0.43-0.49)	0.33 (0.31-0.35)	0.85 (0.83-0.86)	0.71 (0.70-0.73)
	Ours	0.51 (0.47-0.53)	0.38 (0.35-0.40)	0.87 (0.86-0.88)	0.76 (0.74-0.77)

Table R3.2.1 Comparison results of short-term (2-year) and long-term (5-year) BC risk prediction with baseline, SOTA risk model¹ and SOTA self-supervised learning (SSL) model⁵. *Baseline*: the model was trained for risk prediction from scratch without using age prediction pretrained weights; *Ours*: we finetuned the model with age prediction pretrained weights for risk prediction. *SOTA model*: MIRAI risk prediction model, tested directly. *SOTA-Finetune model*: Finetuned MIRAI on the same finetuning set. *SSL model*: We applied the Mammo-clip model, which was pretrained with mammograms and radiology reports. AUPRC: Area Under Precision-Recall Curve; AUROC: Area Under Receiver Operating Characteristic Curve.

Therefore, evaluating MIRAI with its original trained weights and reserving CSAW-CC exclusively for testing provides a rigorous and fair benchmark aligned with real-world application scenarios.

Action taken: We have added the following text in the validation section related to Figure 7, on page 21, lines 441–445: “Given that MIRAI is already a well-validated model trained on large and diverse cohorts^{1,2}, we evaluated it without additional finetuning to preserve its benchmark integrity. To ensure a fair comparison and reflect real-world usage, the CSAW-CC (risk and diagnostic tasks) and CMMD (diagnostic task) datasets were used exclusively as external test sets. No training or finetuning was performed on CSAW-CC and CMMD for any of the models.”

* What do the error bars in the figure represent?

The error bars represent the 95% confidence intervals (CI) of AUROC and AUPRC metrics, estimated using 1,000 bootstrap resamples for each measure⁶. We have provided this information in the Methods section (page 21, line 446) and added to the corresponding figure caption (page 30, lines 753–754).

Reviewer #4:

Thank you for the time and effort spent reviewing our manuscript.

Reviewer #5:

Comments for Authors: This paper focuses on estimating breast age using healthy mammograms via deep learning techniques. The authors developed models based on internal cross-validation datasets and utilized external datasets for further validation purposes. Based on the current version, the authors have made efforts to address the reviewers' comments from the previous round of review. In this report, I have some additional comments and hope to improve the quality of the statistical analyses involved in the paper.

We sincerely thank the referee for their positive and encouraging feedback on our study and for acknowledging our efforts in performing rigorous model cross-validation and external validation. We appreciate their professional suggestions for the statistical analyses.

1. If I understand correctly, the authors stratified the risk of 10 and 5 years of breast cancer in two different datasets. The use of a categorical variable could be a simpler and straightforward alternative. What is the authors' opinion?

We appreciate this insightful suggestion. We agree that using a categorical variable based on the breast age gap (e.g., dichotomizing at the median) offers a simple and intuitive approach to stratifying breast cancer risk. Both the categorical- (Fig. R5.1.1 and Table R5.1.1) and continuous- (Table R5.1.1) based analyses were included in the manuscript (Fig. 6C and Table 3), and we can also refer to the previous response (#R1-Comment14).

Specifically, **in the categorical analysis**, we dichotomized the breast age gap at the median and stratified women into high- and low-risk groups in both datasets. Kaplan–Meier analysis (Fig. R5.1.1) and Cox proportional hazards models revealed significantly higher cancer incidence in the high-gap group ($p < 0.05$ in both datasets; Table R5.1.1).

In addition, treating the breast age gap as a **continuous variable** also provided clinically meaningful and precise insights. As shown in Table R5.1.1, each 1-year increase in breast age gap was associated with a 1.6% and 2.0% increase in breast cancer risk on the inhouse and EMBED datasets, respectively (Inhouse dataset: HR = 1.016 [95% CI: 1.009-1.023], P < 0.001; EMBED dataset: HR = 1.020 [95% CI: 1.008-1.033], P = 0.002; Model 2).

Overall, these findings suggest that both categorical and continuous approaches are useful, offering complementary insights into risk stratification.

BreastAGE Years, mean±SD	N	Events	Inc.	Model 1 (Unadj.)		Model 2 (CA-ACR-adj.)		Model 3 (RF-adj.)		
				HR (95%CI)	P value	HR (95%CI)	P value	HR (95%CI)	P value	
Inhouse BC events (10Ys)	0.51±4.55	10,392	3,656	35.2	-	-	-	-	-	
Age gap, per one age (Ys)	-	-	-	-	1.013 (1.006-1.020)	< 0.001	1.016 (1.009-1.023)	< 0.001	1.014 (1.007-1.022)	< 0.001
Low-risk group (Gap < 0.36)	-3.08±2.70	5,196	1,722	33.1	Reference	-	Reference	-	Reference	-
High-risk group (Gap > 0.36)	4.10±2.88	5,196	1,934	37.2	1.164 (1.091-1.242)	< 0.001	1.157 (1.084-1.235)	< 0.001	1.140 (1.067-1.217)	< 0.001
EMBED BC events (5Ys)	0.08±5.025	58,983	1,029	3.49	-	-	-	-	-	
Age gap, per one age (Ys)	-	-	-	-	1.022(1.001-1.034)	< 0.001	1.020 (1.008-1.033)	0.002	1.020 (1.008-1.033)	0.002
Low-risk group (Gap < -0.01)	-3.75±2.85	29,492	463	3.14	Reference	-	Reference	-	Reference	-
High-risk group (Gap > -0.01)	3.88±3.06	29,491	566	3.84	1.225 (1.083-1.385)	< 0.001	1.152 (1.071-1.305)	0.026	1.151 (1.016-1.304)	0.027

Inc = incidence per 1,000 person-years; CI = confidence interval; BC =breast cancer; HR = hazard ratio; Unadj. HR = unadjusted HR; CA-ACR-adj. HR =HR adjusted HR on chronological age (CA) and breast density (ACR); RF-adj. HR (Inhouse) = HR adjusted HR on more risk factors for inhouse dataset, including CA, ACR, Race, Gene mutation, Menarche, Menopausal status, and Manufacturer of mammograms; RF-adj. HR (EMBED) = HR adjusted HR on more risk factors for EMBED dataset, including CA, ACR, Ethnicity, and Manufacturer of mammograms;
BreastAGE = deep learning-based breast biological age.

Table R5.1.1 Association between the breast age gap and future breast cancer using Cox proportional hazards regression models.

2. Have the authors’ assessed whether the underlying Cox proportional hazard assumption is satisfied? Any sensitivity analyses and reflection on how investigators could tell when the proposed method will fail to work?

We appreciate the reviewer for this professional suggestion. Yes, the assumption of proportional hazards was verified by visual inspection and by testing the Schoenfeld residuals^{7,8}. For covariates that violated this assumption, such as chronological age, we employed stratification in the Cox models to account for their nonproportionality.

To assess the proportional hazard assumption, we plotted the estimates of the time-dependent coefficients of the model, adjusted for chronological age and breast density, for the future breast cancer event, and tested the association between the Schoenfeld residuals and time (**Fig. R5.2.1**). The Schoenfeld residuals were visually independent of time and their correlations were statistically non-significant, indicating that the assumption was satisfied.

Fig. R5.2.1 Association between the breast age gap and future breast cancer using Cox proportional hazards regression models. Estimates of the time-dependent coefficients of the models adjusted for chronological age and breast density for outcomes on two datasets. The p-values corresponded to the chi-squared test on the correlation between the Schoenfeld residuals and time.

Furthermore, we conducted several sensitivity analyses to assess the robustness and generalizability of our findings:

Covariate adjustment: We adjusted for chronological age, breast density, and additional known risk factors in multivariable Cox model analyses (Table 3). Importantly, these results suggest that positive breast age gaps (where the breast appears older than the chronological age) are associated with a substantially increased risk of breast cancer ($P < 0.05$). Further adjusting for additional breast cancer risk factors (Table 3, Model 3) did not substantially alter the association between breast age gap and cancer risk, supporting its robustness and independence.

Dataset-specific validation: We repeated the risk analyses (Table 3) using both Inhouse and the EMBED datasets. The consistent results across both cohorts reinforce the generalizability of the association between breast age gap and future cancer risk.

The above sensitivity analyses and cross-dataset validations provide evidence that the Cox models we used are appropriate and stable under varying assumptions and populations.

Action taken: Considering the suggestion from the reviewer, as well as the bioinformatician and data analyst in the team, we have added Figure R5.2.1 to the Supplementary Material (Fig. S7) to visualize and support the assessment of the Cox proportional hazards assumption. Page 18, lines 391-392: “To ensure the validity of the Cox model estimates, the proportional hazards assumption was verified by visual inspection and Schoenfeld residuals test^{7,8} (as shown in Fig. S7).”

We have also updated the descriptions of the statistical analysis in the method section (page 30, lines 748–751) to include the following description: “We verified the proportional hazards assumption using the Schoenfeld residuals test^{7,8}. For covariates that violated this assumption, we employed stratification in the Cox models to account for their nonproportionality. Several sensitivity analyses were conducted, including covariate adjustments and dataset-specific model validation.”

3. Line 135: What is the bias-variance consideration when performing cross-validation? Why was a five-fold validation chosen over a ten-fold validation?

We appreciate the reviewer's attention to this good point. In selecting five-fold cross-validation, we considered both computational feasibility and empirical stability, consistent with standard practice in large-scale medical imaging studies (e.g., Lee et al., *Nat. Aging*, 2022; Argentieri et al., *Nat. Med.*, 2022)^{9,10}.

We acknowledge the bias–variance trade-off inherent in cross-validation design^{11,12}. While increasing the number of folds (e.g., from five to ten) may theoretically reduce the variance of validation error estimates, in the context of large-scale datasets and computationally intensive deep learning models, this comes at the cost of substantially higher computational demands. In our study, which involves over 140,000 mammographic exams, each fold requires approximately 180 GPU (A6000 48GB) hours to train an ensemble Mammo-AGE model. Given the available computational resources and extensive comparison / ablation experiments conducted, ten-fold cross-validation was considered not computationally efficient.

Importantly, the model demonstrated strong stability and reproducibility under five-fold cross-validation, with a mean absolute error (MAE) of 4.174 ± 0.028 (Table 2, in the manuscript). It consistently outperformed other methods across both multiple internal datasets and independent external datasets, with low variance, demonstrating the robustness and generalizability of our model. Furthermore, the subgroup analysis showed the robustness of the model across diverse subsets (Fig. S3, S4). This evidence supports the appropriateness of our choice to use five-fold cross-validation, achieving a balance between computational efficiency and estimation reliability.

Action taken: We have revised the Methods section to include the following explanation, page 25, lines 596-598: *“Considering that the breast age model is trained on a large-scale population, five-fold cross-validation was adopted to balance computational cost (approximately 180 GPU hours per fold of an ensemble Mammo-AGE model) and estimation stability, following common practice in large-scale imaging studies^{9,10}.”*

4. The authors reported the Pearson correlation coefficient, which is based on the linearity assumption. Would there be a change in conclusions should more robust nonparametric correlation metrics, such as Spearman rho, be employed?

We agree with the reviewer that Pearson correlation assumes linearity and that nonparametric alternatives such as Spearman's rank correlation can provide a more robust assessment. We have now conducted additional analyses using Spearman correlation coefficients, which yielded similar results and trends to those based on Pearson correlation (e.g., as shown in **Table R5.4.1**).

Additionally, we have included the Spearman correlation results in Table 2 of the manuscript, reporting model performance across different baseline models and individual backbone-based Mammo-AGE models.

Action taken: We have added the Spearman correlation analysis and the detailed results to the manuscript (page 9, Table 2) and supplementary material (Table S4). Page 7, Lines 163-167: *“To evaluate the Mammo-AGE model, we employed four metrics: ... and the Spearman's rank correlation coefficient (r_s) between predicted breast age and chronological age. ... and $r_s = 0.896 \pm 0.001$ on the combined test set (Table 2).”*

Page 30, Lines 737-740: “The Pearson correlation coefficient (r) is calculated based on the assumed linear relationship between predicted breast age and chronological age. The nonparametric based Spearman’s rank correlation coefficient (r_s) is also reported.”

Method		Combined Dataset	Inhouse Dataset	RSNA Dataset	VinDr Dataset	EMBED Dataset	CMMD Dataset	
Compare with other methods (ResNet18)	Single view	MAE	7.429±0.156	6.988±0.134	8.362±0.300	7.991±0.373	9.741±0.338	7.213±0.354
		Pearson Correlation (r)	0.676±0.017	0.718±0.011	0.591±0.028	0.509±0.031	0.600±0.024	0.438±0.048
		Spearman Correlation (r_s)	0.704±0.017	0.740±0.010	0.602±0.025	0.509±0.021	0.610±0.025	0.400±0.034
		CS ($\alpha=5$ year)	45.6%±1.0%	47.0%±1.0%	40.4%±1.4%	46.0%±2.1%	36.0%±1.3%	49.4%±2.1%
	POE	MAE	5.689±0.064	5.460±0.025	5.872±0.056	6.303±0.297	6.376±0.030	6.054±0.122
		Pearson Correlation (r)	0.786±0.005	0.799±0.002	0.729±0.005	0.646±0.008	0.748±0.002	0.594±0.012
		Spearman Correlation (r_s)	0.795±0.005	0.809±0.002	0.729±0.006	0.642±0.008	0.752±0.002	0.521±0.006
		CS ($\alpha=5$ year)	58.1%±0.5%	59.5%±0.3%	56.5%±0.7%	55.1%±2.3%	52.8%±0.2%	54.1%±1.3%
	MVL	MAE	5.578±0.037	5.406±0.030	5.767±0.043	5.985±0.094	6.313±0.024	6.005±0.079
		Pearson Correlation (r)	0.796±0.004	0.805±0.004	0.741±0.005	0.658±0.008	0.759±0.001	0.595±0.006
		Spearman Correlation (r_s)	0.804±0.005	0.814±0.003	0.741±0.006	0.656±0.011	0.763±0.001	0.528±0.007
		CS ($\alpha=5$ year)	58.9%±0.2%	59.7%±0.2%	57.5%±0.6%	57.7%±0.9%	53.1%±0.2%	54.1%±1.3%
	GLT	MAE	5.629±0.036	5.491±0.021	5.806±0.018	5.927±0.133	6.349±0.047	6.095±0.144
		Pearson Correlation (r)	0.793±0.002	0.800±0.001	0.738±0.002	0.661±0.008	0.753±0.002	0.598±0.016
		Spearman Correlation (r_s)	0.803±0.002	0.810±0.001	0.737±0.002	0.656±0.011	0.758±0.002	0.526±0.011
		CS ($\alpha=5$ year)	58.3%±0.3%	58.8%±0.3%	56.9%±0.6%	58.0%±0.7%	52.8%±0.4%	54.2%±1.1%
	Multi-view	MAE	5.079±0.035	4.841±0.051	5.514±0.083	5.456±0.141	5.944±0.112	6.192±0.136
		Pearson Correlation (r)	0.829±0.004	0.845±0.006	0.758±0.007	0.680±0.011	0.785±0.004	0.602±0.018
Spearman Correlation (r_s)		0.838±0.004	0.852±0.006	0.761±0.008	0.689±0.010	0.792±0.004	0.531±0.022	
	CS ($\alpha=5$ year)	63.7%±0.2%	65.0%±0.4%	60.1%±0.6%	62.7%±1.3%	55.9%±1.1%	53.0%±1.7%	
Ours ensembled Mammo-AGE	MAE	4.174±0.028	4.090±0.047	4.548±0.022	4.075±0.010	5.010±0.040	6.103±0.222	
	Pearson Correlation (r)	0.891±0.001	0.892±0.002	0.844±0.001	0.847±0.002	0.855±0.002	0.705±0.009	
	Spearman Correlation (r_s)	0.896±0.001	0.898±0.002	0.849±0.002	0.843±0.003	0.861±0.002	0.634±0.009	
	CS ($\alpha=5$ year)	72.2%±0.7%	73.0%±0.8%	68.2%±0.7%	73.8%±0.8%	63.4%±0.5%	54.8%±1.5%	

Table R5.4.1 Comparisons of model performance for different methods. Ours: Mammo-AGE model; SV: single-view-based baseline method. POE: probabilistic ordinal embedding method; MVL: mean-variance loss method; GLT: global-local transformer method; MV: multi-view-based baseline method; MAE: mean absolute error, lower is better; CS: cumulative score based on a threshold of error within five years ($\alpha=5y$), higher is better. Pearson and Spearman correlation coefficients between predicted breast age and chronological age are also reported.

5. I appreciate the authors’ response to the reviewer, explaining more details regarding the global-local transformer (GLT) framework. I agree this can perform decently in vision tasks. Still, I also feel that the work heavily focuses on recent and perhaps non-standard deep learning techniques. Would the work be extended if using alternative deep learning techniques, such as CNN-based advanced models, or other transformer-based variants?

We appreciate the reviewer acknowledging that our designed model leverages the recent novel global-local transformer framework. Moreover, our model also incorporates various advanced CNN-backbones (e.g., EfficientNet¹³, ConvNeXt¹⁴, DenseNet¹⁵, and ResNet¹⁶). Following the reviewer’s suggestion, we have further compared our method with additional advanced standard techniques (as shown in Table R5.5.1).

Specifically, in addition to ResNet-18, which was included in our initial comparison experiments (Fig 3., i.e. “SV” method), we have further compared our method with more advanced CNN models (Table R5.5.1). Furthermore, we have also extended the comparison to include two transformer-based variants: the Vision Transformer (ViT) ¹⁷ and the Swin Transformer (Swin-ViT) ¹⁸. All these experiments were conducted using official implementations with pretrained weights from the PyTorch model library (<https://docs.pytorch.org/vision/main/models.html>). These results show that our proposed architectures consistently outperform both standard CNNs and ViTs, demonstrating that the performance gain from the synergistic design involving multi-view integration, instance-bag transformer, multi-task learning, and ensemble prediction is stable. For more detailed results across different dataset, please refer to the revised Manuscript (Table 2).

Model	MAE	Pearson Correlation (r)	Spearman Correlation	CS % ($\alpha=5$ year)
ResNet-18 ¹⁶	7.429±0.156	0.676±0.017	0.704±0.017	45.6%±1.0%
ResNet-50 ¹⁶	9.029±0.175	0.593±0.042	0.638±0.030	37.1%±0.7%
DenseNet-121 ¹⁵	8.144±0.313	0.657±0.012	0.705±0.010	41.2%±2.0%
ConvNeXt-Tiny ¹⁴	6.152±0.245	0.741±0.018	0.760±0.017	55.0%±1.9%
EfficientNet-B0 ¹³	6.109±0.128	0.747±0.013	0.769±0.011	55.3%±1.1%
ViT ¹⁷	6.998±0.030	0.672±0.003	0.686±0.003	49.2%±0.3%
Swin-ViT ¹⁸	6.541±0.058	0.718±0.004	0.729±0.004	52.2%±0.5%
Ours (Mammo-AGE)	4.174±0.028	0.891±0.001	0.896±0.001	72.2%±0.7%

Table R5.5.1 Comparison of model performance across baseline methods and Mammo-AGE. MAE: mean absolute error, lower is better; CS: cumulative score based on a threshold of error within five years ($\alpha=5y$), higher is better. Pearson and Spearman correlation coefficients between predicted breast age and chronological age are also reported.

Action taken: We have included these new experimental results in the revised manuscript (Table 2) and highlighted the comparative advantages of our proposed architecture over standard deep learning frameworks.

Page 8, lines 190-197: “Our ensemble Mammo-AGE incorporates various advanced convolutional neural network (CNN) backbones (e.g., EfficientNet ¹³, ConvNeXt ¹⁴, DenseNet ¹⁵, and ResNet ¹⁶). The detailed results for each CNN backbone-based Mammo-AGE are presented in Table 2. To evaluate the effectiveness of our proposed framework, we compared each backbone-based Mammo-AGE model with its corresponding CNN counterpart, including ResNet-18 ¹⁶, ResNet-50 ¹⁶, EfficientNet-b0 ¹³, ConvNeXt-tiny ¹⁴, and DenseNet-121 ¹⁵. Furthermore, we extended the comparison to include two transformer-based variants: Vision Transformer (ViT) ¹⁷ and Swin Transformer (Swin-ViT) ¹⁸. All these experiments were conducted using official implementations with pretrained weights. These results show that our proposed architecture consistently outperforms both standard CNNs and ViTs.”

Reference

1. Yala, A. *et al.* Toward robust mammography-based models for breast cancer risk. *Sci. Transl. Med.* **13**, eaba4373 (2021).
2. Yala, A. *et al.* Multi-Institutional Validation of a Mammography-Based Breast Cancer Risk Model. *J. Clin. Oncol.* 8–10 (2021) doi:10.1200/jco.21.01337.
3. Dembrower, K., Lindholm, P. & Strand, F. A multi-million mammography image dataset and population-based screening cohort for the training and evaluation of deep neural networks—the cohort of screen-aged women (CSAW). *Journal of digital imaging* **33**, 408–413 (2020).
4. Strand, F. CSAW-CC (mammografi) – ett dataset för AI-forskning för att förbättra screening, diagnostik och prognostik för bröstcancer CSAW-CC (mammography) – a dataset for AI research to improve screening, diagnostics and prognostics of breast cancer. 8.81 MiB, 19 variables, 8723 cases Karolinska Institutet <https://doi.org/10.5878/45VM-T798> (2022).
5. Ghosh, S., Poynton, C. B., Visweswaran, S. & Batmanghelich, K. Mammo-CLIP: A Vision Language Foundation Model to Enhance Data Efficiency and Robustness in Mammography. in *Medical Image Computing and Computer Assisted Intervention – MICCAI 2024* (eds. Linguraru, M. G. *et al.*) vol. 15012 632–642 (Springer Nature Switzerland, Cham, 2024).
6. Carpenter, J. & Bithell, J. Bootstrap confidence intervals: When, which, what? A practical guide for medical statisticians. *Statist. Med.* **19**, 1141–1164 (2000).
7. Zhu, Z. *et al.* Retinal age gap as a predictive biomarker for mortality risk. *Br. J. Ophthalmol.* **107**, 547–554 (2023).
8. Nusinovici, S. *et al.* Application of a deep-learning marker for morbidity and mortality prediction derived from retinal photographs: A cohort development and validation study. *The Lancet Healthy Longevity* **5**, 100593 (2024).
9. Lee, J. *et al.* Deep learning-based brain age prediction in normal aging and dementia. *Nature Aging* **2**, 412–424 (2022).
10. Argentieri, M. A. *et al.* Proteomic aging clock predicts mortality and risk of common age-related diseases in diverse populations. *Nat Med* **30**, 2450–2460 (2024).
11. Bradshaw, T. J., Huemann, Z., Hu, J. & Rahmim, A. A guide to cross-validation for artificial intelligence in medical imaging. *Radiology: Artificial Intelligence* **5**, e220232 (2023).
12. Bates, S., Hastie, T. & Tibshirani, R. Cross-validation: What does it estimate and how well does it do it? *Journal of the American Statistical Association* **119**, 1434–1445 (2024).
13. Tan, M. & Le, Q. Efficientnet: Rethinking model scaling for convolutional neural networks. in *International conference on machine learning* 6105–6114 (PMLR, 2019).

14. Liu, Z. *et al.* A convnet for the 2020s. in *Proceedings of the IEEE/CVF conference on computer vision and pattern recognition* 11976–11986 (2022).
15. Huang, G., Liu, Z., Van Der Maaten, L. & Weinberger, K. Q. Densely connected convolutional networks. in *Proceedings of the IEEE conference on computer vision and pattern recognition* 4700–4708 (2017).
16. He, K., Zhang, X., Ren, S. & Sun, J. Deep residual learning for image recognition. in *Proceedings of the IEEE conference on computer vision and pattern recognition* 770–778 (2016).
17. Dosovitskiy, A. *et al.* An image is worth 16x16 words: Transformers for image recognition at scale. Preprint at <https://doi.org/10.48550/arXiv.2010.11929> (2021).
18. Liu, Z. *et al.* Swin transformer: Hierarchical vision transformer using shifted windows. in *2021 IEEE/CVF International Conference on Computer Vision (ICCV)* 9992–10002 (IEEE, Montreal, QC, Canada, 2021). doi:10.1109/ICCV48922.2021.00986.

Comments for Authors:

This paper focuses on estimating breast age using healthy mammograms via deep learning techniques. The authors developed models based on internal cross-validation datasets and utilized external datasets for further validation purposes. Based on the current version, the authors have made efforts to address the reviewers' comments from the previous round of review. In this report, I have some additional comments and hope to improve the quality of the statistical analyses involved in the paper.

1. If I understand correctly, the authors stratified the risk of 10 and 5 years of breast cancer in two different datasets. The use of a categorical variable could be a simpler and straightforward alternative. What is the authors' opinion?
2. Have the authors' assessed whether the underlying Cox proportional hazard assumption is satisfied? Any sensitivity analyses and reflection on how investigators could tell when the proposed method will fail to work?
3. Line 135: What is the bias-variance consideration when performing cross-validation? Why was a five-fold validation chosen over a ten-fold validation?
4. The authors reported the Pearson correlation coefficient, which is based on the linearity assumption. Would there be a change in conclusions should more robust nonparametric correlation metrics, such as Spearman rho, be employed?
5. I appreciate the authors' response to the reviewer, explaining more details regarding the global-local transformer (GLT) framework. I agree this can perform decently in vision tasks. Still, I also feel that the work heavily focuses on recent and perhaps non-standard deep learning techniques. Would the work be extended if using alternative deep learning techniques, such as CNN-based advanced models, or other transformer-based variants?

References for Authors' Information